# Provably Efficient Lifelong Reinforcement Learning with Linear Representation

**Sanae Amani**
University of California, Los Angeles
samani@ucla.edu

**Lin F. Yang**
University of California, Los Angeles
linyang@ee.ucla.edu

**Ching-An Cheng**
Microsoft Research, Redmond
chinganc@microsoft.com

## ABSTRACT

We theoretically study lifelong reinforcement learning (RL) with linear representation in a regret minimization setting. The goal of the agent is to learn a multi-task policy based on a linear representation while solving a sequence of tasks that may be adaptively chosen based on the agent's past behaviors. We frame the problem as a linearly parameterized contextual Markov decision process (MDP), where each task is specified by a context and the transition dynamics is context-independent, and we introduce a new completeness-style assumption on the representation which is sufficient to ensure the optimal multi-task policy is realizable under the linear representation. Under this assumption, we propose an algorithm, called UCB Lifelong Value Distillation (UCBlvd), that provably achieves sublinear regret for any sequence of tasks while using only sublinear planning calls. Specifically, for $K$ task episodes of horizon $H$, our algorithm has a regret bound $\tilde{\mathcal{O}}(\sqrt{(d^3 + d'd)H^4K})$ based on $\mathcal{O}(dH \log(K))$ number of planning calls, where $d$ and $d'$ are the feature dimensions of the dynamics and rewards, respectively. This theoretical guarantee implies that our algorithm can enable a lifelong learning agent to learn to internalize experiences into a multi-task policy and rapidly solve new tasks.

## 1 INTRODUCTION

Recently, there has been a surging interest in designing *lifelong learning* agents that can continuously learn to solve multiple sequential decision making problems in their lifetimes (Thrun & Mitchell, 1995; Khetarpal et al., 2020; Silver et al., 2013; Xie & Finn, 2021). This scenario is in particular motivated by building multi-purpose embodied intelligence, such as robots working in a weakly structured environment (Roy et al., 2021). Typically, curating all tasks beforehand for such problems is nearly infeasible, and the problems the agent is tasked with may be adaptively selected based on the agent's past behaviors. Consider a household robot as an example. Since each household is unique, it is difficult to anticipate upfront all scenarios the robot would encounter. Moreover, the tasks the robot faces are not independent and identically distributed (i.i.d.). Instead, what the robot has done before can affect the next task and its starting state; e.g., if the robot fails to bring a glass of water and breaks it, then the user is likely to command the robot to clean up the mess. Thus, it is critical that the agent continuously improves and generalizes learned abilities to different tasks, regardless of their order.

In this work, we theoretically study lifelong reinforcement learning (RL) in a regret minimization setting (Thrun & Mitchell, 1995; Ammar et al., 2015), where the agent needs to solve a sequence of tasks using rewards in an unknown environment while balancing exploration and exploitation. Motivated by the embodied intelligence scenario, we suppose that tasks differ in rewards, but share the same state and action spaces and transition dynamics (Xie & Finn, 2021).To be realistic, we make *no* assumptions on how the tasks and initial states are selected[1]; generally we allow them to be chosen from a continuous set by an adversary based on the agent's past behaviors. Once a task is specified

---

[1]We adopt a stricter definition of lifelong RL here to distinguish it from multi-task RL, while there are existing works on lifelong RL (e.g. Brunskill & Li (2014); Lecarpentier et al. (2021)) assuming i.i.d. tasks.

and revealed, the agent has one chance (i.e., executing one rollout from its current state) to complete the task and then it moves to the next task.

The agent's goal is to perform near optimally for the tasks it faces, despite the online nature of the problem. This means that the accumulated regret of the learner compared with the best policy for each task should be sublinear in its lifetime. We assume that there is no memory constraint; this is usually the case for robotics applications where real-world interactions are the main bottleneck (Xie & Finn, 2021). Nonetheless, we require that the agent eventually learns to make decisions without frequent deliberate planning, because planning is time consuming and creates undesirable wait time for user-interactive scenarios. In other words, the agent needs to learn a multi-task policy, generalizing from not only past samples but also past computation, to solve new tasks.

Formally, we consider an episodic setup based on the framework of contextual Markov decision process (CMDP) (Abbasi-Yadkori & Neu, 2014; Hallak et al., 2015). It repeats the following steps: *1)* At the beginning of an episode, the agent is set to an initial state and receives a context specifying the task reward, both of which can be arbitrarily chosen. *2)* When needed, the agent uses its past experiences to plan for the current task. *3)* The agent runs a policy in the environment for a fixed horizon in an attempt to solve the assigned task and gains experience from its policy execution. The agent's performance is measured as the regret with respect to the optimal policy of the corresponding task. We require that, for *any* task sequence, *both* the agent's overall regret and number of planning calls to be sublinear in the number of episodes.

While lifelong RL is not new, the realistic need of *simultaneously* achieving *1)* sublinear regret and *2)* sublinear number of planning calls for *3)* a potentially adversarial sequence of tasks and initial states makes the setup considered here particularly challenging. To our knowledge, existing works only address a strict subset of these requirements; especially, the computation aspect is often ignored. Most provable works in lifelong RL make the assumption that the tasks are finitely many (Ammar et al., 2015; Zhan et al., 2017; Brunskill & Li, 2015), or are i.i.d. (Ammar et al., 2014; Brunskill & Li, 2014; Abel et al., 2018a;b; Lecarpentier et al., 2021), while others considering similar setups to ours do not provide regret guarantees (Isele et al., 2016; Xie & Finn, 2021). On the technical side, the closest lines of work are Modi & Tewari (2020); Abbasi-Yadkori & Neu (2014); Hallak et al. (2015); Modi et al. (2018); Kakade et al. (2020) for contextual MDP and Wu et al. (2021); Abels et al. (2019) for the dynamic setting of multi-objective RL, which study the sample complexity for arbitrary task sequences; however, they either assume the problem is tabular or require a model-based planning oracle with unknown complexity. Importantly, none of the existing works properly addresses the need of sublinear planning calls, which creates a large gap between the abstract setup and practice need.

In this paper, we aim to establish a foundation for designing agents meeting these three practically important requirements, a problem which has been overlooked in the literature. As the first step, here we study lifelong RL with linear representation. We suppose that the contextual MDP is linearly parameterized (Yang & Wang, 2019; Jin et al., 2020) and the agent needs to learn a multi-task policy based on this linear representation. To make this possible, we introduce a new completeness-style assumption on the representation which is sufficient to ensure the optimal multi-task policy is realizable under the linear representation. Under these assumptions, we propose the first provably efficient lifelong RL algorithm, **U**pper **C**onfidence **B**ound **L**ifelong **V**alue **D**istillation (UCBlvd, pronounced as "UC Boulevard"), that possesses all three desired qualities. Specifically, for $K$ episodes of horizon $H$, we prove a regret bound $\tilde{\mathcal{O}}(\sqrt{(d^3 + d'd)H^4K})$ using $\tilde{\mathcal{O}}(dH \log(K))$ planning calls, where $d$ and $d'$ are the feature dimensions of the dynamics and rewards, respectively.

From a high-level viewpoint, UCBlvd uses a linear structure to identify what to transfer and operates by interleaving *1)* independent planning for a set of representative tasks and *2)* distilling the planned results into a multi-task value-based policy. UCBlvd also constantly monitors whether the new experiences it gained are sufficiently significant, based on a doubling schedule, to avoid unnecessary planning. On the technical side, UCBlvd's design is inspired by single-task LSVI-UCB (Jin et al., 2020), however, we introduce a novel distillation step based on QCQP, along with a new completeness assumption, to enable computation sharing across tasks; we also extend the low-switching cost technique (Abbasi-Yadkori et al., 2011; Gao et al., 2021; Wang et al., 2021) for single-task RL to the lifelong setup to achieve sublinear number of planning calls.

**Notation.** Throughout the paper, we use lower-case letters for scalars, lower-case bold letters for vectors, and upper-case bold letters for matrices. The Euclidean-norm of $\mathbf{x}$ is denoted by $\|\mathbf{x}\|_2$. We denote the transpose of a vector $\mathbf{x}$ by $\mathbf{x}^\top$. For any vectors $\mathbf{x}$ and $\mathbf{y}$, we use $\langle \mathbf{x}, \mathbf{y} \rangle$ to denote their

inner product. We denote the Kronecker product by $\mathbf{A} \otimes \mathbf{B}$. Let $\mathbf{A} \in \mathbb{R}^{d \times d}$ be a positive definite and $\boldsymbol{\nu} \in \mathbb{R}^d$. The weighted 2-norm of $\boldsymbol{\nu}$ with respect to $\mathbf{A}$ is defined by $\|\boldsymbol{\nu}\|_{\mathbf{A}} := \sqrt{\boldsymbol{\nu}^\top \mathbf{A} \boldsymbol{\nu}}$. For a positive integer $n$, $[n]$ denotes the $\{1, 2, \ldots, n\}$. For a real number $\alpha$, we denote $\{\alpha\}^+ = \max\{\alpha, 0\}$. Finally, we use the notation $\tilde{\mathcal{O}}$ for big-O notation that ignores logarithmic factors.

## 2 PRELIMINARIES

We formulate lifelong RL as a regret minimization problem in contextual MDP (Abbasi-Yadkori & Neu, 2014; Hallak et al., 2015) with adversarial context and initial state sequences. We suppose that a context determines the task reward but does not affect the dynamics. Such a context dependency is common for the lifelong learning scenario where an embodied agent consecutively solves multiple tasks. Below we give the formal problem definition.

**Finite-horizon contextual MDP.** We consider a finite-horizon contextual MDP denoted by $M = (\mathcal{S}, \mathcal{A}, \mathcal{W}, H, \mathbb{P}, r)$, where $\mathcal{S}$ is the state space, $\mathcal{A}$ is the action space, $\mathcal{W}$ is the task context space, $H$ is the horizon (length of each episode), $\mathbb{P} = \{\mathbb{P}_h\}_{h=1}^H$ are the transition probabilities, and $r = \{r_h\}_{h=1}^H$ are the reward functions. We allow $\mathcal{S}$ and $\mathcal{W}$ to be continuous or infinitely large, while we assume $\mathcal{A}$ is finite such that $\max_{a \in \mathcal{A}}$ can be performed easily. For $h \in [H]$, $r_h(s, a, w)$ denotes the reward function whose range is assumed to be in $[0, 1]$, and $\mathbb{P}_h(s'|s, a)$ denotes the probability of transitioning to state $s'$ upon playing action $a$ at state $s$. In short, a contextual MDP can be viewed as an MDP with state space $\mathcal{S} \times \mathcal{W}$ and action space $\mathcal{A}$ where the context part of the state remains constant in an episode.[2] To simplify the notation, for any function $f$, we write $\mathbb{P}_h[f](s, a) := \mathbb{E}_{s' \sim \mathbb{P}_h(.|s,a)}[f(s')]$.

**Policy and value functions.** In a finite-horizon contextual MDP, a policy $\pi = \{\pi_h\}_{h=1}^H$ is a sequence where $\pi_h : \mathcal{S} \times \mathcal{W} \to \mathcal{A}$ determines the agent's action at time-step $h$. Given $\pi$, we define its state value function as $V_h^\pi(s, w) := \mathbb{E}[\sum_{h'=h}^H r_{h'}(s_{h'}, \pi_{h'}(s_{h'}, w), w)|s_h = s]$ and its action-value function as $Q_h^\pi(s, a, w) := r_h(s, a, w) + \mathbb{P}_h[V_{h+1}^\pi(., w)](s, a)$, where $Q_{H+1}^\pi = 0$. We denote the optimal policy as $\pi_h^*(s, w) := \sup_\pi V_h^\pi(s, w)$, and let $V_h^* := V_h^{\pi^*}$ and $Q_h^* := Q_h^{\pi^*}$ denote the optimal value functions. Lastly, we recall the Bellman equation of the optimal policy:

$$Q_h^*(s, a, w) = r_h(s, a, w) + \mathbb{P}_h[V_{h+1}^*(., w)](s, a), \quad V_h^*(s, w) = \max_{a \in \mathcal{A}} Q_h^*(s, a, w). \qquad (1)$$

**Interaction protocol of lifelong RL.** The agent interacts with a contextual MDP $M$ in episodes. For presentation simplicity, we assume that the reward functions $r$ are known, while the transition probabilities $\mathbb{P}$ are *unknown* and must be learned online; we will discuss how reward learning can be naturally incorporated in Section 4.3. At the beginning of episode $k$, the agent receives a task context $w^k \in \mathcal{W}$ and is set to an initial state $s_1^k$, both of which can be adversarially chosen. The agent can use past experiences to plan for the current task, if needed. Then the agent executes its policy $\pi^k$: at each time-step $h \in [H]$, it observes the state $s_h^k$, plays an action $a_h^k = \pi_h^k(s_h^k, w^k)$, observes a reward $r_h^k := r_h(s_h^k, a_h^k, w^k)$, and goes to the next state $s_{h+1}^k$ according to $\mathbb{P}_h(.|s_h^k, a_h^k)$. Let $K$ be the total number of episodes. The agent's goal is to achieve sublinear regret, where the regret is defined as

$$R_K := \sum_{k=1}^K V_1^*(s_1^k, w^k) - V_1^{\pi^k}(s_1^k, w^k). \qquad (2)$$

As the comparator policy above (namely $\pi^*$ that defines $V_1^*$) also knows the task context, achieving sublinear regret implies that the agent would attain near task-specific optimal performance on average.

**Linear Model Representation.** We focus on MDPs with linear transition kernels and reward functions (Jin et al., 2020; Yang & Wang, 2019) that are encapsulated in the following assumption.

**Assumption 1** (Linear MDPs). $M = (\mathcal{S}, \mathcal{A}, H, \mathbb{P}, r, \mathcal{W})$ *is a linear MDP with feature maps* $\boldsymbol{\phi} : \mathcal{S} \times \mathcal{A} \to \mathbb{R}^d$ *and* $\boldsymbol{\psi} : \mathcal{S} \times \mathcal{A} \times \mathcal{W} \to \mathbb{R}^{d'}$. *That is, for any* $h \in [H]$, *there exist a vector* $\boldsymbol{\eta}_h$ *and $d$ measures* $\boldsymbol{\mu}_h := [\mu_h^{(1)}, \ldots, \mu_h^{(d)}]^\top$ *over $\mathcal{S}$ such that* $\mathbb{P}_h(.|s, a) = \langle \boldsymbol{\mu}_h(.), \boldsymbol{\phi}(s, a) \rangle$ *and* $r_h(s, a, w) = \langle \boldsymbol{\eta}_h, \boldsymbol{\psi}(s, a, w) \rangle$, *for all* $(s, a, w) \in \mathcal{S} \times \mathcal{A} \times \mathcal{W}$. *Without loss of generality,* $\|\boldsymbol{\phi}(s, a)\|_2 \le 1$, $\|\boldsymbol{\psi}(s, a, w)\|_2 \le 1$, $\|\boldsymbol{\mu}_h(s)\|_2 \le \sqrt{d}$, *and* $\|\boldsymbol{\eta}_h\|_2 \le \sqrt{d'}$ *for all* $(s, a, w, h) \in \mathcal{S} \times \mathcal{A} \times \mathcal{W} \times [H]$.

In real-world problems, we can use the context to model the task specification of a problem. For example, if we want to design household robots to assist humans with a series of tasks like cooking, cleaning, washing dishes, lawn mowing, vacuuming, we can treat the the context as a natural language

---

[2]In general, a context-dependent dynamics would take the form $\mathbb{P}_h(s'|s, a, w)$.

instruction that the human user would give to the robot, and we can view the representations $\psi$ and $\phi$ as the embedding of a deep neural network model that has been pre-trained.

**Example 1** (Weighted Rewards). *An interesting and common special case is $\psi(s, a, w) = \phi(s, a) \otimes \rho(w)$, for some mapping $\rho : \mathcal{W} \to \mathbb{R}^m$. In this case, it holds that $d' = md$ and $r_h(s, a, w) = \langle \rho(w), \mathbf{r}_h(s, a) \rangle$, where $\mathbf{r}_h(s, a) = \mathbf{A}_h \phi(s, a) \in \mathbb{R}^m$, for some $\mathbf{A}_h \in \mathbb{R}^{m \times d}$, is the vector reward functions at time-step $h$. We can view $r_h(s, a, w)$ as a weighted reward with weights $\rho(w)$ that depend on task $w$. This setting is closely related to Multi-Objective RL studied for tabular case in Wu et al. (2021), which studies the case where $\rho(w) = w \in \mathbb{R}^m$ along with tabular $\mathcal{S}$ and $\mathcal{A}$.*

## 3 A WARM-UP ALGORITHM FOR LIFELONG RL

We first present a warm-up algorithm based on linear representation, termed Lifelong Least-Squares Value Iteration (Lifelong-LSVI), in Algorithm 1, which is a straightforward extension of the single-task LSVI-UCB algorithm proposed by Jin et al. (2020) to the lifelong learning setting. The motivation of this warm-up algorithm is to give intuitions on how the problem structure in Assumption 1 can be used to achieve small regret and discuss the computational difficulty in lifelong learning.

We will show that Lifelong-LSVI has a sublinear regret bound, which matches the minimax optimal rate in the special case studied by Wu et al. (2021) in terms of number of objectives, $m$ (see Example 1). However, we will also show that Lifelong-LSVI is *not* computationally efficient, in the sense that the number of planning calls it requires grows linearly with the number of episodes, which would mean the overall computational complexity grows quadratically. This high computation cost is because the agent never learns to internalize the task solving skills but requires going though all past experiences for planning every time a new task arrives. Importantly, we will discuss why it cannot be made computationally efficient in an easy manner without further assumptions on the representation. This drawback motivates our new completeness assumption and our main algorithm, UCBlvd, which is provably efficient in terms of both regret and number of planning calls, in Section 4.

We remark that Lifelong-LSVI is only a warm-up algorithm that guides the reader to understand the mechanisms used for addressing the problem, motivates the need for UCBlvd, and shows what regret bound is possible when computational complexity is not a concern (though being impractical).

### 3.1 ALGORITHMIC NOTATIONS

To begin, we introduce the template and the notations that will be used commonly in presenting the warm-up algorithm, Lifelong-LSVI, and later our main algorithm, UCBlvd. For each algorithm, first we will define an algorithm-specific action-value function $Q_h^k : \mathcal{S} \times \mathcal{A} \times \mathcal{W} \to \mathbb{R}$, which determines the agent's policy at time-step $h$ in episode $k$; then we present the full algorithm and its analysis using the quantities below, which are defined with respect to each algorithm's definition of $Q_h^k$.

Given $\{Q_h^k\}_{h \in [H]}$, we define state value functions and their backups as

$$V_h^k(s, w) := \min \left\{ \max_{a \in \mathcal{A}} Q_h^k(s, a, w), H \right\}, \quad \boldsymbol{\theta}_h^k(w) := \int_{\mathcal{S}} V_{h+1}^k(s', w) d\boldsymbol{\mu}_h(s'), \qquad (3)$$

Thanks to the linear MDP structure in Assumption 1, it holds that

$$\mathbb{P}_h \left[ V_{h+1}^k(., w) \right] (s, a) = \left\langle \boldsymbol{\theta}_h^k(w), \phi(s, a) \right\rangle. \qquad (4)$$

Let $\lambda > 0$ be a constant. We define the $\lambda$-regularized least squares estimator of $\boldsymbol{\theta}_h^k(w)$ as

$$\tilde{\boldsymbol{\theta}}_h^k(w) := \left( \mathbf{\Lambda}_h^k \right)^{-1} \sum_{\tau=1}^{k-1} \phi_h^\tau V_{h+1}^k(s_{h+1}^\tau, w), \text{ where } \mathbf{\Lambda}_h^k := \lambda \mathbf{I}_d + \sum_{\tau=1}^{k-1} \phi_h^\tau \phi_h^{\tau \top}, \qquad (5)$$

and $\tilde{\boldsymbol{\theta}}_h^k(w)$ is the solution to $\min_{\boldsymbol{\theta} \in \mathbb{R}^d} \sum_{\tau=1}^{k-1} (\langle \boldsymbol{\theta}, \phi(s_h^\tau, a_h^\tau) \rangle - V_{h+1}^k(s_{h+1}^\tau, w))^2 + \lambda \|\boldsymbol{\theta}\|_2^2$, $\phi_h^\tau := \phi(s_h^\tau, a_h^\tau)$, and $\mathbf{I}_d \in \mathbb{R}^{d \times d}$ is the identity matrix.

### 3.2 DETAILS OF LIFELONG-LSVI AND ITS THEORETICAL GUARANTEES

We define the upper confidence bound (UCB) style action-value function of Lifelong-LSVI as follows:

$$Q_h^k(s, a, w) := r_h(s, a, w) + \left\langle \tilde{\boldsymbol{\theta}}_h^k(w), \phi(s, a) \right\rangle + \beta \left\| \phi(s, a) \right\|_{(\mathbf{\Lambda}_h^k)^{-1}}, \qquad (6)$$

---

**Algorithm 1:** Lifelong-LSVI

---

1  **Set:** $Q_{H+1}^k(.,.,.) = 0, \ \forall k \in [K]$
2  **for** episodes $k = 1, \ldots, K$ **do**
3      Observe the initial state $s_1^k$ and the task context $w^k$.
4      **for** time-steps $h = H, \ldots, 1$ **do**
5         Compute $\tilde{\boldsymbol{\theta}}_h^k(w^k)$ as in (5) using $Q_{h+1}^k$ defined in (6).
6      **for** time-steps $h = 1, \ldots, H$ **do**
7         Compute $Q_h^k(s_h^k, a, w^k)$ for all $a \in \mathcal{A}$ as in (6).
8         Play $a_h^k = \arg\max_{a \in \mathcal{A}} Q_h^k(s_h^k, a, w^k)$ and observe $s_{h+1}^k$ and $r_h^k$.

---

where $Q_{H+1}^k = 0$ and $\tilde{\boldsymbol{\theta}}_h^k(w)$ and $\boldsymbol{\Lambda}_h^k$ are defined in (5). Here, $\beta$ is an exploration factor that will be appropriately chosen in Theorem 1. At episode $k$, given $w^k$, Lifelong-LSVI first performs planning backward in time based on past data to compute $\tilde{\boldsymbol{\theta}}_h^k(w^k)$ in (5) using $Q_{h+1}^k$ defined in (6) (Lines 4-5). Then, in execution, it uses $\tilde{\boldsymbol{\theta}}_h^k(w^k)$ to compute $Q_h^k(s_h^k, a, w^k)$ for the current state and all $a \in \mathcal{A}$ (Line 7) and executes the action with the highest value (Line 8).

We show that Lifelong-LSVI achieves sublinear regret for our lifelong RL setup. The complete proof is reported in Appx. A, which follows the ideas of LSVI-UCB (Jin et al., 2020).

**Theorem 1.** *Let $T = KH$. Under Assumption 1, there exists an absolute constant $c > 0$ such that for any fixed $\delta \in (0, 0.5)$, if we set $\lambda = 1$ and $\beta = cH \left( d + \sqrt{d'} \right) \sqrt{\log(dd'T/\delta)}$ in Algorithm 1, then with probability at least $1 - 2\delta$, it holds that $R_K \leq \tilde{\mathcal{O}} \left( \sqrt{(d^3 + dd')H^3 T} \right)$.*

Before introducing our main algorithm in Section 4, we make a few remarks on the regret and number of planning calls of Lifelong-LSVI. First, Theorem 1 implies that for the special case studied by Wu et al. (2021) (Example 1), the regret bound of Lifelong-LSVI becomes $\tilde{\mathcal{O}}(\sqrt{md^3 H^3 T})$. This rate is optimal in terms of its dependency on $m$, as shown in Wu et al. (2021). Furthermore, this rate matches the LSVI-UCB's regret dependencies on $d$ and $H$ for the single-task setting (Jin et al., 2020).

While Lifelong-LSVI has a decent regret guarantee, it requires computing $\tilde{\boldsymbol{\theta}}_h^k(w^k)$ for all $h \in [H]$, whenever a distinct new task $w^k$ arrives. Since the number of unique tasks may be as large as $K$, the total number of planning calls required in Lifelong-LSVI is $K$ in the worst case.

Unfortunately, the number of planning calls of Lifelong-LSVI cannot be easily improved, because under Assumption 1 alone, the optimal Q-function $Q_h^*(s, a, w)$ of the CMDP can be *nonlinear* in the representation $\boldsymbol{\psi}$. As a result, for any algorithm that represents its policy linearly based on both $\boldsymbol{\psi}$ and $\boldsymbol{\phi}$, in general it is necessary to recompute the coefficients for every new $w$ to be optimal. For Lifelong-LSVI specifically, this nonlinear dependency shows up in $\tilde{\boldsymbol{\theta}}_h^k(w)$ of $Q_h^k(s, a, w)$ in (6).

In the next section, we discuss how placing a completeness-style assumption, which ensures $Q_h^*(s, a, w)$ can be linearly parameterized by $\boldsymbol{\psi}$, would circumvent the issue of non-linear dependency of the action-value functions on $w$, and consequently would enable computation sharing to decrease the number of planning calls to $\mathcal{O}(dH \log(K))$.

## 4  UCB Lifelong Value Distillation (UCBlvd)

In this section, we present our main algorithm, **UCB L**ifelong **V**alue **D**istillation (UCBlvd), in Algorithm 2. Under new completeness-style assumption that we will introduce in Section 4.1, we show that UCBlvd shares the same regret bound as Lifelong-LSVI but significantly reduces the number of planning calls to be logarithmic in $K$. In contrast to Lifelong-LSVI which learns individual action-value function for each $w^k$, UCBlvd learns a single action-value function for all $w \in \mathcal{W}$ based on $\boldsymbol{\psi}(s, a, w)$ to enable computation sharing across tasks, which is made possible by the extra completeness-style assumption. In general, in order to directly extend Lifelong-LSVI to only use feature $\boldsymbol{\psi}(s, a, w) \in \mathbb{R}^{d'}$ with $d' \geq d$, we need a context-dependent dynamics structure, which would eventually increase the regret. UCBlvd maintains the same order of regret as Lifelong-LSVI

by separating the planning into a novel two-step process: *1)* independent planning with $\phi$ for a set of representative task contexts and *2)* distilling the planned results into a multi-task value function parameterized by $\psi$. In addition, UCBlvd runs a doubling schedule to decide whether replanning is necessary, which makes the total number of planning calls logarithmic in $K$.

## 4.1 ENABLING COMPUTATION SHARING

As lifelong RL with Assumption 1 alone would require replanning in every episode in general (see Section 3), here we introduce new structural assumptions on $\psi$ to enable computation sharing across tasks. First, we define the following class of functions

$$\mathcal{F} = \left\{ f : f(s, w) = \min\left\{ \max_{a \in \mathcal{A}} \left\{ \langle \boldsymbol{\nu}, \boldsymbol{\psi}(s, a, w) \rangle + \beta \|\boldsymbol{\phi}(s, a)\|_{\boldsymbol{\Lambda}^{-1}} \right\}^+, H \right\}, \boldsymbol{\nu} \in \mathbb{R}^{d'}, \boldsymbol{\Lambda} \in \mathbf{S}_{++}^d, \beta \geq 0 \right\},$$

where $\mathbf{S}_{++}^d$ denotes the set of symmetric positive definite matrices. We now state our main completeness-style assumption.

**Assumption 2** (Completeness). *For any $f \in \mathcal{F}$ and $h \in [H]$, there exists a vector $\boldsymbol{\xi}_h^f \in \mathbb{R}^{d'}$ with $\left\|\boldsymbol{\xi}_h^f\right\| \leq H\sqrt{d'}$ such that $\mathbb{P}_h\left[f(., w)\right](s, a) = \langle \boldsymbol{\xi}_h^f, \boldsymbol{\psi}(s, a, w) \rangle$.*

This assumption says that the backups of functions in $\mathcal{F}$ are captured by the feature $\psi$ with bounded parameters. The definition of $\mathcal{F}$ closely models the structure of action-value function used by Lifelong-LSVI in (6), except $\langle \tilde{\boldsymbol{\theta}}_h^k(w), \boldsymbol{\phi}(s, a) \rangle$ there is replaced by functions linear in $\boldsymbol{\psi}(s, a, w)$. We will see that the action-value function used by UCBlvd defined in the next section is contained in $\mathcal{F}$. In addition, by setting $\beta = 0$ in $\mathcal{F}$ and (1), we see $Q_h^*(s, a, w)$ is linearly realizable by $\psi$ under Assumption 2. We note that a similar notion of this assumption is mentioned in previous work for single-task settings under the name of "optimistic closure" (Wang et al., 2020).

Inspired by Example 1, we now introduce the next assumption on the structure of $\psi$.

**Assumption 3** (Mappings). *We assume $\boldsymbol{\psi}(s, a, w) = \boldsymbol{\phi}(s, a) \otimes \boldsymbol{\rho}(w)$, for some mapping $\boldsymbol{\rho} : \mathcal{W} \to \mathbb{R}^m$, i.e., $d' = md$. We assume that there is a known set $\{w^{(1)}, w^{(2)}, \ldots, w^{(n)}\}$ of $n \leq m$ task contexts such that $\boldsymbol{\rho}(w) \in \mathrm{Span}(\{\boldsymbol{\rho}(w^{(j)})\}_{j \in [n]})$ for all $w \in \mathcal{W}$. That is, for any $w \in \mathcal{W}$, there exist coefficients $\{c_j(w)\}_{j \in [n]}$ such that $\boldsymbol{\rho}(w) = \sum_{j \in [n]} c_j(w) \boldsymbol{\rho}(w^{(j)})$. We assume $\sum_{j \in [n]} |c_j(w)| \leq L$ for all $w \in \mathcal{W}$ and some $L < \infty$.*

Note that, for finite-dimensional representations, such set $\{\boldsymbol{\rho}(w^{(j)})\}_{j \in [n]}$ always exists. We assume that this set $\{w^{(1)}, w^{(2)}, \ldots, w^{(n)}\}$ is known to the algorithm

## 4.2 DETAILS OF UCBLVD

We define the UCB style action-value function of UCBlvd as follows:

$$Q_h^k(s, a, w) := \left\{ r_h(s, a, w) + \left\langle \hat{\boldsymbol{\xi}}_h^k, \boldsymbol{\psi}(s, a, w) \right\rangle + 2L\beta \|\boldsymbol{\phi}(s, a)\|_{(\boldsymbol{\Lambda}_h^k)^{-1}} \right\}^+. \tag{7}$$

The parameter $\hat{\boldsymbol{\xi}}_h^k$ is computed by solving the convex quadratically constrained quadratic program (QCQP) in (8), which is defined on a set of representative task contexts $\{w^{(1)}, w^{(2)}, \ldots, w^{(n)}\}$ in Assumption 3 and state-action pairs $\mathcal{D} := \left\{ (s, a) : \boldsymbol{\phi}(s, a) \text{ are } d \text{ linearly independent vectors.} \right\}$.

$$\hat{\boldsymbol{\xi}}_h^k, \{\hat{\boldsymbol{\theta}}_h^{k(j)}\}_{j \in [n]} = \operatorname*{arg\,min}_{\boldsymbol{\xi}, \{\boldsymbol{\theta}^{(j)}\}_{j \in [n]}} \sum_{j \in [n]} \sum_{(s,a) \in \mathcal{D}} \left( \langle \boldsymbol{\theta}^{(j)}, \boldsymbol{\phi}(s, a) \rangle - \langle \boldsymbol{\xi}, \boldsymbol{\psi}(s, a, w^{(j)}) \rangle \right)^2 \tag{8}$$

$$\text{s.t. } \left\|\boldsymbol{\theta}^{(j)} - \tilde{\boldsymbol{\theta}}_h^k(w^{(j)})\right\|_{\boldsymbol{\Lambda}_h^k} \leq \beta, \; \forall j \in [n] \quad \text{and} \quad \|\boldsymbol{\xi}\|_2 \leq H\sqrt{md},$$

where $\tilde{\boldsymbol{\theta}}_h^k(w)$ and $\boldsymbol{\Lambda}_h^k$ are defined in (5). In Appx. B.3, we will show that the action-value function in (7) is an optimistic estimate of the optimal action-value function.

UCBlvd also uses the linear dependency of $Q_h^k$ on $\psi$ to reduce calls of the planning step in (8). The agent triggers replanning only when it has gathered enough new information compared to the last update at episode $\tilde{k}$. This is measured by tracking the variations in Gram matrices $\{\boldsymbol{\Lambda}_h^k\}_{h \in [H]}$ (Line 4 for Algorithm 2). Finally, when executing the policy at episode $k$, the agent chooses the action according to $Q_h^{\tilde{k}}$ in Line 10.

---

**Algorithm 2:** UCBlvd (UCB Lifelong Value Distillation)

1  **Set:** $Q_{H+1}^k(.,.,.) = 0, \ \forall k \in [K], \ \tilde{k} = 1$
2  **for** episodes $k = 1, \ldots, K$ **do**
3     Observe the initial state $s_1^k$ and the task context $w^k$.
4     **if** $\exists h \in [H]$ *such that* $\log \det \mathbf{\Lambda}_h^k - \log \det \mathbf{\Lambda}_h^{\tilde{k}} > 1$ **then**
5         $\tilde{k} = k$
6         **for** time-steps $h = H, \ldots, 1$ **do**
7             Compute $\hat{\boldsymbol{\xi}}_h^{\tilde{k}}$ as in (8).
8     **for** time-steps $h = 1, \ldots, H$ **do**
9         Compute $Q_h^{\tilde{k}}(s_h^k, a, w^k)$ for all $a \in \mathcal{A}$ as in (7).
10         Play $a_h^k = \arg\max_{a \in \mathcal{A}} Q_h^{\tilde{k}}(s_h^k, a, w^k)$ and observe $s_{h+1}^k$ and $r_h^k$.

---

### 4.3  Theoretical analysis of UCBlvd

We present our main theoretical result which shows UCBlvd achieves sublinear regret in lifelong RL using sublinear number of planning calls, for any sequence of tasks. The proof is given in Appx. B.

**Theorem 2.** *Let $T = KH$. Under Assumptions 1, 2, and 3, the number of planning calls in Algorithm 2 is at most $dH \log(1 + \frac{K}{d\lambda})$, and there exists an absolute constant $c > 0$ such that for any fixed $\delta \in (0, 0.5)$, if we set $\lambda = 1$ and $\beta = cH(d + \sqrt{md})\sqrt{\log(mdT/\delta)}$ in Algorithm 2, then with probability at least $1 - 2\delta$, it holds that $R_K \leq \tilde{\mathcal{O}}\left(L\sqrt{(d^3 + md^2)H^3 T}\right)$.*

Theorem 2 shows that UCBlvd has the same regret bound as Lifelong-LSVI in Theorem 1, but reduces the number of planning calls from $K$ to $dH \log(1 + K/d\lambda)$. As we discussed before, this is made possible by the unique QCQP-based distillation step of UCBlvd in (8). If we were to simply perform least-squares regression to fit $\langle \boldsymbol{\psi}(s, a, w), \hat{\boldsymbol{\xi}}_h^k \rangle$ to $\{\langle \boldsymbol{\phi}(s, a), \tilde{\boldsymbol{\theta}}_h^k(w^{(j)})\rangle\}_{j \in [n]}$ for distillation, we cannot guarantee the required optimism, because $\langle \boldsymbol{\phi}(s, a), \tilde{\boldsymbol{\theta}}_h^k(w)\rangle$ computed based on finite samples can be an irregular function that cannot be modelled by $\boldsymbol{\psi}(s, a, w)$.

**Remark 1.** *If the rewards are unknown, we can adopt a slightly different completeness assumption with an extra bonus in terms of $\boldsymbol{\psi}$, and then combine tools from linear bandits (Abbasi-Yadkori et al., 2011) and our proof of Theorem 2. Because reward learning affects the radius of the confidence intervals for $\boldsymbol{\theta}_h^k(w)$, the number of planning calls and regret would increase by factors of $\mathcal{O}(m)$ and $\mathcal{O}(\sqrt{m})$ [3], respectively, compared to those in Theorem 2. See Appx. C for details.*

**Remark 2.** *It is possible to eliminate the assumption that $\boldsymbol{\psi}(s, a, w) = \boldsymbol{\phi}(s, a) \otimes \boldsymbol{\rho}(w)$. In this case, our analysis would instead require a set $\{w^{(1)}, w^{(2)}, \ldots, w^{(n)}\}$ of $n$ tasks such that $\boldsymbol{\psi}(s, a, w) \in \mathrm{Span}(\{\boldsymbol{\psi}(s, a, w^{(j)})\}_{j \in [n]})$ for all $(s, a, w) \in \mathcal{S} \times \mathcal{A} \times \mathcal{W}$. In Appx. D, we provide details of this relaxation, and show that this version still enjoys the same planning calls and regret as in Theorem 2.*

**Remark 3.** *We can eliminate Assumptions 1 and 3 and instead design a computation-sharing version of Lifelong-LSVI under a sightly different completeness assumption with a class $\mathcal{F}$, whose exploration bonus is $\beta \| \boldsymbol{\psi}(s, a, w) \|_{\tilde{\mathbf{\Lambda}}^{-1}}$. This assumption naturally includes settings with linear MDP in which dynamics also change with task context, i.e., for all $h \in [H]$, it holds that $\mathbb{P}_h(.|s, a, w) = \langle \boldsymbol{\mu}_h(.), \boldsymbol{\psi}(s, a, w) \rangle$ for $d'$ unknown measures $[\mu_h^{(1)}, \ldots, \mu_h^{(d')}]^\top$. Under this assumption, a slightly modified version of Lifelong-LSVI would use $Q_h^k(s, a, w) = \{r_h(s, a, w) + \langle \tilde{\boldsymbol{\nu}}_h^k, \boldsymbol{\psi}(s, a, w) \rangle + \beta \| \boldsymbol{\psi}(s, a, w) \|_{(\tilde{\mathbf{\Lambda}}_h^k)^{-1}}\}^+$, where $\tilde{\boldsymbol{\nu}}_h^k = (\tilde{\mathbf{\Lambda}}_h^k)^{-1} \sum_{\tau=1}^{k-1} \boldsymbol{\psi}_h^\tau \cdot \min\{\max_{a \in \mathcal{A}} Q_{h+1}^k(s_{h+1}^\tau, a, w^\tau), H\}$, $\tilde{\mathbf{\Lambda}}_h^k = \lambda \mathbf{I}_{d'} + \sum_{\tau=1}^{k-1} \boldsymbol{\psi}_h^\tau \boldsymbol{\psi}_h^{\tau \top}$, $\boldsymbol{\psi}_h^\tau = \boldsymbol{\psi}(s_h^\tau, a_h^\tau, w^\tau)$, and $\beta = \tilde{\mathcal{O}}(d')$. However, in Appx. E, we show how these new algorithm and assumption result in $\tilde{\mathcal{O}}(mdH)$ number of planning calls and a regret*

---

[3] While for both settings in this remark and Remark 3, the action-value functions contain exploration bonus in terms of $\boldsymbol{\psi}$, the regret here is better by a factor of $\sqrt{m}$ and this is because the multiplicative factor $\beta$ here saves a factor $\sqrt{m}$ compared to that in Remark 3.

*scaling with $\tilde{\mathcal{O}}(\sqrt{m^3 d^3})$ for settings with $\boldsymbol{\psi}(s, a, w) = \boldsymbol{\phi}(s, a) \otimes \boldsymbol{\rho}(w)$. These are worse than the number of planning calls and regret in Theorem 2 of UCBlvd by a factor of $\mathcal{O}(m)$.*

**Remark 4.** *A natural follow-up relaxation of Assumption 2 is when the equality holds up to an error of $\zeta$. In Appx. F, we show that this relaxation results in a regret $\tilde{\mathcal{O}}\left(\sqrt{mdT}\zeta + \sqrt{\lambda(d^3 + md^2)H^3T}\right)$ and the same number of planning calls as that in Theorem 2. When $\zeta$ is sufficiently small, i.e., $\zeta = \mathcal{O}(\sqrt{d^2 H^3 / mT})$, UCBlvd will still enjoy a regret of the same order as that in Theorem 2.*

### 4.4 PROOF SKETCH OF THEOREM 2

Because the proof of planning calls' upper bound follows standard arguments in low switching cost analysis of Abbasi-Yadkori et al. (2011), in this section, we focus on the proof sketch for the regret bound. We start by introducing the high probability event $\mathcal{E}_1$, which is the foundation of our analysis:

$$\mathcal{E}_1(w) := \left\{ \left\| \boldsymbol{\theta}_h^k(w) - \tilde{\boldsymbol{\theta}}_h^k(w) \right\|_{\boldsymbol{\Lambda}_h^k} \leq \beta, \forall (h, k) \in [H] \times [K] \right\}. \tag{9}$$

The following lemma highlights the importance of the carefully designed planning step in (8), which ensures good estimators for $\boldsymbol{\xi}_h^{V_{h+1}^*}$ without the need of bonus term $\left\| \boldsymbol{\psi}(s, a, w) \right\|_{\left(\tilde{\boldsymbol{\Lambda}}_h^k\right)^{-1}}$. This step saves a factor $\mathcal{O}(m)$ in planning calls and regret.

**Lemma 1.** *Let $\widetilde{\mathcal{W}} = \{w^\tau : \tau \in [K]\} \cup \{w^{(j)} : j \in [n]\}$. Under the setting of Theorem 2 and conditioned on events $\{\mathcal{E}_1(w)\}_{w \in \widetilde{\mathcal{W}}}$ defined in (9), for all $(s, a, w, h, k) \in \mathcal{S} \times \mathcal{A} \times \widetilde{\mathcal{W}} \times [H] \times [K]$, it holds that $\left| \langle \hat{\boldsymbol{\xi}}_h^k, \boldsymbol{\psi}(s, a, w) \rangle - \mathbb{P}_h[V_{h+1}^k(., w)](s, a) \right| \leq 2L\beta \left\| \boldsymbol{\phi}(s, a) \right\|_{\left(\boldsymbol{\Lambda}_h^k\right)^{-1}}.$*

As the final step in the regret analysis, we use Lemma 1 to prove the optimistic nature of UCBlvd, i.e., $Q_h^k(s, a, w^k) \geq Q_h^*(s, a, w^k)$ for all $(s, a, h, k) \in \mathcal{S} \times \mathcal{A} \times [H] \times [K]$. Then following the standard analysis of single-task LSVI-UCB we derive the regret bound in Theorem 2.

### 4.5 EXPERIMENTS

We implemented our main algorithm UCBlvd on synthetic environments and compared its performance with the warm-up algorithm Lifelong-LSVI, which is viewed as an idealized baseline ignoring the computational complexity. In all the experiments, the same setting, task sequences and feature mappings were used for both UCBlvd and Lifelong-LSVI. Figure 1a depicts per-episode rewards for the main setup considered throughout the paper, and Figure 1b shows those for the setup in Remark 2. The plots verify that Lifelong-LSVI and UCBlvd statistically perform almost the same while UCBlvd uses much smaller numbers of planning calls (1000 vs $\sim 20$). We remark that Lifelong-LSVI has an overall computation complexity of $\mathcal{O}(K^2)$, which makes it not practical for the lifelong learning setting, as its planning complexity increases linearly with the number of samples. The details on the parameters of simulations are deferred to Appx. H.

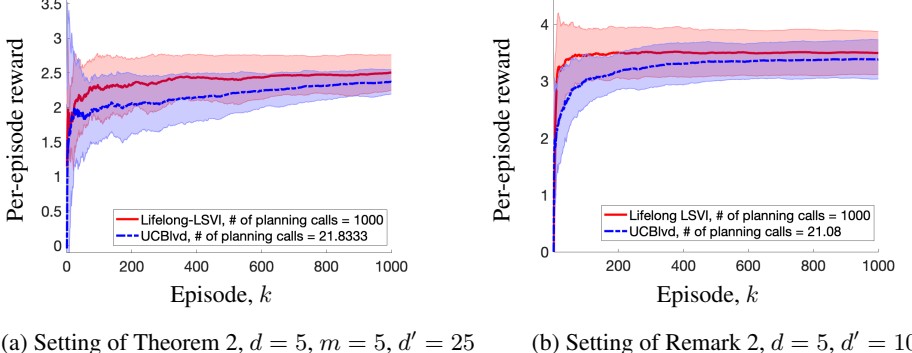

(a) Setting of Theorem 2, $d = 5$, $m = 5$, $d' = 25$      (b) Setting of Remark 2, $d = 5$, $d' = 10$

Figure 1: UCBlvd vs Lifelong-LSVI. The experimental results include 50 seeds.

## 5 RELATED WORK

We consider the regret minimization setup of lifelong RL under the contextual MDP framework, where the agent receives tasks specified by contexts in sequence and needs to achieve a sublinear regret for any task sequence. Below, we contrast our work with related work in the literature.

**Lifelong RL.** Generally lifelong RL studies how to learn to solve a streaming sequence of tasks using rewards. While it was originally motivated by the need of endless learning of robots (Thrun & Mitchell, 1995), historically many works on lifelong RL (Ammar et al., 2014; Brunskill & Li, 2014; Abel et al., 2018a;b; Lecarpentier et al., 2021) assume that the tasks are i.i.d. (similar to multi-task RL; see below). There are works for adversarial sequences, but most of them assume finite number of tasks (Brunskill & Li, 2015; Ammar et al., 2015; Zhan et al., 2017) or are purely empirical (Xie & Finn, 2021). The work by Isele et al. (2016) uses contexts to enable zero-shot learning like here, but it (as well as most works above) does not provide formal regret guarantees.[4] Brunskill & Li (2015) and Xie & Finn (2021) assume the task identity is latent, which requires additional exploration; in this sense, their problem is harder than the setup here where the task context is revealed. Extending the setup here to consider latent context is an important future direction.

**Contextual MDP and multi-objective RL.** Our setup is closely related to the exploration problem studied in the contextual MDP literature, though contextual MDP is originally not motivated from the lifelong learning perspective. A similar mathematical problem appears in the dynamic setup of multi-objective RL (Wu et al., 2021; Abels et al., 2019), which can be viewed as a special case of contextual MDP where the context linearly determines the reward function but not the dynamics. Most contextual MDP works allow adversarial contexts and initial states, but a majority of them focuses on the tabular setup (Abbasi-Yadkori & Neu, 2014; Hallak et al., 2015; Modi et al., 2018; Modi & Tewari, 2020; Levy & Mansour, 2022; Wu et al., 2021), whereas our setup allows continuous states. Kakade et al. (2020) and Du et al. (2019) allow continuous state and action spaces, but the former assumes a planning oracle with unclear computational complexity and the latter focuses on only LQG problems. While generally contextual MDP allows both the reward and the dynamics to vary with contexts, we focus on the effects of context-independent dynamics similar to Kakade et al. (2020); Wu et al. (2021). In particular, the recent work of Wu et al. (2021) is the closest to ours, but they study the sample complexity in the tabular setup with linearly parameterized rewards. In view of Example 1, their proposed algorithm has a regret bound $\tilde{\mathcal{O}}(\sqrt{\min\{m,|\mathcal{S}|\}H|\mathcal{S}||\mathcal{A}|K})$. However, they need linear number of planning calls. On the contrary, our algorithm, UCBlvd, allows continuous states, nonlinear context dependency, and has both sublinear regret and number of planning calls.

**Multi-task RL.** Another closely related line of work is multi-task RL. Compared to our setting, multi-task RL assumes that there are beforehand known finite tasks and/or they are i.i.d. samples from a fixed distribution. For example, in Yang et al. (2020); Hessel et al. (2019); Brunskill & Li (2013); Fifty et al. (2021); Zhang & Wang (2021); Sodhani et al. (2021), tasks are assumed to be chosen from a known finite set, and in Yang et al. (2020); Wilson et al. (2007); Brunskill & Li (2013); Sun et al. (2021), tasks are sampled from a fixed distribution. By contrast, our setting provides guarantees on regret and number of planning calls for adversarial task sequences.

## 6 DISCUSSION

In this paper, we frame lifelong RL as contextual MDPs and identify a new completeness-style assumption to enable provably efficient lifelong RL with linear representation. We propose UCBlvd, an algorithm that *simultaneously* satisfies the practical need of achieving *1)* sublinear regret and *2)* sublinear number of planning calls for *3)* any sequence of tasks and initial states. Specifically, for $K$ task episodes of horizon $H$, we prove that UCBlvd has a regret bound $\tilde{\mathcal{O}}(\sqrt{(d^3 + d'd)H^4K})$ based on $\tilde{\mathcal{O}}(dH\log(K))$ number of planning calls, where $d$ and $d'$ are the feature dimensions of the dynamics and rewards, respectively. We believe that our results would inspire new research directions in the literature of CMDP and multi-objective RL, as existing work to our knowledge does not cover the computation-sharing aspect of lifelong RL. That said, our work's limitations motivate further investigations in the following directions: *1)* extension to more general class of MDPs, potentially using general function approximation/representation tools, *2)* establishing an information-theoretic lower bound on the number of planning calls/computation complexity.

---

[4]Ammar et al. (2015) give regret bounds but only for linearized value difference; Brunskill & Li (2015) show regret bounds only for finite number of tasks.

## ACKNOWLEDGEMENT

This work is partially supported by DARPA grant HR00112190130 and NSF grant 2221871. Sanae Amani is partially supported by Amazon science hub fellowship.

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

## A   PROOFS OF SECTION 3

To prove Theorem 1, we will use the high probability event $\mathcal{E}_2$ defined in Lemma 3 to prove the UCB nature of Lifelong-LSVI in Lemma 4, which is the key to controlling the regret. We first state the following lemma that will be used in the proof of Lemma 3.

**Lemma 2.** *Under the setting of Theorem 1, let $c_\beta$ be the constant in the definition of $\beta$. Then, for a fixed $w$, there is an absolute constant $c_0$ independent of $c_\beta$, such that for all $(h, k) \in [H] \times [K]$, with probability at least $1 - \delta$ it holds that*

$$\left\| \sum_{\tau=1}^{k-1} \phi_h^\tau \cdot \left( V_{h+1}^k(s_{h+1}^\tau, w) - \mathbb{P}_h[V_{h+1}^k(., w)](s_h^\tau, a_h^\tau) \right) \right\|_{\left(\mathbf{\Lambda}_h^k\right)^{-1}} \leq c_0 H \left( d + \sqrt{d'} \right) \sqrt{\log((c_\beta + 1)dd'T/\delta)},$$

*where $c_0$ and $c_\beta$ are two independent absolute constants.*

*Proof.* We note that $\|\boldsymbol{\eta}_h\|_2 \leq \sqrt{d'}$ (Assumption 1), $\left\|\theta_h^k(w)\right\|_2 \leq H\sqrt{d}$ (Lemma 18), and $\left\|\left(\mathbf{\Lambda}_h^k\right)^{-1}\right\| \leq \frac{1}{\lambda}$. Thus, Lemmas 19 and 21 together imply that for all $(h, k) \in [H] \times [K]$, with probability at least $1 - \delta$ it holds that

$$\left\| \sum_{\tau=1}^{k-1} \phi_h^\tau \left( V_{h+1}^k(s_{h+1}^\tau, w) - \mathbb{P}_h[V_{h+1}^k(., w)](s_h^\tau, a_h^\tau) \right) \right\|_{\left(\mathbf{\Lambda}_h^k\right)^{-1}}^2$$

$$\leq 4H^2 \left( \frac{d}{2} \log\left( \frac{k+\lambda}{\lambda} \right) + d' \log(1 + 4d'/\epsilon) + d \log(1 + 4Hd/\epsilon) + d^2 \log\left( \frac{1 + 8B^2\sqrt{d}}{\lambda\epsilon^2} \right) + \log\left( \frac{1}{\delta} \right) \right) + \frac{8k^2\epsilon^2}{\lambda}.$$

If we let $\epsilon = \frac{dH}{k}$ and $\beta = c_\beta(d + \sqrt{d'})H\sqrt{\log(dT/\delta)}$, then, there exists an absolute constant $C > 0$ that is independent of $c_\beta$ such that

$$\left\| \sum_{\tau=1}^{k-1} \phi_h^\tau \left( V_{h+1}^k(s_{h+1}^\tau, w) - \mathbb{P}_h[V_{h+1}^k(., w)](s_h^\tau, a_h^\tau) \right) \right\|_{\left(\mathbf{\Lambda}_h^k\right)^{-1}}^2 \leq C(d' + d^2)H^2 \log\left( (c_\beta + 1)dd'T/\delta \right).$$

$\square$

**Lemma 3.** *Let the setting of Theorem 1 holds. The event*

$$\mathcal{E}_2(w) := \left\{ \left\| \boldsymbol{\theta}_h^k(w) - \tilde{\boldsymbol{\theta}}_h^k(w) \right\|_{\mathbf{\Lambda}_h^k} \leq \beta, \forall (h, k) \in [H] \times [K] \right\}. \tag{10}$$

*holds with probability at least $1 - \delta$ for a fixed $w$.*

*Proof.*

$$\boldsymbol{\theta}_h^k(w) - \tilde{\boldsymbol{\theta}}_h^k(w) = \boldsymbol{\theta}_h^k(w) - \left(\mathbf{\Lambda}_h^k\right)^{-1} \sum_{\tau=1}^{k-1} \phi_h^\tau V_{h+1}^k(s_{h+1}^\tau, w)$$

$$= \left(\mathbf{\Lambda}_h^k\right)^{-1} \left( \mathbf{\Lambda}_h^k \boldsymbol{\theta}_h^k(w) - \sum_{\tau=1}^{k-1} \phi_h^\tau V_{h+1}^k(s_{h+1}^\tau, w) \right)$$

$$= \underbrace{\lambda \left(\mathbf{\Lambda}_h^k\right)^{-1} \boldsymbol{\theta}_h^k(w)}_{\mathbf{q}_1} - \underbrace{\left(\mathbf{\Lambda}_h^k\right)^{-1} \left( \sum_{\tau=1}^{k-1} \phi_h^\tau \left( V_{h+1}^k(s_{h+1}^\tau, w) - \mathbb{P}_h[V_{h+1}^k(., w)](s_h^\tau, a_h^\tau) \right) \right)}_{\mathbf{q}_2}.$$

Thus, in order to upper bound $\left\|\boldsymbol{\theta}_h^k(w) - \tilde{\boldsymbol{\theta}}_h^k(w)\right\|_{\boldsymbol{\Lambda}_h^k}$, we bound $\|\mathbf{q}_1\|_{\boldsymbol{\Lambda}_h^k}$ and $\|\mathbf{q}_2\|_{\boldsymbol{\Lambda}_h^k}$ separately.

From Lemma 18, we have

$$\|\mathbf{q}_1\|_{\boldsymbol{\Lambda}_h^k} = \lambda\left\|\boldsymbol{\theta}_h^k(w)\right\|_{\left(\boldsymbol{\Lambda}_h^k\right)^{-1}} \leq \sqrt{\lambda}\left\|\boldsymbol{\theta}_h^k(w)\right\|_2 \leq H\sqrt{\lambda d}. \tag{11}$$

Thanks to Lemma 2, for all $(w, h, k)$, with probability at least $1 - \delta$, it holds that

$$\|\mathbf{q}_2\|_{\boldsymbol{\Lambda}_h^k} \leq \left\|\sum_{\tau=1}^{k-1} \boldsymbol{\phi}_h^\tau \left(V_{h+1}^k(s_{h+1}^\tau, w) - \mathbb{P}_h[V_{h+1}^k(., w)](s_h^\tau, a_h^\tau)\right)\right\|_{\left(\boldsymbol{\Lambda}_h^k\right)^{-1}} \leq c_0 H\left(d + \sqrt{d'}\right)\sqrt{\log((c_\beta + 1)dd'T/\delta)}, \tag{12}$$

where $c_0$ and $c_\beta$ are two independent absolute constants.

Combining (11) and (12), for all $(w, h, k)$, with probability at least $1 - \delta$, it holds that

$$\left\|\boldsymbol{\theta}_h^k(w) - \tilde{\boldsymbol{\theta}}_h^k(w)\right\|_{\boldsymbol{\Lambda}_h^k} \leq cH\left(d + \sqrt{d'}\right)\sqrt{\lambda \log(dd'T/\delta)}$$

for some absolute constant $c > 0$.

$\square$

**Lemma 4.** *Let $\widetilde{\mathcal{W}} = \{w^1, w^2, \ldots, w^K\}$. Under the setting of Theorem 1 and conditioned on events $\{\mathcal{E}_2(w)\}_{w \in \widetilde{\mathcal{W}}}$ defined in (10), and with $Q_h^k$ computed as in (6), it holds that $Q_h^k(s, a, w) \geq Q_h^*(s, a, w)$ for all $(s, a, w, h, k) \in \mathcal{S} \times \mathcal{A} \times \widetilde{\mathcal{W}} \times [H] \times [K]$.*

*Proof.* We first note that conditioned on events $\{\mathcal{E}_2(w)\}_{w \in \widetilde{\mathcal{W}}}$, for all $(s, a, w, h, k) \in \mathcal{S} \times \mathcal{A} \times \widetilde{\mathcal{W}} \times [H] \times [K]$, it holds that

$$\left|r_h(s, a, w) + \left\langle \tilde{\boldsymbol{\theta}}_h^k(w), \boldsymbol{\phi}(s, a) \right\rangle - Q_h^\pi(s, a, w) - \mathbb{P}_h\left[V_{h+1}^k(., w) - V_{h+1}^\pi(., w)\right](s, a)\right|$$

$$= \left|r_h(s, a, w) + \left\langle \tilde{\boldsymbol{\theta}}_h^k(w), \boldsymbol{\phi}(s, a) \right\rangle - r_h(s, a, w) - \mathbb{P}_h\left[V_{h+1}^k(., w)\right](s, a)\right|$$

$$= \left|\left\langle \tilde{\boldsymbol{\theta}}_h^k(w), \boldsymbol{\phi}(s, a) \right\rangle - \mathbb{P}_h\left[V_{h+1}^k(., w)\right](s, a)\right|$$

$$= \left|\left\langle \tilde{\boldsymbol{\theta}}_h^k(w) - \boldsymbol{\theta}_h^k(w), \boldsymbol{\phi}(s, a) \right\rangle\right|$$

$$\leq \left\|\tilde{\boldsymbol{\theta}}_h^k(w) - \boldsymbol{\theta}_h^k(w)\right\|_{\boldsymbol{\Lambda}_h^k}\left\|\boldsymbol{\phi}(s, a)\right\|_{\left(\boldsymbol{\Lambda}_h^k\right)^{-1}}$$

$$\leq \beta\left\|\boldsymbol{\phi}(s, a)\right\|_{\left(\boldsymbol{\Lambda}_h^k\right)^{-1}}, \tag{Lemma 3}$$

for any policy $\pi$.

Now, we prove the lemma by induction. The statement holds for $H$ because $Q_{H+1}^k(., ., .) = Q_{H+1}^*(., ., .) = 0$ and thus conditioned on events $\{\mathcal{E}_2(w)\}_{w \in \widetilde{\mathcal{W}}}$, defined in (10), for all $(s, a, w, k) \in \mathcal{S} \times \mathcal{A} \times \widetilde{\mathcal{W}} \times [K]$, we have

$$\left|r_H(s, a, w) + \left\langle \boldsymbol{\theta}_H^k(w), \boldsymbol{\psi}(s, a) \right\rangle - Q_H^*(s, a, w)\right| \leq \beta\left\|\boldsymbol{\phi}(s, a)\right\|_{\left(\boldsymbol{\Lambda}_H^k\right)^{-1}}.$$

Therefore, conditioned on events $\{\mathcal{E}_2(w)\}_{w \in \widetilde{\mathcal{W}}}$, for all $(s, a, w, k) \in \mathcal{S} \times \mathcal{A} \times \widetilde{\mathcal{W}} \times [K]$, we have

$$Q_H^*(s, a, w) \leq r_H(s, a, w) + \left\langle \boldsymbol{\theta}_H^k(w), \boldsymbol{\phi}(s, a) \right\rangle + \beta \|\boldsymbol{\phi}(s, a)\|_{(\boldsymbol{\Lambda}_H^k)^{-1}} = Q_H^k(s, a, w).$$

Now, suppose the statement holds at time-step $h+1$ and consider time-step $h$. Conditioned on events $\{\mathcal{E}_2(w)\}_{w \in \widetilde{\mathcal{W}}}$, for all $(s, a, w, h, k) \in \mathcal{S} \times \mathcal{A} \times \widetilde{\mathcal{W}} \times [H] \times [K]$, we have

$$0 \leq r_h(s, a, w) + \left\langle \boldsymbol{\theta}_h^k(w), \boldsymbol{\phi}(s, a) \right\rangle - Q_h^*(s, a, w) - \mathbb{P}_h \left[ V_{h+1}^k(., w) - V_{h+1}^*(., w) \right](s, a) + \beta \|\boldsymbol{\phi}(s, a)\|_{(\boldsymbol{\Lambda}_h^k)^{-1}}$$

$$\leq r_h(s, a, w) + \left\langle \boldsymbol{\theta}_h^k(w), \boldsymbol{\phi}(s, a) \right\rangle - Q_h^*(s, a, w) + \beta \|\boldsymbol{\phi}(s, a)\|_{(\boldsymbol{\Lambda}_h^k)^{-1}}.$$

(Induction assumption)

Therefore, conditioned on events $\{\mathcal{E}_2(w)\}_{w \in \widetilde{\mathcal{W}}}$, for all $(s, a, w, h, k) \in \mathcal{S} \times \mathcal{A} \times \widetilde{\mathcal{W}} \times [H] \times [K]$, we have

$$Q_h^*(s, a, w) \leq r_h(s, a, w) + \left\langle \boldsymbol{\theta}_h^k(w), \boldsymbol{\phi}(s, a) \right\rangle + \beta \|\boldsymbol{\phi}(s, a)\|_{(\boldsymbol{\Lambda}_h^k)^{-1}} = Q_h^k(s, a, w).$$

This completes the proof.

$\square$

## A.1 PROOF OF THEOREM 1

Let $\delta_h^k = V_h^k(s_h^k, w^k) - V_h^{\pi^k}(s_h^k, w^k)$ and $\xi_{h+1}^k = \mathbb{E}\left[\delta_{h+1}^k | s_h^k, a_h^k\right] - \delta_{h+1}^k$. Conditioned on events $\{\mathcal{E}_2(w)\}_{w \in \widetilde{\mathcal{W}}}$, for all $(s, a, w, h, k) \in \mathcal{S} \times \mathcal{A} \times \widetilde{\mathcal{W}} \times [H] \times [K]$, we have

$$Q_h^k(s, a, w) - Q_h^{\pi^k}(s, a, w) = r_h(s, a, w) + \left\langle \boldsymbol{\theta}_h^k(w), \boldsymbol{\phi}(s, a) \right\rangle - Q_h^{\pi^k}(s, a, w) + \beta \|\boldsymbol{\phi}(s, a)\|_{(\boldsymbol{\Lambda}_h^k)^{-1}}$$

$$\leq \mathbb{P}_h \left[ V_{h+1}^k(., w) - V_{h+1}^{\pi^k}(., w) \right](s, a) + 2\beta \|\boldsymbol{\phi}(s, a)\|_{(\boldsymbol{\Lambda}_h^k)^{-1}}. \quad (13)$$

Note that $\delta_h^k \leq Q_h^k(s_h^k, a_h^k, w^k) - Q_h^{\pi^k}(s_h^k, a_h^k, w^k)$. Thus, combining (13), Lemma 3, and a union bound over $\widetilde{\mathcal{W}}$, we conclude that for all $(h, k) \in [H] \times [K]$, with probability at least $1 - \delta$, it holds that

$$\delta_h^k \leq \xi_{h+1}^k + \delta_{h+1}^k + 2\beta \|\boldsymbol{\phi}(s_h^k, a_h^k)\|_{(\boldsymbol{\Lambda}_h^k)^{-1}}.$$

Now, we complete the regret analysis

$$R_K = \sum_{k=1}^K V_1^*(s_1^k, w^k) - V_1^{\pi^k}(s_1^k, w^k)$$

$$\leq \sum_{k=1}^K V_1^k(s_1^k, w^k) - V_1^{\pi^k}(s_1^k, w^k) \qquad \text{(Lemma 4)}$$

$$= \sum_{k=1}^K \delta_1^k$$

$$\leq \sum_{k=1}^K \sum_{h=1}^H \xi_h^k + 2\beta \sum_{k=1}^K \sum_{h=1}^H \|\boldsymbol{\phi}(s_h^k, a_h^k)\|_{(\boldsymbol{\Lambda}_h^k)^{-1}}$$

$$\leq 2H\sqrt{T \log(dT/\delta)} + 2H\beta\sqrt{2dK \log(1 + K/\lambda)}$$

$$\leq \tilde{\mathcal{O}}\left(\sqrt{\lambda(d^3 + dd')H^3 T}\right).$$

The third inequality is true because of the following: we observe that $\{\xi_h^k\}$ is a martingale difference sequence satisfying $|\xi_h^k| \leq 2H$. Thus, thanks to Azuma-Hoeffding inequality, we have

$$\mathbb{P}\left(\sum_{k=1}^K \sum_{h=1}^H \xi_h^k \leq 2H\sqrt{T \log(dT/\delta)}\right) \geq 1 - \delta. \quad (14)$$

In order to bound $\sum_{k=1}^{K} \sum_{h=1}^{H} \left\| \phi_h^k \right\|_{(\Lambda_h^k)^{-1}}$, note that for any $h \in [H]$, we have

$$\sum_{k=1}^{K} \left\| \phi_h^k \right\|_{(\Lambda_h^k)^{-1}} \leq \sqrt{K \sum_{k=1}^{K} \left\| \phi_h^k \right\|_{(\Lambda_h^k)^{-1}}^2} \qquad \text{(Cauchy-Schwartz inequality)}$$

$$\leq \sqrt{2K \log \left( \frac{\det \left( \Lambda_h^K \right)}{\det \left( \Lambda_h^1 \right)} \right)} \tag{15}$$

$$\leq \sqrt{2dK \log \left( 1 + \frac{K}{d\lambda} \right)}. \tag{16}$$

In inequality (15), we used the standard argument in regret analysis of linear bandits (Abbasi-Yadkori et al., 2011, Lemma 11) as follows:

$$\sum_{t=1}^{n} \min \left( \|\mathbf{y}_t\|_{\mathbf{V}_t^{-1}}^2, 1 \right) \leq 2 \log \frac{\det \mathbf{V}_{n+1}}{\det \mathbf{V}_1} \quad \text{where} \quad \mathbf{V}_n = \mathbf{V}_1 + \sum_{t=1}^{n-1} \mathbf{y}_t \mathbf{y}_t^\top. \tag{17}$$

In inequality (16), we used Assumption 1 and the fact that $\det(\mathbf{A}) = \prod_{i=1}^{d} \lambda_i(\mathbf{A}) \leq (\text{trace}(\mathbf{A})/d)^d$.

# B  PROOFS OF SECTION 4

We start by introducing the high probability event $\mathcal{E}_1$, which is the foundation of our analysis in the following lemma.

**Lemma 5.** *Follow the setting of Theorem 2. The event*

$$\mathcal{E}_1(w) := \left\{ \left\| \boldsymbol{\theta}_h^k(w) - \tilde{\boldsymbol{\theta}}_h^k(w) \right\|_{\Lambda_h^k} \leq \beta, \forall (h, k) \in [H] \times [K] \right\}. \tag{18}$$

*holds with probability at least $1 - \delta$ for a fixed $w$.*

Proof of Lemma 5 is given in Appx. B.1.

## B.1  PROOF OF LEMMA 5

First, we state the following lemma that will be used in the proof of Lemma 5.

**Lemma 6.** *Under the setting of Lemma 5, let $c_\beta$ be a constant in the definition of $\beta$. Then, for a fixed $w$, there is an absolute constant $c_0$ independent of $c_\beta$, such that for all $(h, k) \in [H] \times [K]$, with probability at least $1 - \delta$ it holds that*

$$\left\| \sum_{\tau=1}^{k-1} \phi_h^\tau \cdot \left( V_{h+1}^k(s_{h+1}^\tau, w) - \mathbb{P}_h[V_{h+1}^k(., w)](s_h^\tau, a_h^\tau) \right) \right\|_{(\Lambda_h^k)^{-1}} \leq c_0 H \left( d + \sqrt{md} \right) \sqrt{\log((c_\beta + 1)mdT/\delta)},$$

*where $c_0$ and $c_\beta$ are two independent absolute constants.*

*Proof.* We note that $\left\| \boldsymbol{\eta}_h + \hat{\boldsymbol{\xi}}_h^k \right\|_2 \leq (1 + H)\sqrt{md}$ and $\left\| \left( \Lambda_h^k \right)^{-1} \right\| \leq \frac{1}{\lambda}$. Thus, Lemmas 19 and 22 together imply that for all $(h, k) \in [H] \times [K]$, with probability at least $1 - \delta$ it holds that

$$\left\| \sum_{\tau=1}^{k-1} \phi_h^\tau \left( V_{h+1}^k(s_{h+1}^\tau, w) - \mathbb{P}_h[V_{h+1}^k(., w)](s_h^\tau, a_h^\tau) \right) \right\|_{(\Lambda_h^k)^{-1}}^2$$

$$\leq 4H^2 \left( \frac{d}{2} \log \left( \frac{k+\lambda}{\lambda} \right) + md \log(1 + 8H\sqrt{md}/\epsilon) + d^2 \log \left( \frac{1 + 32L^2\beta^2\sqrt{d}}{\lambda\epsilon^2} \right) + \log \left( \frac{1}{\delta} \right) \right) + \frac{8k^2\epsilon^2}{\lambda}.$$

If we let $\epsilon = \frac{dH}{k}$ and $\beta = c_\beta(d + \sqrt{md})H\sqrt{\log(dT/\delta)}$, then, there exists an absolute constant $C > 0$ that is independent of $c_\beta$ such that

$$\left\|\sum_{\tau=1}^{k-1} \phi_h^\tau \left(V_{h+1}^k(s_{h+1}^\tau, w) - \mathbb{P}_h[V_{h+1}^k(., w)](s_h^\tau, a_h^\tau)\right)\right\|_{\left(\mathbf{\Lambda}_h^k\right)^{-1}}^2 \leq C(md + d^2)H^2 \log\left((c_\beta + 1)mdT/\delta\right).$$

$\square$

Now, we begin the formal proof of Lemma 5:

$$\begin{aligned}
\boldsymbol{\theta}_h^k(w) - \tilde{\boldsymbol{\theta}}_h^k(w) &= \boldsymbol{\theta}_h^k(w) - \left(\mathbf{\Lambda}_h^k\right)^{-1} \sum_{\tau=1}^{k-1} \phi_h^\tau V_{h+1}^k(s_{h+1}^\tau, w) \\
&= \left(\mathbf{\Lambda}_h^k\right)^{-1} \left(\mathbf{\Lambda}_h^k \boldsymbol{\theta}_h^k(w) - \sum_{\tau=1}^{k-1} \phi_h^\tau V_{h+1}^k(s_{h+1}^\tau, w)\right) \\
&= \underbrace{\lambda \left(\mathbf{\Lambda}_h^k\right)^{-1} \boldsymbol{\theta}_h^k(w)}_{\mathbf{q}_1} - \underbrace{\left(\mathbf{\Lambda}_h^k\right)^{-1} \left(\sum_{\tau=1}^{k-1} \phi_h^\tau \left(V_{h+1}^k(s_{h+1}^\tau, w) - \mathbb{P}_h[V_{h+1}^k(., w)](s_h^\tau, a_h^\tau)\right)\right)}_{\mathbf{q}_2}.
\end{aligned}$$

Thus, in order to upper bound $\left\|\boldsymbol{\theta}_h^k(w) - \tilde{\boldsymbol{\theta}}_h^k(w)\right\|_{\mathbf{\Lambda}_h^k}$, we bound $\|\mathbf{q}_1\|_{\mathbf{\Lambda}_h^k}$ and $\|\mathbf{q}_2\|_{\mathbf{\Lambda}_h^k}$ separately.

From Lemma 18, we have

$$\|\mathbf{q}_1\|_{\mathbf{\Lambda}_h^k} = \lambda \left\|\boldsymbol{\theta}_h^k(w)\right\|_{\left(\mathbf{\Lambda}_h^k\right)^{-1}} \leq \sqrt{\lambda} \left\|\boldsymbol{\theta}_h^k(w)\right\|_2 \leq H\sqrt{\lambda d}. \tag{19}$$

Thanks to Lemma 6, for all $(w, h, k)$, with probability at least $1 - \delta$, it holds that

$$\begin{aligned}
\|\mathbf{q}_2\|_{\mathbf{\Lambda}_h^k} &\leq \left\|\sum_{\tau=1}^{k-1} \phi_h^\tau \left(V_{h+1}^k(s_{h+1}^\tau, w) - \mathbb{P}_h[V_{h+1}^k(., w)](s_h^\tau, a_h^\tau)\right)\right\|_{\left(\mathbf{\Lambda}_h^k\right)^{-1}} \\
&\leq c_0 H \left(d + \sqrt{md}\right) \sqrt{\log((c_\beta + 1)mdT/\delta)}, \tag{20}
\end{aligned}$$

where $c_0$ and $c_\beta$ are two independent absolute constants.

Combining (19) and (20), for all $(h, k) \in [H] \times [K]$, with probability at least $1 - \delta$, it holds that

$$\left\|\boldsymbol{\theta}_h^k(w) - \tilde{\boldsymbol{\theta}}_h^k(w)\right\|_{\mathbf{\Lambda}_h^k} \leq cH \left(d + \sqrt{md}\right) \sqrt{\lambda \log(mdT/\delta)}$$

for some absolute constant $c > 0$.

## B.2   PROOF OF LEMMA 1

Thanks to Assumption 2 and conditioned on events $\{\mathcal{E}_1(w)\}_{w \in \widetilde{\mathcal{W}}}$, one set of solution for (8) is $\left\{\boldsymbol{\theta}_h^k\left(w^{(j)}\right)\right\}_{j \in [n]}$ and $\boldsymbol{\xi}_h^{V_{h+1}^k}$ with corresponding zero optimal objective value. Therefore, it holds that

$$\left\langle \hat{\boldsymbol{\theta}}_h^{k(j)}, \phi(s, a) \right\rangle = \left\langle \hat{\boldsymbol{\xi}}_h^k, \psi\left(s, a, w^{(j)}\right) \right\rangle, \quad \forall(j, (s, a)) \in [n] \times \mathcal{D}. \tag{21}$$

Let $\left(s^{(i)}, a^{(i)}\right)$ be the $i$-th element of $\mathcal{D}$ and $\{c_i'(s,a)\}_{i \in [d]}$ be the coefficients such that

$$\phi(s,a) = \sum_{i \in [d]} c_i'(s,a) \phi\left(s^{(i)}, a^{(i)}\right).$$

For any triple $(s, a, j) \in \mathcal{S} \times \mathcal{A} \times [n]$, we have

$$
\begin{aligned}
\left\langle \hat{\boldsymbol{\xi}}_h^k, \boldsymbol{\psi}\left(s, a, w^{(j)}\right) \right\rangle &= \left\langle \hat{\boldsymbol{\xi}}_h^k, \phi(s,a) \otimes \boldsymbol{\rho}\left(w^{(j)}\right) \right\rangle \\
&= \left\langle \hat{\boldsymbol{\xi}}_h^k, \sum_{i \in [d]} c_i'(s,a) \phi\left(s^{(i)}, a^{(i)}\right) \otimes \boldsymbol{\rho}\left(w^{(j)}\right) \right\rangle \\
&= \sum_{i \in [d]} c_i'(s,a) \left\langle \hat{\boldsymbol{\xi}}_h^k, \boldsymbol{\psi}\left(s^{(i)}, a^{(i)}, w^{(j)}\right) \right\rangle && \text{(Assumption 3)} \\
&= \sum_{i \in [d]} c_i'(s,a) \left\langle \hat{\boldsymbol{\theta}}_h^{k(j)}, \phi\left(s^{(i)}, a^{(i)}\right) \right\rangle && \text{(Eqn. (21))} \\
&= \left\langle \hat{\boldsymbol{\theta}}_h^{k(j)}, \phi(s,a) \right\rangle. && (22)
\end{aligned}
$$

For any $(s, a, w) \in \mathcal{S} \times \mathcal{A} \times \mathcal{W}$, it holds that

$$
\begin{aligned}
\mathbb{P}_h\left[V_{h+1}^k(., w)\right](s,a) &= \left\langle \boldsymbol{\theta}_h^k(w), \phi(s,a) \right\rangle && \text{(Eqn. (4))} \\
&= \left\langle \boldsymbol{\xi}_h^{V_{h+1}^k}, \boldsymbol{\psi}(s,a,w) \right\rangle && \text{(Assumption 2)} \\
&= \sum_{j \in [n]} c_j(w) \left\langle \boldsymbol{\xi}_h^{V_{h+1}^k}, \boldsymbol{\psi}\left(s, a, w^{(j)}\right) \right\rangle && \text{(Assumption 3)} \\
&= \sum_{j \in [n]} c_j(w) \mathbb{P}_h\left[V_{h+1}^k\left(., w^{(j)}\right)\right](s,a) && \text{(Assumption 2)} \\
&= \sum_{j \in [n]} c_j(w) \left\langle \boldsymbol{\theta}_h^k\left(w^{(j)}\right), \phi(s,a) \right\rangle. && (23)
\end{aligned}
$$

Finally, conditioned on events $\{\mathcal{E}_1(w)\}_{w \in \widetilde{\mathcal{W}}}$, for all $(s, a, w, h, k) \in \mathcal{S} \times \mathcal{A} \times \widetilde{\mathcal{W}} \times [H] \times [K]$, it holds that

$$
\left| \left\langle \hat{\boldsymbol{\xi}}_h^k, \boldsymbol{\psi}(s, a, w) \right\rangle - \mathbb{P}_h \left[ V_{h+1}^k(., w) \right](s, a) \right|
$$

$$
= \left| \left\langle \hat{\boldsymbol{\xi}}_h^k, \boldsymbol{\psi}(s, a, w) \right\rangle - \left\langle \boldsymbol{\theta}_h^k(w), \boldsymbol{\phi}(s, a) \right\rangle \right|
$$

$$
= \left| \sum_{j \in [n]} c_j(w) \left( \left\langle \hat{\boldsymbol{\xi}}_h^k, \boldsymbol{\psi}\left(s, a, w^{(j)}\right) \right\rangle - \left\langle \boldsymbol{\theta}_h^k\left(w^{(j)}\right), \boldsymbol{\phi}(s, a) \right\rangle \right) \right|
$$

(Assumption 3 and Eqn. (23))

$$
\leq \left| \sum_{j \in [n]} c_j(w) \left( \left\langle \hat{\boldsymbol{\xi}}_h^k, \boldsymbol{\psi}\left(s, a, w^{(j)}\right) \right\rangle - \left\langle \hat{\boldsymbol{\theta}}_h^{k(j)}, \boldsymbol{\phi}(s, a) \right\rangle \right) \right|
$$

$$
+ \left| \sum_{j \in [n]} c_j(w) \left\langle \hat{\boldsymbol{\theta}}_h^{k(j)} - \tilde{\boldsymbol{\theta}}_h^k\left(w^{(j)}\right), \boldsymbol{\phi}(s, a) \right\rangle \right| + \left| \sum_{j \in [n]} c_j(w) \left\langle \tilde{\boldsymbol{\theta}}_h^k\left(w^{(j)}\right) - \boldsymbol{\theta}_h^k\left(w^{(j)}\right), \boldsymbol{\phi}(s, a) \right\rangle \right|
$$

$$
= \left| \sum_{j \in [n]} c_j(w) \left\langle \hat{\boldsymbol{\theta}}_h^{k(j)} - \tilde{\boldsymbol{\theta}}_h^k\left(w^{(j)}\right), \boldsymbol{\phi}(s, a) \right\rangle \right| + \left| \sum_{j \in [n]} c_j(w) \left\langle \tilde{\boldsymbol{\theta}}_h^k\left(w^{(j)}\right) - \boldsymbol{\theta}_h^k\left(w^{(j)}\right), \boldsymbol{\phi}(s, a) \right\rangle \right|
$$

(Eqn. (22))

$$
\leq 2L\beta \left\| \boldsymbol{\phi}(s, a) \right\|_{(\boldsymbol{\Lambda}_h^k)^{-1}}.
$$

(Lemma 5)

### B.3 PROOF OF OPTIMISTIC NATURE OF UCBLVD

**Lemma 7.** *Let* $\widetilde{\mathcal{W}} = \{w^\tau : \tau \in [K]\} \cup \{w^{(j)} : j \in [n]\}$. *Under the setting of Theorem 2 and conditioned on events* $\{\mathcal{E}_1(w)\}_{w \in \widetilde{\mathcal{W}}}$ *defined in* (9), *and with* $Q_h^k$ *computed as in* (7), *it holds that* $Q_h^k(s, a, w) \geq Q_h^*(s, a, w)$ *for all* $(s, a, w, h, k) \in \mathcal{S} \times \mathcal{A} \times \widetilde{\mathcal{W}} \times [H] \times [K]$.

*Proof.* We first note that conditioned on events $\{\mathcal{E}_1(w)\}_{w \in \widetilde{\mathcal{W}}}$, for all $(s, a, w, h, k) \in \mathcal{S} \times \mathcal{A} \times \widetilde{\mathcal{W}} \times [H] \times [K]$, it holds that

$$
\left| r_h(s, a, w) + \left\langle \hat{\boldsymbol{\xi}}_h^k, \boldsymbol{\psi}(s, a, w) \right\rangle - Q_h^\pi(s, a, w) - \mathbb{P}_h \left[ V_{h+1}^k(., w) - V_{h+1}^\pi(., w) \right](s, a) \right|
$$

$$
= \left| r_h(s, a, w) + \left\langle \hat{\boldsymbol{\xi}}_h^k, \boldsymbol{\psi}(s, a, w) \right\rangle - r_h(s, a, w) - \mathbb{P}_h \left[ V_{h+1}^k(., w) \right](s, a) \right|
$$

$$
= \left| \left\langle \hat{\boldsymbol{\xi}}_h^k, \boldsymbol{\psi}(s, a, w) \right\rangle - \mathbb{P}_h \left[ V_{h+1}^k(., w) \right](s, a) \right|
$$

$$
\leq 2L\beta \left\| \boldsymbol{\phi}(s, a) \right\|_{(\boldsymbol{\Lambda}_h^k)^{-1}},
$$

(Lemma 1)

for any policy $\pi$.

Now, we prove the lemma by induction. The statement holds for $H$ because $Q_{H+1}^k(., ., .) = Q_{H+1}^*(., ., .) = 0$ and thus conditioned events $\{\mathcal{E}_1(w)\}_{w \in \widetilde{\mathcal{W}}}$, defined in (9), for all $(s, a, w, k) \in \mathcal{S} \times \mathcal{A} \times \widetilde{\mathcal{W}} \times [K]$, we have

$$
\left| r_H(s, a, w) + \left\langle \hat{\boldsymbol{\xi}}_H^k, \boldsymbol{\psi}(s, a, w) \right\rangle - Q_H^*(s, a, w) \right| \leq 2L\beta \left\| \boldsymbol{\phi}(s, a) \right\|_{(\boldsymbol{\Lambda}_H^k)^{-1}}.
$$

Therefore, conditioned on events $\{\mathcal{E}_1(w)\}_{w \in \widetilde{\mathcal{W}}}$, for all $(s, a, w, k) \in \mathcal{S} \times \mathcal{A} \times \widetilde{\mathcal{W}} \times [K]$, we have

$$
\begin{aligned}
Q_H^*(s, a, w) &\leq r_H(s, a, w) + \left\langle \hat{\boldsymbol{\xi}}_H^k, \boldsymbol{\psi}(s, a, w) \right\rangle + 2L\beta \big\| \boldsymbol{\phi}(s, a) \big\|_{(\boldsymbol{\Lambda}_H^k)^{-1}} \\
&= \left\{ r_H(s, a, w) + \left\langle \hat{\boldsymbol{\xi}}_H^k, \boldsymbol{\psi}(s, a, w) \right\rangle + 2L\beta \big\| \boldsymbol{\phi}(s, a) \big\|_{(\boldsymbol{\Lambda}_H^k)^{-1}} \right\}^+ \\
&= Q_H^k(s, a, w),
\end{aligned}
$$

where the first equality follows from the fact that $Q_H^*(s, a, w) \geq 0$. Now, suppose the statement holds at time-step $h + 1$ and consider time-step $h$. Conditioned on events $\{\mathcal{E}_1(w)\}_{w \in \widetilde{\mathcal{W}}}$, for all $(s, a, w, h, k) \in \mathcal{S} \times \mathcal{A} \times \widetilde{\mathcal{W}} \times [H] \times [K]$, we have

$$
0 \leq r_h(s, a, w) + \left\langle \hat{\boldsymbol{\xi}}_h^k, \boldsymbol{\psi}(s, a, w) \right\rangle - Q_h^*(s, a, w) - \mathbb{P}_h \left[ V_{h+1}^k(., w) - V_{h+1}^*(., w) \right](s, a) + 2L\beta \big\| \boldsymbol{\phi}(s, a) \big\|_{(\boldsymbol{\Lambda}_h^k)^{-1}}
$$

$$
\leq r_h(s, a, w) + \left\langle \hat{\boldsymbol{\xi}}_h^k, \boldsymbol{\psi}(s, a, w) \right\rangle - Q_h^*(s, a, w) + 2L\beta \big\| \boldsymbol{\phi}(s, a) \big\|_{(\boldsymbol{\Lambda}_h^k)^{-1}}.
$$

(Induction assumption)

Therefore, conditioned on events $\{\mathcal{E}_1(w)\}_{w \in \widetilde{\mathcal{W}}}$, for all $(s, a, w, h, k) \in \mathcal{S} \times \mathcal{A} \times \widetilde{\mathcal{W}} \times [H] \times [K]$, we have

$$
\begin{aligned}
Q_h^*(s, a, w) &\leq r_h(s, a, w) + \left\langle \hat{\boldsymbol{\xi}}_h^k, \boldsymbol{\psi}(s, a, w) \right\rangle + 2L\beta \big\| \boldsymbol{\phi}(s, a) \big\|_{(\boldsymbol{\Lambda}_h^k)^{-1}} \\
&= \left\{ r_h(s, a, w) + \left\langle \hat{\boldsymbol{\xi}}_h^k, \boldsymbol{\psi}(s, a, w) \right\rangle + 2L\beta \big\| \boldsymbol{\phi}(s, a) \big\|_{(\boldsymbol{\Lambda}_h^k)^{-1}} \right\}^+ \\
&= Q_h^k(s, a, w),
\end{aligned}
$$

where the first equality follows from the fact that $Q_h^*(s, a, w) \geq 0$. This completes the proof. $\qquad \square$

## B.4 PROOF OF THEOREM 2

First, we bound the number of times Algorithm 2 updates $\hat{\boldsymbol{\xi}}_h^k$, i.e., number of planning calls. Let $P$ be the total number of updates and $k_p$ be the episode at which, the agent did replanning for the $p$-th time. Note that $\det \boldsymbol{\Lambda}_h^1 = \lambda^d$ and $\det \boldsymbol{\Lambda}_h^K \leq \operatorname{trace}(\boldsymbol{\Lambda}_h^K/d)^d \leq \left( \lambda + \frac{K}{d} \right)^d$, and consequently:

$$
\frac{\det \boldsymbol{\Lambda}_h^K}{\det \boldsymbol{\Lambda}_h^1} = \prod_{p=1}^{P} \frac{\det \boldsymbol{\Lambda}_h^{k_p}}{\det \boldsymbol{\Lambda}_h^{k_{p-1}}} \leq \left( 1 + \frac{K}{d\lambda} \right)^d,
$$

and therefore

$$
\prod_{h=1}^{H} \frac{\det \boldsymbol{\Lambda}_h^K}{\det \boldsymbol{\Lambda}_h^1} = \prod_{h=1}^{H} \prod_{p=1}^{P} \frac{\det \boldsymbol{\Lambda}_h^{k_p}}{\det \boldsymbol{\Lambda}_h^{k_{p-1}}} \leq \left( 1 + \frac{K}{d\lambda} \right)^{dH}. \tag{24}
$$

Since $1 \leq \frac{\det \boldsymbol{\Lambda}_h^{k_p}}{\det \boldsymbol{\Lambda}_h^{k_{p-1}}}$ for all $p \in [P]$, we can deduce from (24) that

$$
\exists h \in [H] \quad \text{such that} \quad e < \frac{\det \boldsymbol{\Lambda}_h^k}{\det \boldsymbol{\Lambda}_h^{\tilde{k}}}
$$

happens for at most $dH \log \left( 1 + \frac{K}{d\lambda} \right)$ number of episodes $k \in [K]$. This concludes that the number of planing calls in UCBlvd is $dH \log \left( 1 + \frac{K}{d\lambda} \right)$.

Now, we prove the regret bound. Let $\delta_h^k = V_h^{\tilde{k}}(s_h^k, w^k) - V_h^{\pi^k}(s_h^k, w^k)$ and $\xi_{h+1}^k = \mathbb{E}\left[\delta_{h+1}^k | s_h^k, a_h^k\right] - \delta_{h+1}^k$. Conditioned on events $\{\mathcal{E}_1(w)\}_{w \in \widetilde{\mathcal{W}}}$, for all $(s, a, w, h, k) \in \mathcal{S} \times \mathcal{A} \times \widetilde{\mathcal{W}} \times [H] \times [K]$, we have

$$Q_h^{\tilde{k}}(s, a, w) - Q_h^{\pi^k}(s, a, w) = r_h(s, a, w) + \left\langle \hat{\boldsymbol{\xi}}_h^{\tilde{k}}, \boldsymbol{\psi}(s, a, w) \right\rangle - Q_h^{\pi^k}(s, a, w) + 2L\beta \left\| \boldsymbol{\phi}(s, a) \right\|_{(\boldsymbol{\Lambda}_h^{\tilde{k}})^{-1}}$$

$$\leq \mathbb{P}_h \left[ V_{h+1}^{\tilde{k}}(., w) - V_{h+1}^{\pi^k}(., w) \right](s, a) + 4L\beta \left\| \boldsymbol{\phi}(s, a) \right\|_{(\boldsymbol{\Lambda}_h^{\tilde{k}})^{-1}}. \tag{25}$$

Note that $\delta_h^k \leq Q_h^{\tilde{k}}(s_h^k, a_h^k, w^k) - Q_h^{\pi^k}(s_h^k, a_h^k, w^k)$. Thus, combining (25), Lemma 5, and a union bound over $\widetilde{\mathcal{W}}$, we conclude that for all $(h, k) \in [H] \times [K]$, with probability at least $1 - \delta$, it holds that gives

$$\delta_h^k \leq \xi_{h+1}^k + \delta_{h+1}^k + 4L\beta \left\| \boldsymbol{\phi}(s_h^k, a_h^k) \right\|_{(\boldsymbol{\Lambda}_h^{\tilde{k}})^{-1}}.$$

Note that for any positive semi-definite matrices $\mathbf{A}$, $\mathbf{B}$, and $\mathbf{C}$ such that $\mathbf{A} = \mathbf{B} + \mathbf{C}$, we have:

$$\det(\mathbf{A}) \geq \det(\mathbf{B}), \ \det(\mathbf{A}) \geq \det(\mathbf{C}), \tag{26}$$

and for any $\mathbf{x} \neq 0$ ((Abbasi-Yadkori et al., 2011, Lemm. 12)):

$$\frac{\|\mathbf{x}\|_{\mathbf{A}}^2}{\|\mathbf{x}\|_{\mathbf{B}}^2} \leq \frac{\det(\mathbf{A})}{\det(\mathbf{B})} \quad \text{and} \quad \frac{\|\mathbf{x}\|_{\mathbf{B}^{-1}}^2}{\|\mathbf{x}\|_{\mathbf{A}^{-1}}^2} \leq \frac{\det(\mathbf{A})}{\det(\mathbf{B})}. \tag{27}$$

Now, we complete the regret analysis following similar steps as those of Theorem 1's proof:

$$R_K = \sum_{k=1}^{K} V_1^*(s_1^k, w^k) - V_1^{\pi^k}(s_1^k, w^k)$$

$$\leq \sum_{k=1}^{K} V_1^{\tilde{k}}(s_1^k, w^k) - V_1^{\pi^k}(s_1^k, w^k) \tag{Lemma 7}$$

$$= \sum_{k=1}^{K} \delta_1^k$$

$$\leq \sum_{k=1}^{K} \sum_{h=1}^{H} \xi_h^k + 4L\beta \sum_{k=1}^{K} \sum_{h=1}^{H} \left\| \boldsymbol{\phi}(s_h^k, a_h^k) \right\|_{\left(\boldsymbol{\Lambda}_h^{\tilde{k}}\right)^{-1}}$$

$$\leq \sum_{k=1}^{K} \sum_{h=1}^{H} \xi_h^k + 4L\beta \sum_{k=1}^{K} \sum_{h=1}^{H} \left\| \boldsymbol{\phi}(s_h^k, a_h^k) \right\|_{\left(\boldsymbol{\Lambda}_h^{k}\right)^{-1}} \sqrt{\frac{\det \boldsymbol{\Lambda}_h^k}{\det \boldsymbol{\Lambda}_h^{\tilde{k}}}} \tag{Eqn. (27)}$$

$$\leq 2H\sqrt{T \log(dT/\delta)} + 8HL\beta\sqrt{2dK \log(1 + K/\lambda)}$$

$$\leq \tilde{\mathcal{O}}\left( L\sqrt{\lambda(d^3 + md^2)H^3 T} \right).$$

## B.5 Discussion on the time complexity of UCBlvd and Lifelong-LSVI

In what follows, we clarify on how the time complexity of UCBlvd compares to that of Lifelong LSVI. When we compute $\left(\boldsymbol{\Lambda}_h^k\right)^{-1}$ by the Sherman-Morrison formula, the computational complexity of Lifelong-LSVI is dominated by Line 5 in computing $\max_{a \in \mathcal{A}} Q_{h+1}^k(s_{h+1}^\tau, a)$ for all $\tau \in [k]$. This takes $\mathcal{O}(d^2|\mathcal{A}|K)$ per step, which gives a total runtime $\mathcal{O}(d^2|\mathcal{A}|HK^2)$. In UCBlvd, every planning call takes $\tilde{\mathcal{O}}(md^2|\mathcal{A}|K + m^3d^3)$, where the second term is the time-complexity of thE convex QCQP with $m + 1$ constraints and $2md$ variables. This gives a total runtime of $\tilde{\mathcal{O}}(H^2(md^3|\mathcal{A}|K + m^3d^4))$. Therefore, UCBlvd enjoys a smaller time complexity by a factor of $K$ compared to that of Lifelong-LSVI, which is a significant reduction in practical scenarios where $K >> d' = md$.

---

**Algorithm 3:** UCBlvd with Unknown Rewards

---

1  **Set:** $Q_{H+1}^k(.,.,.) = 0, \ \forall k \in [K], \ \tilde{k} = 1$
2  **for** episodes $k = 1, \dots, K$ **do**
3       Observe the initial state $s_1^k$ and the task context $w^k$.
4       **if** $\exists h \in [H]$ *such that* $\frac{\det \mathbf{\Lambda}_h^k}{\det \mathbf{\Lambda}_h^{\tilde{k}}} > e$ **or** $\frac{\det \tilde{\mathbf{\Lambda}}_h^k}{\det \tilde{\mathbf{\Lambda}}_h^{\tilde{k}}} > e$ **then**
5           $\tilde{k} = k$
6           **for** time-steps $h = H, \dots, 1$ **do**
7               Compute $\hat{\boldsymbol{\xi}}_h^k$ as in (30).
8       **for** time-steps $h = 1, \dots, H$ **do**
9           Compute $Q_h^{\tilde{k}}(s_h^k, a, w^k)$ for all $a \in \mathcal{A}$ as in (28).
10          Play $a_h^k = \arg\max_{a \in \mathcal{A}} Q_h^{\tilde{k}}(s_h^k, a, w^k)$ and observe $s_{h+1}^k$ and $r_h^k$.

---

## C  DETAILS OF REMARK 1: UCBLVD WITH UNKNOWN REWARDS

In order for our analysis to go through, we need a slightly different completeness assumption as below:

**Assumption 4.** *Given feature maps $\boldsymbol{\phi} : \mathcal{S} \times \mathcal{A} \to \mathbb{R}^d$ and $\boldsymbol{\psi} : \mathcal{S} \times \mathcal{A} \times \mathcal{W} \to \mathbb{R}^{d'}$, consider function class*

$$\mathcal{F} = \left\{ f : f(s, w) = \min\left\{ \max_{a \in \mathcal{A}} \left\{ \langle \boldsymbol{\nu}, \boldsymbol{\psi}(s, a, w) \rangle + \beta \|\boldsymbol{\phi}(s, a)\|_{\mathbf{\Lambda}^{-1}} + \tilde{\beta} \|\boldsymbol{\psi}(s, a, w)\|_{\tilde{\mathbf{\Lambda}}^{-1}} \right\}^+, H \right\} \right.$$
$$\left. , \boldsymbol{\nu} \in \mathbb{R}^{d'}, \mathbf{\Lambda} \in \mathbf{S}_{++}^d, \tilde{\mathbf{\Lambda}} \in \mathbf{S}_{++}^{d'}, \beta \geq 0, \tilde{\beta} \geq 0 \right\}.$$

*Then for any $f \in \mathcal{F}$, and $h \in [H]$, there exists a vector $\boldsymbol{\xi}_h^f \in \mathbb{R}^{d'}$ with $\left\| \boldsymbol{\xi}_h^f \right\| \leq H\sqrt{d'}$ such that*

$$\mathbb{P}_h \left[ f(., w) \right](s, a) = \langle \boldsymbol{\xi}_h^f, \boldsymbol{\psi}(s, a, w) \rangle.$$

### C.1  OVERVIEW

Let $\boldsymbol{\psi}_h^\tau = \boldsymbol{\psi}(s_h^\tau, a_h^\tau, w^\tau)$. UCBlvd with unknown rewards works with the following action-value functions:

$$Q_h^k(s, a, w) = \left\{ \left\langle \tilde{\boldsymbol{\eta}}_h^k + \hat{\boldsymbol{\xi}}_h^k, \boldsymbol{\psi}(s, a, w) \right\rangle + \beta \|\boldsymbol{\phi}(s, a)\|_{(\mathbf{\Lambda}_h^k)^{-1}} + \tilde{\beta} \|\boldsymbol{\psi}(s, a, w)\|_{(\tilde{\mathbf{\Lambda}}_h^k)^{-1}} \right\}^+, \quad (28)$$

where

$$\tilde{\boldsymbol{\eta}}_h^k = \left( \tilde{\mathbf{\Lambda}}_h^k \right)^{-1} \sum_{\tau=1}^{k-1} \boldsymbol{\psi}_h^\tau . r_h^\tau \quad \text{and} \quad \tilde{\mathbf{\Lambda}}_h^k = \lambda \mathbf{I}_{md} + \sum_{\tau=1}^{k-1} \boldsymbol{\psi}_h^\tau \boldsymbol{\psi}_h^{\tau\top}, \quad (29)$$

and

$$\hat{\boldsymbol{\xi}}_h^k, \left\{ \hat{\boldsymbol{\theta}}_h^{k(j)} \right\}_{j \in [n]} = \operatorname*{arg\,min}_{\boldsymbol{\xi}, \left\{ \boldsymbol{\theta}^{(j)} \right\}_{j \in [n]}} \sum_{j \in [n]} \sum_{(s,a) \in \mathcal{D}} \left( \left\langle \boldsymbol{\theta}^{(j)}, \boldsymbol{\phi}(s, a) \right\rangle - \left\langle \boldsymbol{\xi}, \boldsymbol{\psi}\left( s, a, w^{(j)} \right) \right\rangle \right)^2 \quad (30)$$

$$\text{s.t.} \left\| \boldsymbol{\theta}^{(j)} - \tilde{\boldsymbol{\theta}}_h^k \left( w^{(j)} \right) \right\|_{\mathbf{\Lambda}_h^k} \leq \beta, \ \forall j \in [n] \quad \text{and} \quad \|\boldsymbol{\xi}\|_2 \leq H\sqrt{md},$$

$\mathcal{D} = \left\{ (s, a) : \boldsymbol{\phi}(s, a) \text{ are } d \text{ linearly independent vectors.} \right\}$, and $\tilde{\boldsymbol{\theta}}_h^k(w)$ and $\mathbf{\Lambda}_h^k$ are defined in (5).

We note that compared to (7), action-value function defined in (28) involves an extra term $\left\langle \tilde{\boldsymbol{\eta}}_h^k, \boldsymbol{\psi}(s,a,w) \right\rangle + \tilde{\beta} \|\boldsymbol{\psi}(s,a,w)\|_{(\tilde{\boldsymbol{\Lambda}}_h^k)^{-1}}$. This term is in fact an upper bound on $r_h(s,a,w)$. Specifically, from Theorem 2 in Abbasi-Yadkori et al. (2011), we know that for $\tilde{\beta} = \sqrt{\lambda md}$, it holds that

$$\left\| \boldsymbol{\eta}_h - \tilde{\boldsymbol{\eta}}_h^k \right\|_{\tilde{\boldsymbol{\Lambda}}_h^k} \leq \tilde{\beta}, \; \forall (h,k) \in [H] \times [K]. \tag{31}$$

**Theorem 3.** *Let $T = KH$. Under Assumptions 1, 3, and 4, the number of planning calls in Algorithm 3 is at most $dH \log\left(1 + \frac{K}{d\lambda}\right) + mdH \log\left(1 + \frac{K}{md\lambda}\right)$, and there exists an absolute constant $c > 0$ such that for any fixed $\delta \in (0, 0.5)$, if we set $\lambda = 1$, $\beta = cH\,(md)\sqrt{\log(mdT/\delta)}$ and $\tilde{\beta} = \sqrt{md}$ in Algorithm 3, then with probability at least $1 - 2\delta$, it holds that*

$$R_K \leq 2H\sqrt{T\log(dT/\delta)} + 4H\sqrt{K}\left(L\beta\sqrt{2d\log(1 + K/\lambda)} + \tilde{\beta}\sqrt{2md\log(1 + K/\lambda)}\right)$$

$$\leq \tilde{\mathcal{O}}\left(L\sqrt{m^2 d^3 H^3 T}\right).$$

## C.2 NECESSARY ANALYSIS FOR THE PROOF OF THEOREM 3

**Lemma 8.** *Let $c_\beta$ be a constant in the definition of $\beta$. Then, under Assumptions 1, 3, and 4, for a fixed $w$, there is an absolute constant $c_0$ independent of $c_\beta$, such that for all $(h,k) \in [H] \times [K]$, with probability at least $1 - \delta$ it holds that*

$$\left\| \sum_{\tau=1}^{k-1} \boldsymbol{\phi}_h^\tau \cdot \left( V_{h+1}^k(s_{h+1}^\tau, w) - \mathbb{P}_h[V_{h+1}^k(.,w)](s_h^\tau, a_h^\tau) \right) \right\|_{(\boldsymbol{\Lambda}_h^k)^{-1}} \leq c_0 mdH\sqrt{\log((c_\beta + 1)mdT/\delta)},$$

*where $c_0$ and $c_\beta$ are two independent absolute constants.*

*Proof.* We note that $\left\| \tilde{\boldsymbol{\eta}}_h^k + \hat{\boldsymbol{\xi}}_h^k \right\|_2 \leq H\sqrt{md} + K/\lambda$ and $\left\| \left(\boldsymbol{\Lambda}_h^k\right)^{-1} \right\| \leq \frac{1}{\lambda}$ and $\left\| \left(\tilde{\boldsymbol{\Lambda}}_h^k\right)^{-1} \right\| \leq \frac{1}{\lambda}$. Thus, Lemmas 19 and 23 together imply that for all $(h,k) \in [H] \times [K]$, with probability at least $1 - \delta$ it holds that

$$\left\| \sum_{\tau=1}^{k-1} \boldsymbol{\phi}_h^\tau \left( V_{h+1}^k(s_{h+1}^\tau, w) - \mathbb{P}_h[V_{h+1}^k(.,w)](s_h^\tau, a_h^\tau) \right) \right\|_{(\boldsymbol{\Lambda}_h^k)^{-1}}^2$$

$$\leq 4H^2 \left( \frac{d}{2}\log\left(\frac{k+\lambda}{\lambda}\right) + md\log(1 + 8H\sqrt{md}/\epsilon) + d^2\log\left(\frac{1 + 32L^2\beta^2\sqrt{d}}{\lambda\epsilon^2}\right) \right)$$

$$+ m^2 d^2 \log\left(\frac{1 + 8\tilde{\beta}^2\sqrt{md}}{\lambda\epsilon^2}\right) + \log\left(\frac{1}{\delta}\right) \right) + \frac{8k^2\epsilon^2}{\lambda}.$$

If we let $\epsilon = \frac{dH}{k}$ and $\beta = c_\beta(md)H\sqrt{\log(mdT/\delta)}$, then, there exists an absolute constant $C > 0$ that is independent of $c_\beta$ such that

$$\left\| \sum_{\tau=1}^{k-1} \boldsymbol{\phi}_h^\tau \left( V_{h+1}^k(s_{h+1}^\tau, w) - \mathbb{P}_h[V_{h+1}^k(.,w)](s_h^\tau, a_h^\tau) \right) \right\|_{(\boldsymbol{\Lambda}_h^k)^{-1}}^2 \leq C(m^2 d^2)H^2 \log\left((c_\beta + 1)mdT/\delta\right).$$

$\square$

**Lemma 9.** *Under Assumptions 1, 3, and 4, if we let $\beta = cmdH\sqrt{\lambda\log(mdT/\delta)}$ with an absolute constant $c > 0$, then the event*

$$\mathcal{E}_3(w) := \left\{ \left\| \boldsymbol{\theta}_h^k(w) - \tilde{\boldsymbol{\theta}}_h^k(w) \right\|_{\boldsymbol{\Lambda}_h^k} \leq \beta, \; \forall (h,k) \in [H] \times [K] \right\}. \tag{32}$$

*holds with probability at least $1 - \delta$ for a fixed $w$.*

*Proof.* The proof follows the same steps as those of Lemma 5, except that it uses Lemma 8 instead of Lemma 6 due to different structure of action-value functions $Q_h^k$ in this section. □

**Lemma 10.** *Let* $\widetilde{\mathcal{W}} = \{w^\tau : \tau \in [K]\} \cup \{w^{(j)} : j \in [n]\}$. *Under the setting of Theorem 3 and conditioned on events* $\{\mathcal{E}_3(w)\}_{w \in \widetilde{\mathcal{W}}}$ *defined in* (32), *for all* $(s, a, w, h, k) \in \mathcal{S} \times \mathcal{A} \times \widetilde{\mathcal{W}} \times [H] \times [K]$, *it holds that*

$$\left| \left\langle \hat{\boldsymbol{\xi}}_h^k, \boldsymbol{\psi}(s, a, w) \right\rangle - \mathbb{P}_h \left[ V_{h+1}^k(., w) \right](s, a) \right| \leq 2L\beta \left\| \boldsymbol{\phi}(s, a) \right\|_{(\boldsymbol{\Lambda}_h^k)^{-1}}.$$

*Proof.* The proof follows the exact same steps as those of Lemma 1's proof. □

**Lemma 11.** *Let* $\widetilde{\mathcal{W}} = \{w^\tau : \tau \in [K]\} \cup \{w^{(j)} : j \in [n]\}$. *Under the setting of Theorem 3 and conditioned on events* $\{\mathcal{E}_3(w)\}_{w \in \widetilde{\mathcal{W}}}$ *defined in* (32), *and with* $Q_h^k$ *computed as in* (28), *it holds that* $Q_h^k(s, a, w) \geq Q_h^*(s, a, w)$ *for all* $(s, a, w, h, k) \in \mathcal{S} \times \mathcal{A} \times \widetilde{\mathcal{W}} \times [H] \times [K]$.

*Proof.* We first note that conditioned on events $\{\mathcal{E}_3(w)\}_{w \in \widetilde{\mathcal{W}}}$, for all $(s, a, w, h, k) \in \mathcal{S} \times \mathcal{A} \times \widetilde{\mathcal{W}} \times [H] \times [K]$, it holds that

$$\left| \left\langle \tilde{\boldsymbol{\eta}}_h^k + \hat{\boldsymbol{\xi}}_h^k, \boldsymbol{\psi}(s, a, w) \right\rangle - Q_h^\pi(s, a, w) - \mathbb{P}_h \left[ V_{h+1}^k(., w) - V_{h+1}^\pi(., w) \right](s, a) \right|$$

$$= \left| \left\langle \tilde{\boldsymbol{\eta}}_h^k + \hat{\boldsymbol{\xi}}_h^k, \boldsymbol{\psi}(s, a, w) \right\rangle - r_h(s, a, w) - \mathbb{P}_h \left[ V_{h+1}^k(., w) \right](s, a) \right|$$

$$\leq \left| \left\langle \hat{\boldsymbol{\xi}}_h^k, \boldsymbol{\psi}(s, a, w) \right\rangle - \mathbb{P}_h \left[ V_{h+1}^k(., w) \right](s, a) \right| + \tilde{\beta} \left\| \boldsymbol{\psi}(s, a, w) \right\|_{(\tilde{\boldsymbol{\Lambda}}_h^k)^{-1}} \qquad \text{(Eqn. (31))}$$

$$\leq 2L\beta \left\| \boldsymbol{\phi}(s, a) \right\|_{(\boldsymbol{\Lambda}_h^k)^{-1}} + \tilde{\beta} \left\| \boldsymbol{\psi}(s, a, w) \right\|_{(\tilde{\boldsymbol{\Lambda}}_h^k)^{-1}}, \qquad \text{(Lemma 10)}$$

for any policy $\pi$.

Now, we prove the lemma by induction. The statement holds for $H$ because $Q_{H+1}^k(., ., .) = Q_{H+1}^*(., ., .) = 0$ and thus conditioned events $\{\mathcal{E}_3(w)\}_{w \in \widetilde{\mathcal{W}}}$, defined in (32), for all $(s, a, w, k) \in \mathcal{S} \times \mathcal{A} \times \widetilde{\mathcal{W}} \times [K]$, we have

$$\left| \left\langle \tilde{\boldsymbol{\eta}}_H^k + \hat{\boldsymbol{\xi}}_H^k, \boldsymbol{\psi}(s, a, w) \right\rangle - Q_H^*(s, a, w) \right| \leq 2L\beta \left\| \boldsymbol{\phi}(s, a) \right\|_{(\boldsymbol{\Lambda}_H^k)^{-1}} + \tilde{\beta} \left\| \boldsymbol{\psi}(s, a, w) \right\|_{(\tilde{\boldsymbol{\Lambda}}_H^k)^{-1}}. \tag{33}$$

Therefore, conditioned on events $\{\mathcal{E}_3(w)\}_{w \in \widetilde{\mathcal{W}}}$, for all $(s, a, w, k) \in \mathcal{S} \times \mathcal{A} \times \widetilde{\mathcal{W}} \times [K]$, we have

$$Q_H^*(s, a, w) \leq \left\langle \tilde{\boldsymbol{\eta}}_H^k + \hat{\boldsymbol{\xi}}_H^k, \boldsymbol{\psi}(s, a, w) \right\rangle + 2L\beta \left\| \boldsymbol{\phi}(s, a) \right\|_{(\boldsymbol{\Lambda}_H^k)^{-1}} + \tilde{\beta} \left\| \boldsymbol{\psi}(s, a, w) \right\|_{(\tilde{\boldsymbol{\Lambda}}_H^k)^{-1}}$$

$$= \left\{ \left\langle \tilde{\boldsymbol{\eta}}_H^k + \hat{\boldsymbol{\xi}}_H^k, \boldsymbol{\psi}(s, a, w) \right\rangle + 2L\beta \left\| \boldsymbol{\phi}(s, a) \right\|_{(\boldsymbol{\Lambda}_H^k)^{-1}} + \tilde{\beta} \left\| \boldsymbol{\psi}(s, a, w) \right\|_{(\tilde{\boldsymbol{\Lambda}}_H^k)^{-1}} \right\}^+$$

$$= Q_H^k(s, a, w),$$

where the first equality follows from the fact that $Q_H^*(s, a, w) \geq 0$. Now, suppose the statement holds at time-step $h + 1$ and consider time-step $h$. Conditioned on events $\{\mathcal{E}_3(w)\}_{w \in \widetilde{\mathcal{W}}}$, for all

$(s, a, w, h, k) \in \mathcal{S} \times \mathcal{A} \times \widetilde{\mathcal{W}} \times [H] \times [K]$, we have

$$
\begin{aligned}
0 \leq & \left\langle \tilde{\boldsymbol{\eta}}_h^k + \hat{\boldsymbol{\xi}}_h^k, \boldsymbol{\psi}(s, a, w) \right\rangle - Q_h^*(s, a, w) - \mathbb{P}_h \left[ V_{h+1}^k(., w) - V_{h+1}^*(., w) \right] (s, a) \\
& + 2L\beta \|\boldsymbol{\phi}(s, a)\|_{(\boldsymbol{\Lambda}_h^k)^{-1}} + \tilde{\beta} \|\boldsymbol{\psi}(s, a, w)\|_{(\tilde{\boldsymbol{\Lambda}}_h^k)^{-1}} \\
\leq & \left\langle \tilde{\boldsymbol{\eta}}_h^k + \hat{\boldsymbol{\xi}}_h^k, \boldsymbol{\psi}(s, a, w) \right\rangle - Q_h^*(s, a, w) + 2L\beta \|\boldsymbol{\phi}(s, a)\|_{(\boldsymbol{\Lambda}_h^k)^{-1}} + \tilde{\beta} \|\boldsymbol{\psi}(s, a, w)\|_{(\tilde{\boldsymbol{\Lambda}}_h^k)^{-1}}.
\end{aligned}
$$

(Induction assumption)

Therefore, conditioned on events $\{\mathcal{E}_3(w)\}_{w \in \widetilde{\mathcal{W}}}$, for all $(s, a, w, h, k) \in \mathcal{S} \times \mathcal{A} \times \widetilde{\mathcal{W}} \times [H] \times [K]$, we have

$$
\begin{aligned}
Q_h^*(s, a, w) & \leq \left\langle \tilde{\boldsymbol{\eta}}_h^k + \hat{\boldsymbol{\xi}}_h^k, \boldsymbol{\psi}(s, a, w) \right\rangle + 2L\beta \|\boldsymbol{\phi}(s, a)\|_{(\boldsymbol{\Lambda}_h^k)^{-1}} + \tilde{\beta} \|\boldsymbol{\psi}(s, a, w)\|_{(\tilde{\boldsymbol{\Lambda}}_h^k)^{-1}} \\
& = \left\{ \left\langle \tilde{\boldsymbol{\eta}}_h^k + \hat{\boldsymbol{\xi}}_h^k, \boldsymbol{\psi}(s, a, w) \right\rangle + 2L\beta \|\boldsymbol{\phi}(s, a)\|_{(\boldsymbol{\Lambda}_h^k)^{-1}} + \tilde{\beta} \|\boldsymbol{\psi}(s, a, w)\|_{(\tilde{\boldsymbol{\Lambda}}_h^k)^{-1}} \right\}^+ \\
& = Q_h^k(s, a, w),
\end{aligned}
$$

where the first equality follows from the fact that $Q_h^*(s, a, w) \geq 0$. This completes the proof. $\qquad\square$

## C.3    PROOF OF THEOREM 3

First, we bound the number of times Algorithm 3 updates $\hat{\boldsymbol{\xi}}_h^k$, i.e., number of planning calls. Let $P$ be the total number of policy updates and $k_p$ be the episode at, the agent did replanning for the $p$-th time. Note that $\det \boldsymbol{\Lambda}_h^1 = \lambda^d$ and $\det \boldsymbol{\Lambda}_h^K \leq \text{trace}(\boldsymbol{\Lambda}_h^K/d)^d \leq \left(\lambda + \frac{K}{d}\right)^d$, and consequently:

$$
\frac{\det \boldsymbol{\Lambda}_h^K}{\det \boldsymbol{\Lambda}_h^1} = \prod_{p=1}^P \frac{\det \boldsymbol{\Lambda}_h^{k_p}}{\det \boldsymbol{\Lambda}_h^{k_{p-1}}} \leq \left(1 + \frac{K}{d\lambda}\right)^d,
$$

and therefore

$$
\prod_{h=1}^H \frac{\det \boldsymbol{\Lambda}_h^K}{\det \boldsymbol{\Lambda}_h^1} = \prod_{h=1}^H \prod_{p=1}^P \frac{\det \boldsymbol{\Lambda}_h^{k_p}}{\det \boldsymbol{\Lambda}_h^{k_{p-1}}} \leq \left(1 + \frac{K}{d\lambda}\right)^{dH}. \tag{34}
$$

We similarly have

$$
\prod_{h=1}^H \frac{\det \tilde{\boldsymbol{\Lambda}}_h^K}{\det \tilde{\boldsymbol{\Lambda}}_h^1} = \prod_{h=1}^H \prod_{p=1}^P \frac{\det \tilde{\boldsymbol{\Lambda}}_h^{k_p}}{\det \tilde{\boldsymbol{\Lambda}}_h^{k_{p-1}}} \leq \left(1 + \frac{K}{md\lambda}\right)^{mdH}. \tag{35}
$$

Since $1 \leq \frac{\det \boldsymbol{\Lambda}_h^{k_p}}{\det \boldsymbol{\Lambda}_h^{k_{p-1}}}$ for all $p \in [P]$, we can deduce from (34) and (35) that

$$
\exists h \in [H] \quad \text{such that} \quad e < \frac{\det \boldsymbol{\Lambda}_h^k}{\det \boldsymbol{\Lambda}_h^{\tilde{k}}} \quad \text{or} \quad e < \frac{\det \tilde{\boldsymbol{\Lambda}}_h^k}{\det \tilde{\boldsymbol{\Lambda}}_h^{\tilde{k}}} \tag{36}
$$

happens for at most $dH \log\left(1 + \frac{K}{d\lambda}\right) + mdH \log\left(1 + \frac{K}{md\lambda}\right)$ number of episodes $k \in [K]$. This concludes that number of planning calls in Algorithm 3 is at most $dH \log\left(1 + \frac{K}{d\lambda}\right) + mdH \log\left(1 + \frac{K}{md\lambda}\right)$.

Now, we prove the regret bound. Let $\delta_h^k = V_h^{\tilde{k}}(s_h^k, w^k) - V_h^{\pi^k}(s_h^k, w^k)$ and $\xi_{h+1}^k = \mathbb{E}\left[\delta_{h+1}^k | s_h^k, a_h^k\right] - \delta_{h+1}^k$. Conditioned on events $\{\mathcal{E}_3(w)\}_{w \in \widetilde{\mathcal{W}}}$, for all $(s, a, w, h, k) \in \mathcal{S} \times \mathcal{A} \times$

$\widetilde{\mathcal{W}} \times [H] \times [K]$, we have

$$Q_h^{\tilde{k}}(s, a, w) - Q_h^{\pi^k}(s, a, w) = \left\langle \tilde{\boldsymbol{\eta}}_h^{\tilde{k}} + \hat{\boldsymbol{\xi}}_h^{\tilde{k}}, \boldsymbol{\psi}(s, a, w) \right\rangle - Q_h^{\pi^k}(s, a, w) + 2L\beta \big\|\boldsymbol{\phi}(s, a)\big\|_{(\boldsymbol{\Lambda}_h^{\tilde{k}})^{-1}} + \tilde{\beta} \big\|\boldsymbol{\psi}(s, a, w)\big\|_{(\tilde{\boldsymbol{\Lambda}}_h^{\tilde{k}})^{-1}}$$

$$\leq \mathbb{P}_h \left[ V_{h+1}^{\tilde{k}}(., w) - V_{h+1}^{\pi^k}(., w) \right](s, a) + 4L\beta \big\|\boldsymbol{\phi}(s, a)\big\|_{(\boldsymbol{\Lambda}_h^{\tilde{k}})^{-1}} + 2\tilde{\beta} \big\|\boldsymbol{\psi}(s, a, w)\big\|_{(\tilde{\boldsymbol{\Lambda}}_h^{\tilde{k}})^{-1}}.$$
(37)

Note that $\delta_h^k \leq Q_h^{\tilde{k}}(s_h^k, a_h^k, w^k) - Q_h^{\pi^k}(s_h^k, a_h^k, w^k)$. Thus, combining (37), Lemma 9, and a union bound over $\widetilde{\mathcal{W}}$, we conclude that for all $(h, k) \in [H] \times [K]$, with probability at least $1 - \delta$, it holds that gives

$$\delta_h^k \leq \xi_{h+1}^k + \delta_{h+1}^k + 4L\beta \big\|\boldsymbol{\phi}(s_h^k, a_h^k)\big\|_{(\boldsymbol{\Lambda}_h^{\tilde{k}})^{-1}} + 2\tilde{\beta} \big\|\boldsymbol{\psi}(s_h^k, a_h^k, w^k)\big\|_{(\tilde{\boldsymbol{\Lambda}}_h^{\tilde{k}})^{-1}}.$$

Now, we complete the regret analysis following similar steps as those of Theorem 1's proof:

$$R_K = \sum_{k=1}^{K} V_1^*(s_1^k, w^k) - V_1^{\pi^k}(s_1^k, w^k)$$

$$\leq \sum_{k=1}^{K} V_1^{\tilde{k}}(s_1^k, w^k) - V_1^{\pi^k}(s_1^k, w^k) \qquad \text{(Lemma 11)}$$

$$= \sum_{k=1}^{K} \delta_1^k$$

$$\leq \sum_{k=1}^{K} \sum_{h=1}^{H} \xi_h^k + 4L\beta \sum_{k=1}^{K} \sum_{h=1}^{H} \big\|\boldsymbol{\phi}(s_h^k, a_h^k)\big\|_{(\boldsymbol{\Lambda}_h^{\tilde{k}})^{-1}} + 2\tilde{\beta} \sum_{k=1}^{K} \sum_{h=1}^{H} \big\|\boldsymbol{\psi}(s_h^k, a_h^k, w^k)\big\|_{(\tilde{\boldsymbol{\Lambda}}_h^{\tilde{k}})^{-1}}$$

$$\leq \sum_{k=1}^{K} \sum_{h=1}^{H} \xi_h^k + 4L\beta \sum_{k=1}^{K} \sum_{h=1}^{H} \big\|\boldsymbol{\phi}(s_h^k, a_h^k)\big\|_{(\boldsymbol{\Lambda}_h^k)^{-1}} \sqrt{\frac{\det \boldsymbol{\Lambda}_h^k}{\det \boldsymbol{\Lambda}_h^{\tilde{k}}}} + 2\tilde{\beta} \sum_{k=1}^{K} \sum_{h=1}^{H} \big\|\boldsymbol{\psi}(s_h^k, a_h^k, w^k)\big\|_{(\tilde{\boldsymbol{\Lambda}}_h^k)^{-1}} \sqrt{\frac{\det \tilde{\boldsymbol{\Lambda}}_h^k}{\det \tilde{\boldsymbol{\Lambda}}_h^{\tilde{k}}}}$$
(Eqn. (27))

$$\leq 2H\sqrt{T \log(dT/\delta)} + 4H\sqrt{K} \left( L\beta\sqrt{2d \log(1 + K/\lambda)} + \tilde{\beta}\sqrt{2md \log(1 + K/\lambda)} \right)$$

$$\leq \tilde{\mathcal{O}}\left( L\sqrt{\lambda m^2 d^3 H^3 T} \right).$$

## D   DETAILS OF REMARK 2: RELAXATION OF ASSUMPTION 3

In this section, we replace Assumption 3 with the following assumption:

**Assumption 5.** *There is a known set $\{w^{(1)}, w^{(2)}, \ldots, w^{(n)}\}$ of $n \leq d'$ tasks such that $\boldsymbol{\psi}(s, a, w) \in$ $\mathrm{Span}\left(\left\{\boldsymbol{\psi}(s, a, w^{(j)})\right\}_{j \in [n]}\right)$ for all $(s, a, w) \in \mathcal{S} \times \mathcal{A} \times \mathcal{W}$. This implies that for any $(s, a, w) \in \mathcal{S} \times \mathcal{A} \times \mathcal{W}$, there exist coefficients $\{c_j(s, a, w)\}_{j \in [n]}$ such that*

$$\boldsymbol{\psi}(s, a, w) = \sum_{j \in [n]} c_j(s, a, w) \boldsymbol{\psi}\left(s, a, w^{(j)}\right). \qquad (38)$$

*Moreover, $\sum_{j \in [n]} |c_j(s, a, w)| \leq L$ for all $(s, a, w) \in \mathcal{S} \times \mathcal{A} \times \mathcal{W}$.*

Define the concatenated mapping $\tilde{\boldsymbol{\psi}}$ : $\mathcal{S} \times \mathcal{A} \times \mathcal{W} \rightarrow \mathbb{R}^{d+d'}$ such that $\tilde{\boldsymbol{\psi}}(s, a, w) = \left[\boldsymbol{\phi}(s, a)^\top, \boldsymbol{\psi}(s, a, w)^\top\right]^\top$. For any $w \in \mathcal{W}$, define $\mathcal{D}(w) = \left\{(s, a) : \tilde{\boldsymbol{\psi}}(s, a, w) \text{ are } d + d' \text{ linearly independent vectors.}\right\}$. Given Assumption 5, we mod-

---

**Algorithm 4:** Modified UCBlvd

---

1 **Set:** $Q_{H+1}^k(.,.,.) = 0, \ \forall k \in [K], \ \tilde{k} = 1$

2 **for** episodes $k = 1, \ldots, K$ **do**

3     Observe the initial state $s_1^k$ and the task context $w^k$.

4     **if** $\exists h \in [H]$ *such that* $\frac{\det \mathbf{\Lambda}_h^k}{\det \mathbf{\Lambda}_h^{\tilde{k}}} > e$ **then**

5        $\tilde{k} = k$

6        **for** time-steps $h = H, \ldots, 1$ **do**

7           Compute $\hat{\boldsymbol{\xi}}_h^k$ as in (39).

8     **for** time-steps $h = 1, \ldots, H$ **do**

9        Compute $Q_h^{\tilde{k}}(s_h^k, a, w^k)$ for all $a \in \mathcal{A}$ as in (7).

10        Play $a_h^k = \arg\max_{a \in \mathcal{A}} Q_h^{\tilde{k}}(s_h^k, a, w^k)$ and observe $s_{h+1}^k$ and $r_h^k$.

---

ify the planning step of UCBlvd to the following:

$$\hat{\boldsymbol{\xi}}_h^k, \left\{\hat{\boldsymbol{\theta}}_h^{k(j)}\right\}_{j \in [n]} = \arg\min_{\boldsymbol{\xi}, \left\{\boldsymbol{\theta}^{(j)}\right\}_{j \in [n]}} \sum_{j \in [n]} \sum_{(s,a) \in \mathcal{D}(w^{(j)})} \left(\left\langle \boldsymbol{\theta}^{(j)}, \boldsymbol{\phi}(s,a)\right\rangle - \left\langle \boldsymbol{\xi}, \boldsymbol{\psi}\left(s, a, w^{(j)}\right)\right\rangle\right)^2 \tag{39}$$

$$\text{s.t. } \left\|\boldsymbol{\theta}^{(j)} - \tilde{\boldsymbol{\theta}}_h^k\left(w^{(j)}\right)\right\|_{\mathbf{\Lambda}_h^k} \leq \beta, \ \forall j \in [n] \quad \text{and} \quad \|\boldsymbol{\xi}\|_2 \leq H\sqrt{d'}.$$

The only change we make in Algorithm 2 is in Line 9, in which $\hat{\boldsymbol{\xi}}_h^k$ is now computed as defined in (39). We present this modification in Algorithm 4 for completeness.

**Theorem 4.** *Let $T = KH$. Under Assumptions 1, 2, and 5, the number or planning calls in Algorithm 4 is at most $dH \log\left(1 + \frac{K}{d\lambda}\right)$ and there exists an absolute constant $c > 0$ such that for any fixed $\delta \in (0, 0.5)$, if we set $\lambda = 1$ and $\beta = cH\left(d + \sqrt{d'}\right)\sqrt{\lambda \log(dd'T/\delta)}$ in Algorithm 4, then with probability at least $1 - 2\delta$, it holds that*

$$R_K \leq 2H\sqrt{T\log(dT/\delta)} + 8HL\beta\sqrt{2dK\log(K)} \leq \tilde{\mathcal{O}}\left(L\sqrt{(d^3 + dd')H^3 T}\right). \tag{40}$$

Proof of Theorem 4 follows exactly the same steps as those of Theorem 2. The only difference is the proof of Lemma 1, which we clarify in the proof of following lemma.

**Lemma 12.** *Let $\widetilde{\mathcal{W}} = \{w^\tau : \tau \in [K]\} \cup \{w^{(j)} : j \in [n]\}$. Under Assumptions 1, 2, and 5, if we let $\beta = cH\left(d + \sqrt{d'}\right)\sqrt{\lambda \log(dd'T/\delta)}$ with an absolute constant $c > 0$, then for all $(s, a, w, h, k) \in \mathcal{S} \times \mathcal{A} \times \mathcal{W} \times [H] \times [K]$ with probability at least $1 - \delta$, it holds that*

$$\left|\left\langle \hat{\boldsymbol{\xi}}_h^k, \boldsymbol{\psi}(s, a, w)\right\rangle - \mathbb{P}_h\left[V_{h+1}^k(., w)\right](s, a)\right| \leq 2L\beta\left\|\boldsymbol{\phi}(s, a)\right\|_{(\mathbf{\Lambda}_h^k)^{-1}}.$$

*Proof.* We let $\tilde{\boldsymbol{\psi}}_i(w) = \left[\boldsymbol{\phi}_i^\top, \boldsymbol{\psi}_i(w)^\top\right]^\top$ be the $i$-th element of $\tilde{\mathcal{D}}(w) = \left\{\tilde{\boldsymbol{\psi}}(s, a, w) : (s, a) \in \mathcal{D}(w)\right\}$ and for any triple $(s, a, w) \in \mathcal{S} \times \mathcal{A} \times \mathcal{W}$, we let $\{c_i'(s, a, w)\}_{i \in [d+d']}$ be the coefficients such that

$$\tilde{\boldsymbol{\psi}}(s, a, w) = \sum_{i \in [d+d']} c_i'(s, a, w)\tilde{\boldsymbol{\psi}}_i(w),$$

which implies that

$$\boldsymbol{\phi}(s, a) = \sum_{i \in [d+d']} c_i'(s, a, w)\boldsymbol{\phi}_i \quad \text{and} \quad \boldsymbol{\psi}(s, a, w) = \sum_{i \in [d+d']} c_i'(s, a, w)\boldsymbol{\psi}_i(w). \tag{41}$$

Thanks to Assumption 2 and conditioned on events $\{\mathcal{E}_1(w)\}_{w \in \widetilde{\mathcal{W}}}$, one set of solution for (39) is $\left\{ \boldsymbol{\theta}_h^k \left( w^{(j)} \right) \right\}_{j \in [n]}$ and $\boldsymbol{\xi}_h^{V_{h+1}^k}$ with corresponding zero optimal objective value. Therefore, it holds that

$$\left\langle \hat{\boldsymbol{\theta}}_h^{k(j)}, \boldsymbol{\phi}_i \right\rangle = \left\langle \hat{\boldsymbol{\xi}}_h^k, \boldsymbol{\psi}_i \left( w^{(j)} \right) \right\rangle, \quad \forall (i,j) \in [d + d'] \times [n]. \tag{42}$$

Moreover, for any triple $(s, a, j) \in \mathcal{S} \times \mathcal{A} \times [n]$, we have

$$\left\langle \hat{\boldsymbol{\xi}}_h^k, \boldsymbol{\psi} \left( s, a, w^{(j)} \right) \right\rangle = \sum_{i \in [d+d']} c_i' \left( s, a, w^{(j)} \right) \left\langle \hat{\boldsymbol{\xi}}_h^k, \boldsymbol{\psi}_i \left( w^{(j)} \right) \right\rangle \tag{Eqn. (41)}$$

$$= \sum_{i \in [d+d']} c_i' \left( s, a, w^{(j)} \right) \left\langle \hat{\boldsymbol{\theta}}_h^{k(j)}, \boldsymbol{\phi}_i \right\rangle \tag{Eqn. (42)}$$

$$= \left\langle \hat{\boldsymbol{\theta}}_h^{k(j)}, \boldsymbol{\phi}(s,a) \right\rangle. \tag{43}$$

For any $(s, a, w) \in \mathcal{S} \times \mathcal{A} \times \mathcal{W}$, it holds that

$$\mathbb{P}_h \left[ V_{h+1}^k(., w) \right] (s,a) = \left\langle \boldsymbol{\theta}_h^k(w), \boldsymbol{\phi}(s,a) \right\rangle \tag{Eqn. (4)}$$

$$= \left\langle \boldsymbol{\xi}_h^{V_{h+1}^k}, \boldsymbol{\psi}(s, a, w) \right\rangle \tag{Assumption 2}$$

$$= \sum_{j \in [n]} c_j(s, a, w) \left\langle \boldsymbol{\xi}_h^{V_{h+1}^k}, \boldsymbol{\psi} \left( s, a, w^{(j)} \right) \right\rangle \tag{Eqn. (38)}$$

$$= \sum_{j \in [n]} c_j(s, a, w) \mathbb{P}_h \left[ V_{h+1}^k \left( ., w^{(j)} \right) \right] (s,a)) \tag{Assumption 2}$$

$$= \sum_{j \in [n]} c_j(s, a, w) \left\langle \boldsymbol{\theta}_h^k \left( w^{(j)} \right), \boldsymbol{\phi}(s,a) \right\rangle. \tag{44}$$

Finally, conditioned on events $\{\mathcal{E}_1(w)\}_{w \in \widetilde{\mathcal{W}}}$, for all $(s, a, w, h, k) \in \mathcal{S} \times \mathcal{A} \times \widetilde{\mathcal{W}} \times [H] \times [K]$, it holds that

$$\left| \left\langle \hat{\boldsymbol{\xi}}_h^k, \boldsymbol{\psi}(s, a, w) \right\rangle - \mathbb{P}_h \left[ V_{h+1}^k(., w) \right] (s,a) \right| \tag{45}$$

$$= \left| \left\langle \hat{\boldsymbol{\xi}}_h^k, \boldsymbol{\psi}(s, a, w) \right\rangle - \left\langle \boldsymbol{\theta}_h^k(w), \boldsymbol{\phi}(s,a) \right\rangle \right|$$

$$= \left| \sum_{j \in [n]} c_j(s, a, w) \left( \left\langle \hat{\boldsymbol{\xi}}_h^k, \boldsymbol{\psi} \left( s, a, w^{(j)} \right) \right\rangle - \left\langle \boldsymbol{\theta}_h^k \left( w^{(j)} \right), \boldsymbol{\phi}(s,a) \right\rangle \right) \right| \tag{Eqns. (38) and (23)}$$

$$\leq \left| \sum_{j \in [n]} c_j(s, a, w) \left( \left\langle \hat{\boldsymbol{\xi}}_h^k, \boldsymbol{\psi} \left( s, a, w^{(j)} \right) \right\rangle - \left\langle \hat{\boldsymbol{\theta}}_h^{k(j)}, \boldsymbol{\phi}(s,a) \right\rangle \right) \right|$$

$$+ \left| \sum_{j \in [n]} c_j(s, a, w) \left\langle \hat{\boldsymbol{\theta}}_h^{k(j)} - \tilde{\boldsymbol{\theta}}_h^k \left( w^{(j)} \right), \boldsymbol{\phi}(s,a) \right\rangle \right| + \left| \sum_{j \in [n]} c_j(s, a, w) \left\langle \tilde{\boldsymbol{\theta}}_h^k \left( w^{(j)} \right) - \boldsymbol{\theta}_h^k \left( w^{(j)} \right), \boldsymbol{\phi}(s,a) \right\rangle \right|$$

$$= \left| \sum_{j \in [n]} c_j(s, a, w) \left\langle \hat{\boldsymbol{\theta}}_h^{k(j)} - \tilde{\boldsymbol{\theta}}_h^k \left( w^{(j)} \right), \boldsymbol{\phi}(s,a) \right\rangle \right| + \left| \sum_{j \in [n]} c_j(s, a, w) \left\langle \tilde{\boldsymbol{\theta}}_h^k \left( w^{(j)} \right) - \boldsymbol{\theta}_h^k \left( w^{(j)} \right), \boldsymbol{\phi}(s,a) \right\rangle \right|$$

$$\tag{Eqn. (22)}$$

$$\leq 2L\beta \left\| \boldsymbol{\phi}(s,a) \right\|_{\left( \boldsymbol{\Lambda}_h^k \right)^{-1}}. \tag{Lemma 5}$$

---

**Algorithm 5:** Standard Lifelong-LSVI with Computation Sharing

---

1 **Set:** $Q_{H+1}^k(.,.,.) = 0, \ \forall k \in [K], \ \tilde{k} = 1$

2 **for** episodes $k = 1, \ldots, K$ **do**

3      Observe the initial state $s_1^k$ and the task context $w^k$.

4      **if** $\exists h \in [H]$ *such that* $\frac{\det \tilde{\mathbf{\Lambda}}_h^k}{\det \tilde{\mathbf{\Lambda}}_h^{\tilde{k}}} > e$ **then**

5          $\tilde{k} = k$

6          **for** time-steps $h = H, \ldots, 1$ **do**

7              Compute Compute $\tilde{\boldsymbol{\nu}}_h^{\tilde{k}}$ as in (49).

8      **for** time-steps $h = 1, \ldots, H$ **do**

9          Compute $Q_h^{\tilde{k}}(s_h^k, a, w^k)$ for all $a \in \mathcal{A}$ as in (48).

10          Play $a_h^k = \arg\max_{a \in \mathcal{A}} Q_h^{\tilde{k}}(s_h^k, a, w^k)$ and observe $s_{h+1}^k$ and $r_h^k$.

---

$\square$

## E    DETAILS OF REMARK 3

In this section, we only rely on the following two assumptions:

**Assumption 6.** *Given a feature map $\boldsymbol{\psi} : \mathcal{S} \times \mathcal{A} \times \mathcal{W} \to \mathbb{R}^{d'}$, consider function class*

$$\mathcal{F} = \left\{ f : f(s,w) = \min \left\{ \max_{a \in \mathcal{A}} \left\{ \langle \boldsymbol{\nu}, \boldsymbol{\psi}(s,a,w) \rangle + \beta \|\boldsymbol{\psi}(s,a,w)\|_{\mathbf{\Lambda}^{-1}} \right\}^+, H \right\} \boldsymbol{\nu} \in \mathbb{R}^{d'}, \beta \geq 0, \mathbf{\Lambda} \in \mathbf{S}_{++}^{d'} \right\}.$$

(46)

*Then for any $f \in \mathcal{F}$ and $h \in [H]$, there exists a vector $\boldsymbol{\nu}_h^f \in \mathbb{R}^{d'}$ with $\left\|\boldsymbol{\nu}_h^f\right\|_2 \leq H\sqrt{d'}$ such that*

$$\mathbb{P}_h \left[ f(.,w) \right] (s,a) = \langle \boldsymbol{\psi}(s,a,w), \boldsymbol{\nu}_h^f \rangle.$$

(47)

*Moreover, for every $h \in [H]$, there exists a vector $\boldsymbol{\eta}_h$ such that $r_h(s,a,w) = \langle \boldsymbol{\eta}_h, \boldsymbol{\psi}(s,a,w) \rangle$.*

**Assumption 7.** *Without loss of generality, $\left\|\boldsymbol{\psi}(s,a,w)\right\|_2 \leq 1$ for all $(s,a,w) \in \mathcal{S} \times \mathcal{A} \times \mathcal{W}$, and $\|\boldsymbol{\eta}_h\|_2 \leq \sqrt{d'}$ for all $h \in [H]$.*

### E.1    OVERVIEW

Let $\boldsymbol{\psi}_h^\tau = \boldsymbol{\psi}(s_h^\tau, a_h^\tau, w^\tau)$. Standard Lifelong-LSVI with computation sharing works with the following action-value functions:

$$Q_h^k(s,a,w) = \left\{ r_h(s,a,w) + \left\langle \tilde{\boldsymbol{\nu}}_h^k, \boldsymbol{\psi}(s,a,w) \right\rangle + \beta \|\boldsymbol{\psi}(s,a,w)\|_{(\tilde{\mathbf{\Lambda}}_h^k)^{-1}} \right\}^+,$$

(48)

where

$$\tilde{\boldsymbol{\nu}}_h^k = \left( \tilde{\mathbf{\Lambda}}_h^k \right)^{-1} \sum_{\tau=1}^{k-1} \boldsymbol{\psi}_h^\tau \cdot \min \left\{ \max_{a \in \mathcal{A}} Q_{h+1}^k(s_{h+1}^\tau, a, w^\tau), H \right\} \quad \text{and} \quad \tilde{\mathbf{\Lambda}}_h^k = \lambda \mathbf{I}_{d'} + \sum_{\tau=1}^{k-1} \boldsymbol{\psi}_h^\tau \boldsymbol{\psi}_h^{\tau\top}.$$

(49)

**Theorem 5.** *Let $T = KH$. Under Assumptions 6 and 7, the number of planning calls in 5 is at most $d'H \log \left( 1 + \frac{K}{d'\lambda} \right)$ and there exists an absolute constant $c > 0$ such that for any fixed $\delta \in (0, 0.5)$, if we set $\lambda = 1$ and $\beta = cd'H\sqrt{\log(d'T/\delta)}$ in Algorithm 5, then with probability at least $1 - 2\delta$, it holds that*

$$R_K \leq 2H\sqrt{T\log(d'T/\delta)} + 4H\beta\sqrt{2d'K\log(K)} \leq \tilde{\mathcal{O}}\left( \sqrt{d'^3 H^3 T} \right).$$

### E.2 NECESSARY ANALYSIS FOR THE PROOF OF THEOREM 5

Thanks to Assumption 6, we have

$$\mathbb{P}_h\left[V_{h+1}^k(.,w)\right](s,a) = \left\langle \boldsymbol{\nu}_h^k, \boldsymbol{\psi}(s,a,w)\right\rangle, \tag{50}$$

where $\boldsymbol{\nu}_h^k = \boldsymbol{\nu}_h^{V_{h+1}^k}$.

**Lemma 13.** *Let $c_\beta$ be a constant in the definition of $\beta$. Then, under Assumption 7, there is an absolute constant $c_0$ independent of $c_\beta$, such that for all $(h,k) \in [H] \times [K]$, with probability at least $1 - \delta$ it holds that*

$$\left\|\sum_{\tau=1}^{k-1} \boldsymbol{\psi}_h^\tau \cdot \left(V_{h+1}^k(s_{h+1}^\tau, w^\tau) - \mathbb{P}_h[V_{h+1}^k(.,w^\tau)](s_h^\tau, a_h^\tau)\right)\right\|_{\left(\tilde{\boldsymbol{\Lambda}}_h^k\right)^{-1}} \le c_0 d' H \sqrt{\log((c_\beta + 1)d'T/\delta)},$$

*where $c_0$ and $c_\beta$ are two independent absolute constants.*

*Proof.* We note that $\left\|\boldsymbol{\eta}_h + \tilde{\boldsymbol{\nu}}_h^k\right\|_2 \le (1+H)\sqrt{d'}$ and $\left\|\left(\tilde{\boldsymbol{\Lambda}}_h^k\right)^{-1}\right\| \le \frac{1}{\lambda}$. Thus, Lemmas 19 and 24 together imply that for all $(h,k) \in [H] \times [K]$, with probability at least $1 - \delta$ it holds that

$$\left\|\sum_{\tau=1}^{k-1} \boldsymbol{\phi}_h^\tau \left(V_{h+1}^k(s_{h+1}^\tau, w^\tau) - \mathbb{P}_h[V_{h+1}^k(.,w^\tau)](s_h^\tau, a_h^\tau)\right)\right\|_{\left(\tilde{\boldsymbol{\Lambda}}_h^k\right)^{-1}}^2$$

$$\le 4H^2\left(\frac{d'}{2}\log\left(\frac{k+\lambda}{\lambda}\right) + d'\log(1 + 8H\sqrt{d'}/\epsilon) + d'^2\log\left(\frac{1 + 32L^2\beta^2\sqrt{d'}}{\lambda\epsilon^2}\right) + \log\left(\frac{1}{\delta}\right)\right) + \frac{8k^2\epsilon^2}{\lambda}.$$

If we let $\epsilon = \frac{dH}{k}$ and $\beta = c_\beta(d' + \sqrt{d'})H\sqrt{\log(dT/\delta)}$, then, there exists an absolute constant $C > 0$ that is independent of $c_\beta$ such that

$$\left\|\sum_{\tau=1}^{k-1} \boldsymbol{\phi}_h^\tau \left(V_{h+1}^k(s_{h+1}^\tau, w^\tau) - \mathbb{P}_h[V_{h+1}^k(.,w^\tau)](s_h^\tau, a_h^\tau)\right)\right\|_{\left(\tilde{\boldsymbol{\Lambda}}_h^k\right)^{-1}}^2 \le C(d' + d'^2)H^2\log\left((c_\beta + 1)d'T/\delta\right).$$

$\qquad\qquad\square$

**Lemma 14.** *Under Assumptions 6 and 7, if we let $\beta = cd'H\sqrt{\lambda\log(d'T/\delta)}$ with an absolute constant $c > 0$, then the event*

$$\mathcal{E}_4 := \left\{\left\|\boldsymbol{\nu}_h^k - \tilde{\boldsymbol{\nu}}_h^k\right\|_{\tilde{\boldsymbol{\Lambda}}_h^k} \le \beta,\ \forall(h,k) \in [H] \times [K]\right\}. \tag{51}$$

*holds with probability at least $1 - \delta$.*

*Proof.*

$$\boldsymbol{\nu}_h^k - \tilde{\boldsymbol{\nu}}_h^k = \boldsymbol{\nu}_h^k - \left(\tilde{\boldsymbol{\Lambda}}_h^k\right)^{-1}\sum_{\tau=1}^{k-1}\boldsymbol{\psi}_h^\tau V_{h+1}^k(s_{h+1}^\tau, w^\tau)$$

$$= \left(\tilde{\boldsymbol{\Lambda}}_h^k\right)^{-1}\left(\tilde{\boldsymbol{\Lambda}}_h^k\boldsymbol{\nu}_h^k - \sum_{\tau=1}^{k-1}\boldsymbol{\psi}_h^\tau V_{h+1}^k(s_{h+1}^\tau, w^\tau)\right)$$

$$= \underbrace{\lambda\left(\tilde{\boldsymbol{\Lambda}}_h^k\right)^{-1}\boldsymbol{\nu}_h^k}_{\mathbf{q}_1} - \underbrace{\left(\tilde{\boldsymbol{\Lambda}}_h^k\right)^{-1}\left(\sum_{\tau=1}^{k-1}\boldsymbol{\psi}_h^\tau\left(V_{h+1}^k(s_{h+1}^\tau, w^\tau) - \mathbb{P}_h[V_{h+1}^k(.,w^\tau)](s_h^\tau, a_h^\tau)\right)\right)}_{\mathbf{q}_2}.$$

(Eqn. (50))

Thus, in order to upper bound $\left\|\boldsymbol{\nu}_h^k - \tilde{\boldsymbol{\nu}}_h^k(w)\right\|_{\tilde{\boldsymbol{\Lambda}}_h^k}$, we bound $\|\mathbf{q}_1\|_{\tilde{\boldsymbol{\Lambda}}_h^k}$ and $\|\mathbf{q}_2\|_{\tilde{\boldsymbol{\Lambda}}_h^k}$ separately.

From Assumption 7, we have

$$\|\mathbf{q}_1\|_{\boldsymbol{\Lambda}_h^k} = \lambda\left\|\boldsymbol{\nu}_h^k\right\|_{\left(\tilde{\boldsymbol{\Lambda}}_h^k\right)^{-1}} \leq \sqrt{\lambda}\left\|\boldsymbol{\nu}_h^k\right\|_2 \leq H\sqrt{\lambda d'}. \tag{52}$$

Thanks to Lemma 13, for all $(h, k) \in [H] \times [K]$, with probability at least $1 - \delta$, it holds that

$$\|\mathbf{q}_2\|_{\tilde{\boldsymbol{\Lambda}}_h^k} \leq \left\|\sum_{\tau=1}^{k-1} \boldsymbol{\psi}_h^\tau \left(V_{h+1}^k(s_{h+1}^\tau, w^\tau) - \mathbb{P}_h[V_{h+1}^k(., w^\tau)](s_h^\tau, a_h^\tau)\right)\right\|_{\left(\boldsymbol{\Lambda}_h^k\right)^{-1}} \leq c_0 d' H\sqrt{\log((c_\beta + 1)d'T/\delta)}, \tag{53}$$

where $c_0$ and $c_\beta$ are two independent absolute constants.

Combining (52) and (53), for all $(h, k) \in [H] \times [K]$, with probability at least $1 - \delta$, it holds that

$$\left\|\boldsymbol{\nu}_h^k - \tilde{\boldsymbol{\nu}}_h^k\right\|_{\tilde{\boldsymbol{\Lambda}}_h^k} \leq cd' H\sqrt{\lambda \log(d'T/\delta)}$$

for some absolute constant $c > 0$. $\qquad\square$

**Lemma 15.** *Let the setting of Lemma 14 holds. Conditioned on events $\mathcal{E}_4$ defined in (51), and with $Q_h^k$ computed as in (48), it holds that $Q_h^k(s, a, w) \geq Q_h^*(s, a, w)$ for all $(s, a, w, h, k) \in \mathcal{S} \times \mathcal{A} \times \mathcal{W} \times [H] \times [K]$.*

*Proof.* We first note that conditioned on the event $\mathcal{E}_4$, for all $(s, a, w, h, k) \in \mathcal{S} \times \mathcal{A} \times \mathcal{W} \times [H] \times [K]$, it holds that

$$\left|r_h(s, a, w) + \left\langle \tilde{\boldsymbol{\nu}}_h^k, \boldsymbol{\psi}(s, a, w)\right\rangle - Q_h^\pi(s, a, w) - \mathbb{P}_h\left[V_{h+1}^k(., w) - V_{h+1}^\pi(., w)\right](s, a)\right|$$

$$= \left|r_h(s, a, w) + \left\langle \tilde{\boldsymbol{\nu}}_h^k, \boldsymbol{\psi}(s, a, w)\right\rangle - r_h(s, a, w) - \mathbb{P}_h\left[V_{h+1}^k(., w)\right](s, a)\right|$$

$$= \left|\left\langle \tilde{\boldsymbol{\nu}}_h^k, \boldsymbol{\psi}(s, a, w)\right\rangle - \mathbb{P}_h\left[V_{h+1}^k(., w)\right](s, a)\right|$$

$$= \left|\left\langle \tilde{\boldsymbol{\nu}}_h^k - \boldsymbol{\nu}_h^k, \boldsymbol{\psi}(s, a, w)\right\rangle\right|$$

$$\leq \left\|\tilde{\boldsymbol{\nu}}_h^k - \boldsymbol{\nu}_h^k\right\|_{\tilde{\boldsymbol{\Lambda}}_h^k} \left\|\boldsymbol{\psi}(s, a, w)\right\|_{\left(\tilde{\boldsymbol{\Lambda}}_h^k\right)^{-1}}$$

$$\leq \beta\left\|\boldsymbol{\psi}(s, a, w)\right\|_{\left(\tilde{\boldsymbol{\Lambda}}_h^k\right)^{-1}}, \tag{Lemma 14}$$

for any policy $\pi$.

Now, we prove the lemma by induction. The statement holds for $H$ because $Q_{H+1}^k(., ., .) = Q_{H+1}^*(., ., .) = 0$ and thus conditioned on the event $\mathcal{E}_4$, defined in (51), for all $(s, a, w, k) \in \mathcal{S} \times \mathcal{A} \times \mathcal{W} \times [K]$, we have

$$\left|r_h(s, a, w) + \left\langle \boldsymbol{\nu}_H^k, \boldsymbol{\psi}(s, a, w)\right\rangle - Q_H^*(s, a, w)\right| \leq \beta\left\|\boldsymbol{\psi}(s, a, w)\right\|_{\left(\tilde{\boldsymbol{\Lambda}}_H^k\right)^{-1}}.$$

Therefore, conditioned on the event $\mathcal{E}_4$, for all $(s, a, w, k) \in \mathcal{S} \times \mathcal{A} \times \mathcal{W} \times [K]$, we have

$$Q_H^*(s, a, w) \leq r_H(s, a, w) + \left\langle \boldsymbol{\nu}_H^k, \boldsymbol{\psi}(s, a, w)\right\rangle + \beta\left\|\boldsymbol{\psi}(s, a, w)\right\|_{\left(\tilde{\boldsymbol{\Lambda}}_H^k\right)^{-1}}$$

$$= \left\{r_H(s, a, w) + \left\langle \boldsymbol{\nu}_H^k, \boldsymbol{\psi}(s, a, w)\right\rangle + \beta\left\|\boldsymbol{\psi}(s, a, w)\right\|_{\left(\tilde{\boldsymbol{\Lambda}}_H^k\right)^{-1}}\right\}^+$$

$$= Q_H^k(s, a, w),$$

where the first equality follows from the fact that $Q_H^*(s, a, w) \geq 0$. Now, suppose the statement holds at time-step $h + 1$ and consider time-step $h$. Conditioned on events $\mathcal{E}_4$, for all $(s, a, w, h, k) \in \mathcal{S} \times \mathcal{A} \times \mathcal{W} \times [H] \times [K]$, we have

$$0 \leq r_h(s, a, w) + \left\langle \boldsymbol{\nu}_h^k, \boldsymbol{\psi}(s, a, w) \right\rangle - Q_h^*(s, a, w) - \mathbb{P}_h \left[ V_{h+1}^k(., w) - V_{h+1}^*(., w) \right](s, a) + \beta \left\| \boldsymbol{\psi}(s, a, w) \right\|_{\left( \tilde{\boldsymbol{\Lambda}}_h^k \right)^{-1}}$$

$$\leq r_h(s, a, w) + \left\langle \boldsymbol{\nu}_h^k, \boldsymbol{\psi}(s, a, w) \right\rangle - Q_h^*(s, a, w) + \beta \left\| \boldsymbol{\psi}(s, a, w) \right\|_{\left( \tilde{\boldsymbol{\Lambda}}_h^k \right)^{-1}}.$$

(Induction assumption)

Therefore, conditioned on events $\mathcal{E}_4$, for all $(s, a, w, h, k) \in \mathcal{S} \times \mathcal{A} \times \mathcal{W} \times [H] \times [K]$, we have

$$Q_h^*(s, a, w) \leq r_h(s, a, w) + \left\langle \boldsymbol{\nu}_h^k, \boldsymbol{\psi}(s, a, w) \right\rangle + \beta \left\| \boldsymbol{\psi}(s, a, w) \right\|_{\left( \tilde{\boldsymbol{\Lambda}}_h^k \right)^{-1}}$$

$$= \left\{ r_h(s, a, w) + \left\langle \boldsymbol{\nu}_h^k, \boldsymbol{\psi}(s, a, w) \right\rangle + \beta \left\| \boldsymbol{\psi}(s, a, w) \right\|_{\left( \tilde{\boldsymbol{\Lambda}}_h^k \right)^{-1}} \right\}^+$$

$$= Q_h^k(s, a, w),$$

where the first equality follows from the fact that $Q_H^*(s, a, w) \geq 0$. This completes the proof.

$\square$

### E.3  PROOF OF THEOREM 5

First, we bound the number of times Algorithm 5 updates $\tilde{\boldsymbol{\nu}}_h^k$. Let $P$ be the total number of updates and $k_p$ be the episode at which, the agent did replanning for the $p$-th time. Note that $\det \tilde{\boldsymbol{\Lambda}}_h^1 = \lambda^{d'}$ and $\det \tilde{\boldsymbol{\Lambda}}_h^K \leq \operatorname{trace}(\tilde{\boldsymbol{\Lambda}}_h^K / d')^{d'} \leq \left( \lambda + \frac{K}{d'} \right)^{d'}$, and consequently:

$$\frac{\det \tilde{\boldsymbol{\Lambda}}_h^K}{\det \tilde{\boldsymbol{\Lambda}}_h^1} = \prod_{p=1}^P \frac{\det \tilde{\boldsymbol{\Lambda}}_h^{k_p}}{\det \tilde{\boldsymbol{\Lambda}}_h^{k_{p-1}}} \leq \left( 1 + \frac{K}{d'\lambda} \right)^{d'},$$

and therefore

$$\prod_{h=1}^H \frac{\det \tilde{\boldsymbol{\Lambda}}_h^K}{\det \tilde{\boldsymbol{\Lambda}}_h^1} = \prod_{h=1}^H \prod_{p=1}^P \frac{\det \tilde{\boldsymbol{\Lambda}}_h^{k_p}}{\det \tilde{\boldsymbol{\Lambda}}_h^{k_{p-1}}} \leq \left( 1 + \frac{K}{d'\lambda} \right)^{d'H}. \tag{54}$$

Since $1 \leq \frac{\det \tilde{\boldsymbol{\Lambda}}_h^{k_p}}{\det \tilde{\boldsymbol{\Lambda}}_h^{k_{p-1}}}$ for all $p \in [P]$, we can deduce from (54) that

$$\exists h \in [H] \quad \text{such that} \quad e < \frac{\det \tilde{\boldsymbol{\Lambda}}_h^k}{\det \tilde{\boldsymbol{\Lambda}}_h^{\tilde{k}}}$$

happens for at most $d'H \log \left( 1 + \frac{K}{d'\lambda} \right)$ number of episodes $k \in [K]$. This concludes that number of planning calls in Algorithm 5 is at most $d'H \log \left( 1 + \frac{K}{d'\lambda} \right)$.

Now, we prove the regret bound. Let $\delta_h^k = V_h^{\tilde{k}}(s_h^k, w^k) - V_h^{\pi^k}(s_h^k, w^k)$ and $\xi_{h+1}^k = \mathbb{E} \left[ \delta_{h+1}^k | s_h^k, a_h^k \right] - \delta_{h+1}^k$. Conditioned on $\mathcal{E}_4$, for all $(s, a, w, h, k) \in \mathcal{S} \times \mathcal{A} \times \mathcal{W} \times [H] \times [K]$, we have

$$Q_h^{\tilde{k}}(s, a, w) - Q_h^{\pi^k}(s, a, w) = r_h(s, a, w) + \left\langle \boldsymbol{\theta}_h^{\tilde{k}}, \boldsymbol{\psi}(s, a, w) \right\rangle - Q_h^{\pi^k}(s, a, w) + \beta \left\| \boldsymbol{\psi}(s, a, w) \right\|_{\left( \tilde{\boldsymbol{\Lambda}}_h^{\tilde{k}} \right)^{-1}}$$

$$\leq \mathbb{P}_h \left[ V_{h+1}^{\tilde{k}}(., w) - V_{h+1}^{\pi^k}(., w) \right](s, a) + 2\beta \left\| \boldsymbol{\psi}(s, a, w) \right\|_{\left( \tilde{\boldsymbol{\Lambda}}_h^v \right)^{-1}}. \tag{55}$$

Note that $\delta_h^{\tilde{k}} \leq Q_h^k(s_h^k, a_h^k, w^k) - Q_h^{\pi^k}(s_h^k, a_h^k, w^k)$. Thus, (55) and Lemma 14 imply that for all $(h, k) \in [H] \times [K]$, it holds that

$$\delta_h^k \leq \xi_{h+1}^k + \delta_{h+1}^k + 2\beta \left\| \boldsymbol{\psi}(s_h^k, a_h^k, w^k) \right\|_{(\tilde{\boldsymbol{\Lambda}}_h^k)^{-1}}.$$

Now, we complete the regret analysis following similar steps as those of Theorem 1's proof:

$$
\begin{aligned}
R_K &= \sum_{k=1}^{K} V_1^*(s_1^k, w^k) - V_1^{\pi^k}(s_1^k, w^k) \\
&\leq \sum_{k=1}^{K} V_1^{\tilde{k}}(s_1^k, w^k) - V_1^{\pi^k}(s_1^k, w^k) && \text{(Lemma 15)} \\
&= \sum_{k=1}^{K} \delta_1^k \\
&\leq \sum_{k=1}^{K} \sum_{h=1}^{H} \xi_h^k + 2\beta \sum_{k=1}^{K} \sum_{h=1}^{H} \left\| \boldsymbol{\psi}(s_h^k, a_h^k, w^k) \right\|_{\left(\tilde{\boldsymbol{\Lambda}}_h^{\tilde{k}}\right)^{-1}} \\
&\leq \sum_{k=1}^{K} \sum_{h=1}^{H} \xi_h^k + 2\beta \sum_{k=1}^{K} \sum_{h=1}^{H} \left\| \boldsymbol{\psi}(s_h^k, a_h^k, w^k) \right\|_{\left(\tilde{\boldsymbol{\Lambda}}_h^k\right)^{-1}} \sqrt{\frac{\det \tilde{\boldsymbol{\Lambda}}_h^k}{\det \tilde{\boldsymbol{\Lambda}}_h^{\tilde{k}}}} && \text{(Eqn. (27))} \\
&\leq 2H\sqrt{T \log(d'T/\delta)} + 4H\beta\sqrt{2\lambda d' K \log(1 + K/\lambda)} \\
&\leq \tilde{\mathcal{O}}\left( \sqrt{\lambda d'^3 H^3 T} \right).
\end{aligned}
$$

## F   DETAILS OF REMARK 4: A MISSPECIFIED SETTING

We first present a definition for an approximate completeness model.

**Assumption 8** ($\zeta$-Approximate Completeness). *Given feature maps $\phi : \mathcal{S} \times \mathcal{A} \to \mathbb{R}^d$ and $\psi : \mathcal{S} \times \mathcal{A} \times \mathcal{W} \to \mathbb{R}^{d'}$ in Assumption 1, consider the function class*

$$\mathcal{F} = \left\{ f : f(s, w) = \min\left\{ \max_{a \in \mathcal{A}} \left\{ \langle \boldsymbol{\nu}, \boldsymbol{\psi}(s, a, w) \rangle + \beta \|\phi(s, a)\|_{\boldsymbol{\Lambda}^{-1}} \right\}^+, H \right\}, \boldsymbol{\nu} \in \mathbb{R}^{d'}, \boldsymbol{\Lambda} \in \mathbf{S}_{++}^d, \beta \geq 0 \right\}.$$

*For any $f \in \mathcal{F}$ and $h \in [H]$, there exists a vector $\boldsymbol{\xi}_h^f \in \mathbb{R}^{d'}$ with $\left\| \boldsymbol{\xi}_h^f \right\| \leq H\sqrt{d'}$ such that for all $(s, a, w) \in \mathcal{S} \times \mathcal{A} \times \mathcal{W}$*

$$\left| \mathbb{P}_h \left[ f(., w) \right](s, a) - \langle \boldsymbol{\xi}_h^f, \boldsymbol{\psi}(s, a, w) \rangle \right| \leq \zeta.$$

**Theorem 6.** *Let $T = KH$. Under Assumptions 1, 8, and 3, the number of planning calls in Algorithm 2 is at most $dH \log(1 + \frac{K}{d\lambda})$, and there exists an absolute constant $c > 0$ such that for any fixed $\delta \in (0, 0.5)$, if we set $\lambda = 1$ and $\beta = cH(d + \sqrt{md})\sqrt{\log(mdT/\delta)}$ in Algorithm 2, then with probability at least $1 - 2\delta$, it holds that*

$$R_K \leq \tilde{\mathcal{O}}\left( \sqrt{md}T\zeta + \sqrt{(d^3 + md^2)H^3 T} \right).$$

### F.1   NECESSARY ANALYSIS FOR THE PROOF OF THEOREM 6

Let $\left( s^{(i)}, a^{(i)} \right)$ be the $i$-th element of $\mathcal{D}$ and $\{c_i'(s, a)\}_{i \in [d]}$ be the coefficients such that

$$\phi(s, a) = \sum_{i \in [d]} c_i'(s, a) \phi\left( s^{(i)}, a^{(i)} \right).$$

Then, $L_\phi$ is a positive constant such that $\sum_{i \in [d]} |c_i'(s, a)| \leq L_\phi$ for all $(s, a) \in \mathcal{S} \times \mathcal{A}$.

**Lemma 16.** *Let $\widetilde{\mathcal{W}} = \{w^\tau : \tau \in [K]\} \cup \{w^{(j)} : j \in [n]\}$. Under the setting of Theorem 6 and conditioned on events $\{\mathcal{E}_1(w)\}_{w \in \widetilde{\mathcal{W}}}$ defined in (9), for all $(s, a, w, h, k) \in \mathcal{S} \times \mathcal{A} \times \widetilde{\mathcal{W}} \times [H] \times [K]$, it holds that*

$$\left| \langle \hat{\boldsymbol{\xi}}_h^k, \boldsymbol{\psi}(s, a, w) \rangle - \mathbb{P}_h[V_{h+1}^k(., w)](s, a) \right| \leq (2L + L_\phi \sqrt{md}) \zeta + 2L\beta \|\boldsymbol{\phi}(s, a)\|_{(\boldsymbol{\Lambda}_h^k)^{-1}}.$$

*Proof.* Thanks to Assumption 8 and conditioned on events $\{\mathcal{E}_1(w)\}_{w \in \widetilde{\mathcal{W}}}$, one set of feasible parameters for (8) is $\left\{ \boldsymbol{\theta}_h^k \left( w^{(j)} \right) \right\}_{j \in [n]}$ and $\boldsymbol{\xi}_h^{V_{h+1}^k}$ such that

$$\left| \left\langle \hat{\boldsymbol{\theta}}_h^{k(j)}, \boldsymbol{\phi}(s, a) \right\rangle - \left\langle \hat{\boldsymbol{\xi}}_h^k, \boldsymbol{\psi}\left(s, a, w^{(j)}\right) \right\rangle \right| \leq \zeta \sqrt{md}, \quad \forall (j, (s, a)) \in [n] \times \mathcal{D}. \tag{56}$$

For any triple $(s, a, j) \in \mathcal{S} \times \mathcal{A} \times [n]$, we have

$$\left\langle \hat{\boldsymbol{\xi}}_h^k, \boldsymbol{\psi}\left(s, a, w^{(j)}\right) \right\rangle = \left\langle \hat{\boldsymbol{\xi}}_h^k, \boldsymbol{\phi}(s, a) \otimes \boldsymbol{\rho}\left(w^{(j)}\right) \right\rangle$$

$$= \left\langle \hat{\boldsymbol{\xi}}_h^k, \sum_{i \in [d]} c_i'(s, a) \boldsymbol{\phi}\left(s^{(i)}, a^{(i)}\right) \otimes \boldsymbol{\rho}\left(w^{(j)}\right) \right\rangle$$

$$= \sum_{i \in [d]} c_i'(s, a) \left\langle \hat{\boldsymbol{\xi}}_h^k, \boldsymbol{\psi}\left(s^{(i)}, a^{(i)}, w^{(j)}\right) \right\rangle \qquad \text{(Assumption 3)}$$

$$\leq \sqrt{md}\zeta \sum_{i \in [d]} c_i'(s, a) + \sum_{i \in [d]} c_i'(s, a) \left\langle \hat{\boldsymbol{\theta}}_h^{k(j)}, \boldsymbol{\phi}\left(s^{(i)}, a^{(i)}\right) \right\rangle \qquad \text{(Eqn. (56))}$$

$$\leq L_\phi \sqrt{md}\zeta + \left\langle \hat{\boldsymbol{\theta}}_h^{k(j)}, \boldsymbol{\phi}(s, a) \right\rangle.$$

Similarly, it holds that $\left\langle \hat{\boldsymbol{\xi}}_h^k, \boldsymbol{\psi}\left(s, a, w^{(j)}\right) \right\rangle \geq -L_\phi \sqrt{md}\zeta + \left\langle \hat{\boldsymbol{\theta}}_h^{k(j)}, \boldsymbol{\phi}(s, a) \right\rangle$. Therefore, for any $(s, a, j) \in \mathcal{S} \times \mathcal{A} \times [n]$, it holds that

$$\left| \left\langle \hat{\boldsymbol{\xi}}_h^k, \boldsymbol{\psi}\left(s, a, w^{(j)}\right) \right\rangle - \left\langle \hat{\boldsymbol{\theta}}_h^{k(j)}, \boldsymbol{\phi}(s, a) \right\rangle \right| \leq L_\phi \sqrt{md}\zeta. \tag{57}$$

For any $(s, a, w) \in \mathcal{S} \times \mathcal{A} \times \mathcal{W}$, it holds that

$$\mathbb{P}_h \left[ V_{h+1}^k(., w) \right](s, a) = \left\langle \boldsymbol{\theta}_h^k(w), \boldsymbol{\phi}(s, a) \right\rangle \qquad \text{(Eqn. (4))}$$

$$\leq \zeta + \left\langle \boldsymbol{\xi}_h^{V_{h+1}^k}, \boldsymbol{\psi}(s, a, w) \right\rangle \qquad \text{(Assumption 8)}$$

$$= \zeta + \sum_{j \in [n]} c_j(w) \left\langle \boldsymbol{\xi}_h^{V_{h+1}^k}, \boldsymbol{\psi}\left(s, a, w^{(j)}\right) \right\rangle \qquad \text{(Assumption 3)}$$

$$\leq \zeta \left( 1 + \sum_{j \in [n]} c_j(w) \right) + \sum_{j \in [n]} c_j(w) \mathbb{P}_h \left[ V_{h+1}^k\left(., w^{(j)}\right) \right](s, a)$$

$$\qquad \qquad \text{(Assumption 8)}$$

$$\leq 2L\zeta + \sum_{j \in [n]} c_j(w) \left\langle \boldsymbol{\theta}_h^k\left(w^{(j)}\right), \boldsymbol{\phi}(s, a) \right\rangle. \qquad \text{(Assumption 3)}$$

Similarly, it holds that $\mathbb{P}_h \left[ V_{h+1}^k(., w) \right](s, a) \geq -2L\zeta + \sum_{j \in [n]} c_j(w) \left\langle \boldsymbol{\theta}_h^k\left(w^{(j)}\right), \boldsymbol{\phi}(s, a) \right\rangle$. Therefore, for any $(s, a, w) \in \mathcal{S} \times \mathcal{A} \times \mathcal{W}$, it holds that

$$\left| \mathbb{P}_h \left[ V_{h+1}^k(.,w) \right] (s,a) - \sum_{j\in[n]} c_j(w) \left\langle \boldsymbol{\theta}_h^k \left( w^{(j)} \right), \boldsymbol{\phi}(s,a) \right\rangle \right| \leq 2L\zeta. \tag{58}$$

Finally, conditioned on events $\{\mathcal{E}_1(w)\}_{w\in\widetilde{\mathcal{W}}}$, for all $(s,a,w,h,k) \in \mathcal{S}\times\mathcal{A}\times\widetilde{\mathcal{W}}\times[H]\times[K]$, it holds that

$$\left| \left\langle \hat{\boldsymbol{\xi}}_h^k, \boldsymbol{\psi}(s,a,w) \right\rangle - \mathbb{P}_h \left[ V_{h+1}^k(.,w) \right] (s,a) \right|$$

$$\leq 2L\zeta + \left| \sum_{j\in[n]} c_j(w) \left( \left\langle \hat{\boldsymbol{\xi}}_h^k, \boldsymbol{\psi}\left(s,a,w^{(j)}\right) \right\rangle - \left\langle \boldsymbol{\theta}_h^k \left( w^{(j)} \right), \boldsymbol{\phi}(s,a) \right\rangle \right) \right|$$

$$\text{(Assumption 3 and Eqn. (58))}$$

$$\leq 2L\zeta + \left| \sum_{j\in[n]} c_j(w) \left( \left\langle \hat{\boldsymbol{\xi}}_h^k, \boldsymbol{\psi}\left(s,a,w^{(j)}\right) \right\rangle - \left\langle \hat{\boldsymbol{\theta}}_h^{k(j)}, \boldsymbol{\phi}(s,a) \right\rangle \right) \right|$$

$$+ \left| \sum_{j\in[n]} c_j(w) \left\langle \hat{\boldsymbol{\theta}}_h^{k(j)} - \tilde{\boldsymbol{\theta}}_h^k \left( w^{(j)} \right), \boldsymbol{\phi}(s,a) \right\rangle \right| + \left| \sum_{j\in[n]} c_j(w) \left\langle \tilde{\boldsymbol{\theta}}_h^k \left( w^{(j)} \right) - \boldsymbol{\theta}_h^k \left( w^{(j)} \right), \boldsymbol{\phi}(s,a) \right\rangle \right|$$

$$\leq (2L + L_{\boldsymbol{\phi}}\sqrt{md})\zeta + \left| \sum_{j\in[n]} c_j(w) \left\langle \hat{\boldsymbol{\theta}}_h^{k(j)} - \tilde{\boldsymbol{\theta}}_h^k \left( w^{(j)} \right), \boldsymbol{\phi}(s,a) \right\rangle \right| + \left| \sum_{j\in[n]} c_j(w) \left\langle \tilde{\boldsymbol{\theta}}_h^k \left( w^{(j)} \right) - \boldsymbol{\theta}_h^k \left( w^{(j)} \right), \boldsymbol{\phi}(s,a) \right\rangle \right|$$

$$\text{(Eqn. (57))}$$

$$\leq (2L + L_{\boldsymbol{\phi}}\sqrt{md})\zeta + 2L\beta \left\| \boldsymbol{\phi}(s,a) \right\|_{(\boldsymbol{\Lambda}_h^k)^{-1}}. \tag{Lemma 5}$$

$\square$

As the final step in the regret analysis, we state the following lemma which uses Lemma 16 to prove the optimistic nature of UCBlvd. Then following the standard analysis of single-task LSVI-UCB we derive the regret bound for misspecified settings.

**Lemma 17.** *Let* $\widetilde{\mathcal{W}} = \{w^\tau : \tau \in [K]\} \cup \{w^{(j)} : j \in [n]\}$. *Under the setting of Theorem 6 and conditioned on events* $\{\mathcal{E}_1(w)\}_{w\in\widetilde{\mathcal{W}}}$ *defined in* (9)*, and with* $Q_h^k$ *computed as in* (7)*, it holds that* $(2L + L_{\boldsymbol{\phi}}\sqrt{md})(H - h + 1)\zeta + Q_h^k(s,a,w) \geq Q_h^*(s,a,w)$ *for all* $(s,a,w,h,k) \in \mathcal{S}\times\mathcal{A}\times\widetilde{\mathcal{W}}\times[H]\times[K]$.

*Proof.* We first note that conditioned on events $\{\mathcal{E}_1(w)\}_{w\in\widetilde{\mathcal{W}}}$, for all $(s,a,w,h,k) \in \mathcal{S}\times\mathcal{A}\times\widetilde{\mathcal{W}}\times[H]\times[K]$, it holds that

$$\left| r_h(s,a,w) + \left\langle \hat{\boldsymbol{\xi}}_h^k, \boldsymbol{\psi}(s,a,w) \right\rangle - Q_h^\pi(s,a,w) - \mathbb{P}_h \left[ V_{h+1}^k(.,w) - V_{h+1}^\pi(.,w) \right] (s,a) \right|$$

$$= \left| r_h(s,a,w) + \left\langle \hat{\boldsymbol{\xi}}_h^k, \boldsymbol{\psi}(s,a,w) \right\rangle - r_h(s,a,w) - \mathbb{P}_h \left[ V_{h+1}^k(.,w) \right] (s,a) \right|$$

$$= \left| \left\langle \hat{\boldsymbol{\xi}}_h^k, \boldsymbol{\psi}(s,a,w) \right\rangle - \mathbb{P}_h \left[ V_{h+1}^k(.,w) \right] (s,a) \right|$$

$$\leq (2L + L_{\boldsymbol{\phi}}\sqrt{md})\zeta + 2L\beta \left\| \boldsymbol{\phi}(s,a) \right\|_{(\boldsymbol{\Lambda}_h^k)^{-1}}, \tag{Lemma 16}$$

for any policy $\pi$.

Now, we prove the lemma by induction. The statement holds for $H$ because $Q_{H+1}^k(.,.,.) = Q_{H+1}^*(.,.,.) = 0$ and thus conditioned events $\{\mathcal{E}_1(w)\}_{w \in \widetilde{\mathcal{W}}}$, defined in (9), for all $(s, a, w, k) \in \mathcal{S} \times \mathcal{A} \times \widetilde{\mathcal{W}} \times [K]$, we have

$$\left| r_H(s, a, w) + \left\langle \hat{\boldsymbol{\xi}}_H^k, \boldsymbol{\psi}(s, a, w) \right\rangle - Q_H^*(s, a, w) \right| \leq (2L + L_\phi \sqrt{md})\zeta + 2L\beta \|\boldsymbol{\phi}(s, a)\|_{(\boldsymbol{\Lambda}_H^k)^{-1}}.$$

Therefore, conditioned on events $\{\mathcal{E}_1(w)\}_{w \in \widetilde{\mathcal{W}}}$, for all $(s, a, w, k) \in \mathcal{S} \times \mathcal{A} \times \widetilde{\mathcal{W}} \times [K]$, we have

$$Q_H^*(s, a, w) \leq r_H(s, a, w) + \left\langle \hat{\boldsymbol{\xi}}_H^k, \boldsymbol{\psi}(s, a, w) \right\rangle + 2L\beta \|\boldsymbol{\phi}(s, a)\|_{(\boldsymbol{\Lambda}_H^k)^{-1}} + (2L + L_\phi \sqrt{md})\zeta$$

$$= \left\{ r_H(s, a, w) + \left\langle \hat{\boldsymbol{\xi}}_H^k, \boldsymbol{\psi}(s, a, w) \right\rangle + 2L\beta \|\boldsymbol{\phi}(s, a)\|_{(\boldsymbol{\Lambda}_H^k)^{-1}} \right\}^+ + (2L + L_\phi \sqrt{md})\zeta$$

$$= Q_H^k(s, a, w) + (2L + L_\phi \sqrt{md})\zeta,$$

where the first equality follows from the fact that $Q_H^*(s, a, w) \geq 0$. Now, suppose the statement holds at time-step $h + 1$ and consider time-step $h$. Conditioned on events $\{\mathcal{E}_1(w)\}_{w \in \widetilde{\mathcal{W}}}$, for all $(s, a, w, h, k) \in \mathcal{S} \times \mathcal{A} \times \widetilde{\mathcal{W}} \times [H] \times [K]$, we have

$$0 \leq r_h(s, a, w) + \left\langle \hat{\boldsymbol{\xi}}_h^k, \boldsymbol{\psi}(s, a, w) \right\rangle - Q_h^*(s, a, w) - \mathbb{P}_h \left[ V_{h+1}^k(., w) - V_{h+1}^*(., w) \right] (s, a)$$

$$+ (2L + L_\phi \sqrt{md})\zeta + 2L\beta \|\boldsymbol{\phi}(s, a)\|_{(\boldsymbol{\Lambda}_h^k)^{-1}}$$

$$\leq r_h(s, a, w) + \left\langle \hat{\boldsymbol{\xi}}_h^k, \boldsymbol{\psi}(s, a, w) \right\rangle - Q_h^*(s, a, w) + (2L + L_\phi \sqrt{md})(H - h + 1)\zeta + 2L\beta \|\boldsymbol{\phi}(s, a)\|_{(\boldsymbol{\Lambda}_h^k)^{-1}}.$$

$$\text{(Induction assumption)}$$

Therefore, conditioned on events $\{\mathcal{E}_1(w)\}_{w \in \widetilde{\mathcal{W}}}$, for all $(s, a, w, h, k) \in \mathcal{S} \times \mathcal{A} \times \widetilde{\mathcal{W}} \times [H] \times [K]$, we have

$$Q_h^*(s, a, w) \leq r_h(s, a, w) + \left\langle \hat{\boldsymbol{\xi}}_h^k, \boldsymbol{\psi}(s, a, w) \right\rangle + (2L + L_\phi \sqrt{md})(H - h + 1)\zeta + 2L\beta \|\boldsymbol{\phi}(s, a)\|_{(\boldsymbol{\Lambda}_h^k)^{-1}}$$

$$= \left\{ r_h(s, a, w) + \left\langle \hat{\boldsymbol{\xi}}_h^k, \boldsymbol{\psi}(s, a, w) \right\rangle + 2L\beta \|\boldsymbol{\phi}(s, a)\|_{(\boldsymbol{\Lambda}_h^k)^{-1}} \right\}^+ + (2L + L_\phi \sqrt{md})(H - h + 1)\zeta$$

$$= Q_h^k(s, a, w) + (2L + L_\phi \sqrt{md})(H - h + 1)\zeta,$$

where the first equality follows from the fact that $Q_h^*(s, a, w) \geq 0$. This completes the proof.

$\square$

### F.2 PROOF OF THEOREM 6

The proof for establishing the upper bound on the number of planning calls for misspecified settings follows exactly the steps as those in the proof of Theorem 2.

Now, we prove the regret bound. Let $\delta_h^k = V_h^{\tilde{k}}(s_h^k, w^k) - V_h^{\pi^k}(s_h^k, w^k)$ and $\xi_{h+1}^k = \mathbb{E}\left[ \delta_{h+1}^k | s_h^k, a_h^k \right] - \delta_{h+1}^k$. Conditioned on events $\{\mathcal{E}_1(w)\}_{w \in \widetilde{\mathcal{W}}}$, for all $(s, a, w, h, k) \in \mathcal{S} \times \mathcal{A} \times \widetilde{\mathcal{W}} \times [H] \times [K]$, we have

$$Q_h^{\tilde{k}}(s, a, w) - Q_h^{\pi^k}(s, a, w) = r_h(s, a, w) + \left\langle \hat{\boldsymbol{\xi}}_h^{\tilde{k}}, \boldsymbol{\psi}(s, a, w) \right\rangle - Q_h^{\pi^k}(s, a, w) + 2L\beta \|\boldsymbol{\phi}(s, a)\|_{(\boldsymbol{\Lambda}_h^{\tilde{k}})^{-1}}$$

$$\leq \mathbb{P}_h \left[ V_{h+1}^{\tilde{k}}(., w) - V_{h+1}^{\pi^k}(., w) \right] (s, a) + (2L + L_\phi \sqrt{md})\zeta + 4L\beta \|\boldsymbol{\phi}(s, a)\|_{(\boldsymbol{\Lambda}_h^{\tilde{k}})^{-1}}.$$

$$(59)$$

Note that $\delta_h^k \leq Q_h^{\tilde{k}}(s_h^k, a_h^k, w^k) - Q_h^{\pi^k}(s_h^k, a_h^k, w^k)$. Thus, combining (59), Lemma 5, and a union bound over $\widetilde{\mathcal{W}}$, we conclude that for all $(h, k) \in [H] \times [K]$, with probability at least $1 - \delta$, it holds that gives

$$\delta_h^k \leq \xi_{h+1}^k + \delta_{h+1}^k + (2L + L_\phi \sqrt{md})\zeta + 4L\beta \left\| \phi(s_h^k, a_h^k) \right\|_{(\Lambda_h^{\tilde{k}})^{-1}}.$$

Now, we complete the regret analysis following similar steps as those of Theorem 1's proof:

$$
\begin{aligned}
R_K &= \sum_{k=1}^K V_1^*(s_1^k, w^k) - V_1^{\pi^k}(s_1^k, w^k) \\
&\leq (2L + L_\phi \sqrt{md})HK\zeta + \sum_{k=1}^K V_1^{\tilde{k}}(s_1^k, w^k) - V_1^{\pi^k}(s_1^k, w^k) \quad \text{(Lemma 17)} \\
&= (2L + L_\phi \sqrt{md})HK\zeta + \sum_{k=1}^K \delta_1^k \\
&\leq (4L + 2L_\phi \sqrt{md})HK\zeta + \sum_{k=1}^K \sum_{h=1}^H \xi_h^k + 4L\beta \sum_{k=1}^K \sum_{h=1}^H \left\| \phi(s_h^k, a_h^k) \right\|_{\left(\Lambda_h^{\tilde{k}}\right)^{-1}} \\
&\leq (4L + 2L_\phi \sqrt{md})HK\zeta + \sum_{k=1}^K \sum_{h=1}^H \xi_h^k + 4L\beta \sum_{k=1}^K \sum_{h=1}^H \left\| \phi(s_h^k, a_h^k) \right\|_{(\Lambda_h^k)^{-1}} \sqrt{\frac{\det \Lambda_h^k}{\det \Lambda_h^{\tilde{k}}}} \\
&\qquad\qquad\qquad\qquad\qquad\qquad\qquad\qquad\qquad\qquad\qquad\qquad\qquad\qquad\text{(Eqn. (27))} \\
&\leq (4L + 2L_\phi \sqrt{md})HK\zeta + 2H\sqrt{T \log(dT/\delta)} + 8HL\beta \sqrt{2dK \log(1 + K/\lambda)} \\
&\leq \tilde{\mathcal{O}}\left( (L + L_\phi \sqrt{md})HK\zeta + L\sqrt{\lambda(d^3 + md^2)H^3T} \right),
\end{aligned}
$$

where the last two inequalities follow from the similar steps in the proof of Theorem 1.

## G  AUXILIARY LEMMAS

**Notations.**  $\mathcal{N}_\epsilon(\mathcal{V})$ denotes the $\epsilon$-covering number of the class $\mathcal{V}$ of functions mapping $\mathcal{S}$ to $\mathbb{R}$ with respect to the distance $\text{dist}(V, V') = \sup_s |V(s) - V'(s)|$.

**Lemma 18** (Bound on Weights $\theta_h^k(w)$). *Under Assumption 1, for any set of action-value functions $\{Q_h^k\}_{h \in [H]}$, and $(w, h, k) \in \mathcal{W} \times [H] \times [K]$, it holds that*

$$\left\| \theta_h^k(w) \right\|_2 \leq H\sqrt{d}.$$

*Proof.* Recall that $V_h^k(s, w) = \min\{\max_{a \in \mathcal{A}} Q_h^k(s, a, w), H\}$ and $\theta_h^k(w) := \int_{\mathcal{S}} V_{h+1}^k(s', w)d\mu_h(s')$. Thus, we have

$$\left\| \theta_h^k(w) \right\|_2 = \left\| \int_{\mathcal{S}} V_{h+1}^k(s', w)d\mu_h(s') \right\| \leq H\sqrt{d}.$$

$\square$

**Lemma 19** (Lemma D.4 in Jin et al. (2020)). *Let $\{s_\tau\}_{\tau=1}^\infty$ be a stochastic process on state space $\mathcal{S}$ with corresponding filtration $\{\mathcal{F}_\tau\}_{\tau=0}^\infty$. Let $\{\phi_\tau\}_{\tau=0}^\infty$ be an $\mathbb{R}^d$-valued stochastic process where $\phi_\tau \in \mathcal{F}_{\tau-1}$, and $\|\phi_\tau\| \leq 1$. Let $\Lambda_k = \lambda \mathbf{I}_d + \sum_{\tau=1}^{k-1} \phi_\tau \phi_\tau^\top$. Then with probability at least $1 - \delta$, for all $k \geq 0$ and $V \in \mathcal{V}$ such that $\sup_{s \in \mathcal{S}} |V(s)| \leq H$, we have*

$$\left\| \sum_{\tau=1}^k \phi_\tau \cdot \left(V(s_\tau) - \mathbb{E}\left[V(s_\tau)|\mathcal{F}_{\tau-1}\right]\right) \right\|_{\Lambda_k^{-1}}^2 \leq 4H^2 \left( \frac{d}{2} \log\left(\frac{k+\lambda}{\lambda}\right) + \log\left(\frac{\mathcal{N}_\epsilon(\mathcal{V})}{\delta}\right) \right) + \frac{8k^2\epsilon^2}{\lambda}.$$

**Lemma 20.** *For any $\epsilon > 0$, the $\epsilon$-covering number of the Euclidean ball in $\mathbb{R}^d$ with radius $R > 0$ is upper bounded by $(1 + 2R/\epsilon)^d$.*

**Lemma 21.** *For a fixed $w$, let $\mathcal{V}$ denote a class of functions mapping from $\mathcal{S}$ to $\mathbb{R}$ with following parametric form*

$$V(.) = \min \left\{ \max_{a \in \mathcal{A}} \langle \mathbf{z}, \boldsymbol{\psi}(., a, w) \rangle + \langle \mathbf{y}, \boldsymbol{\phi}(., a) \rangle + \beta \sqrt{\boldsymbol{\phi}(., a)^\top \mathbf{Y} \boldsymbol{\phi}(., a)}, H \right\},$$

*where the parameters $\beta \in \mathbb{R}$, $\mathbf{z} \in \mathbb{R}^{d'}$, $\mathbf{y} \in \mathbb{R}^d$, and $\mathbf{Y} \in \mathbb{R}^{d \times d}$ satisfy $0 \leq \beta \leq B, \|\mathbf{z}\| \leq z, \|\mathbf{y}\| \leq y,$ and $\|\mathbf{Y}\| \leq \lambda^{-1}$. Assume $\|\boldsymbol{\phi}(s, a)\| \leq 1$ and $\|\boldsymbol{\psi}(s, a, w)\| \leq 1$ for all $(s, a, w) \in \mathcal{S} \times \mathcal{A} \times \mathcal{W}$. Then*

$$\log \left( \mathcal{N}_\epsilon(\mathcal{V}) \right) \leq d' \log(1 + 4z/\epsilon) + d \log(1 + 4y/\epsilon) + d^2 \log \left( \frac{1 + 8B^2 \sqrt{d}}{\lambda \epsilon^2} \right).$$

*Proof.* First, we reparametrize $\mathcal{V}$ by letting $\tilde{\mathbf{Y}} = \beta^2 \mathbf{Y}$. We have

$$V(.) = \min \left\{ \max_{a \in \mathcal{A}} \langle \mathbf{z}, \boldsymbol{\psi}(., a, w) \rangle + \langle \mathbf{y}, \boldsymbol{\phi}(., a) \rangle + \sqrt{\boldsymbol{\phi}(., a)^\top \tilde{\mathbf{Y}} \boldsymbol{\phi}(., a)}, H \right\},$$

for $\|\mathbf{z}\| \leq z, \|\mathbf{y}\| \leq y$, and $\|\tilde{\mathbf{Y}}\| \leq \frac{B^2}{\lambda}$. For any two functions $V_1, V_2 \in \mathcal{V}$ with parameters $\left( \mathbf{z}^1, \mathbf{y}^1, \tilde{\mathbf{Y}}^1 \right)$ and $\left( \mathbf{z}^2, \mathbf{y}^2, \tilde{\mathbf{Y}}^2 \right)$, respectively, we have

$$
\begin{aligned}
\mathrm{dist}(V_1, V_2) &\leq \sup_{(s,a) \in \mathcal{S} \times \mathcal{A}} \left| \left[ \left\langle \mathbf{z}^1, \boldsymbol{\psi}(s, a, w) \right\rangle + \left\langle \mathbf{y}^1, \boldsymbol{\phi}(s, a) \right\rangle + \sqrt{\boldsymbol{\phi}(s, a)^\top \tilde{\mathbf{Y}}^1 \boldsymbol{\phi}(s, a)} \right] \right. \\
&\qquad\qquad \left. - \left[ \left\langle \mathbf{z}^2, \boldsymbol{\psi}(s, a, w) \right\rangle + \left\langle \mathbf{y}^2, \boldsymbol{\phi}(s, a) \right\rangle + \sqrt{\boldsymbol{\phi}(s, a)^\top \tilde{\mathbf{Y}}^2 \boldsymbol{\phi}(s, a)} \right] \right| \\
&\leq \sup_{\boldsymbol{\psi}: \|\boldsymbol{\psi}\| \leq 1, \boldsymbol{\phi}: \|\boldsymbol{\phi}\| \leq 1} \left| \left[ \left\langle \mathbf{z}^1, \boldsymbol{\psi} \right\rangle + \left\langle \mathbf{y}^1, \boldsymbol{\phi} \right\rangle + \sqrt{\boldsymbol{\phi}^\top \tilde{\mathbf{Y}}^1 \boldsymbol{\phi}} \right] - \left[ \left\langle \mathbf{z}^2, \boldsymbol{\psi} \right\rangle + \left\langle \mathbf{y}^2, \boldsymbol{\phi} \right\rangle + \sqrt{\boldsymbol{\phi}^\top \tilde{\mathbf{Y}}^2 \boldsymbol{\phi}} \right] \right| \\
&\leq \sup_{\boldsymbol{\psi}: \|\boldsymbol{\psi}\| \leq 1} \left| \left\langle \mathbf{z}^1 - \mathbf{z}^2, \boldsymbol{\psi} \right\rangle \right| + \sup_{\boldsymbol{\phi}: \|\boldsymbol{\phi}\| \leq 1} \left| \left\langle \mathbf{y}^1 - \mathbf{y}^2, \boldsymbol{\phi} \right\rangle \right| + \sup_{\boldsymbol{\phi}: \|\boldsymbol{\phi}\| \leq 1} \sqrt{\left| \boldsymbol{\phi}^\top \left( \tilde{\mathbf{Y}}^1 - \tilde{\mathbf{Y}}^2 \right) \boldsymbol{\phi} \right|} \\
&\qquad\qquad\qquad\qquad\qquad \text{(because } \left| \sqrt{a} - \sqrt{b} \right| \leq \sqrt{|a - b|} \text{ for } a, b \geq 0) \\
&= \left\| \mathbf{z}^1 - \mathbf{z}^2 \right\| + \left\| \mathbf{y}^1 - \mathbf{y}^2 \right\| + \sqrt{\left\| \tilde{\mathbf{Y}}^1 - \tilde{\mathbf{Y}}^2 \right\|} \\
&\leq \left\| \mathbf{z}^1 - \mathbf{z}^2 \right\| + \left\| \mathbf{y}^1 - \mathbf{y}^2 \right\| + \sqrt{\left\| \tilde{\mathbf{Y}}^1 - \tilde{\mathbf{Y}}^2 \right\|_F}. \qquad\qquad (60)
\end{aligned}
$$

Let $\mathcal{C}_\mathbf{z}$ and $\mathcal{C}_\mathbf{y}$ be $\epsilon/2$-covers of $\{\mathbf{z} \in \mathbb{R}^{d'} : \|\mathbf{z}\| \leq z\}$ and $\{\mathbf{y} \in \mathbb{R}^d : \|\mathbf{y}\| \leq y\}$, respectively, with respect to the 2-norm, and $\mathcal{C}_\mathbf{Y}$ be an $\epsilon^2/4$-cover of $\{\mathbf{Y} \in \mathbb{R}^{d \times d} : \|\mathbf{Y}\|_F \leq \frac{B^2 \sqrt{d}}{\lambda}\}$, with respect to the Frobenius norm. By Lemma 20, we know

$$|\mathcal{C}_\mathbf{z}| \leq (1 + 4z/\epsilon)^{d'}, \quad |\mathcal{C}_\mathbf{y}| \leq (1 + 4y/\epsilon)^d, \quad |\mathcal{C}_\mathbf{Y}| \leq \left( \frac{1 + 8B^2 \sqrt{d}}{\lambda \epsilon^2} \right)^{d^2}.$$

According to (60), it holds that $\mathcal{N}_\epsilon(\mathcal{V}) \leq |\mathcal{C}_\mathbf{z}| |\mathcal{C}_\mathbf{y}| |\mathcal{C}_\mathbf{Y}|$, and therefore

$$\log \left( \mathcal{N}_\epsilon(\mathcal{V}) \right) \leq d' \log(1 + 4z/\epsilon) + d \log(1 + 4y/\epsilon) + d^2 \log \left( \frac{1 + 8B^2 \sqrt{d}}{\lambda \epsilon^2} \right).$$

$\square$

**Lemma 22.** *For a fixed $w$, let $\mathcal{V}$ denote a class of functions mapping from $\mathcal{S}$ to $\mathbb{R}$ with following parametric form*

$$V(.) = \min\left\{\max_{a \in \mathcal{A}}\left\{\langle \mathbf{z}, \boldsymbol{\psi}(.,a,w)\rangle + 2L\beta\sqrt{\boldsymbol{\phi}(.,a)^\top \mathbf{Y} \boldsymbol{\phi}(.,a)}\right\}^+, H\right\},$$

*where the parameters $\beta \in \mathbb{R}$, $\mathbf{z} \in \mathbb{R}^{d'}$ and $\mathbf{Y} \in \mathbb{R}^{d \times d}$ satisfy $0 \le \beta \le B, \|\mathbf{z}\| \le z$, and $\|\mathbf{Y}\| \le \lambda^{-1}$. Assume $\|\boldsymbol{\phi}(s,a)\| \le 1$ and $\|\boldsymbol{\psi}(s,a,w)\| \le 1$ for all $(s,a,w) \in \mathcal{S} \times \mathcal{A} \times \mathcal{W}$. Then*

$$\log\left(\mathcal{N}_\epsilon(\mathcal{V})\right) \le d'\log(1 + 4z/\epsilon) + d^2\log\left(\frac{1 + 8B^2\sqrt{d}}{\lambda\epsilon^2}\right).$$

*Proof.* First, we reparametrize $\mathcal{V}$ by letting $\tilde{\mathbf{Y}} = \beta^2 \mathbf{Y}$. We have

$$V(.) = \min\left\{\max_{a \in \mathcal{A}}\langle \mathbf{z}, \boldsymbol{\psi}(.,a,w)\rangle + \sqrt{\boldsymbol{\phi}(.,a)^\top \tilde{\mathbf{Y}} \boldsymbol{\phi}(.,a)}, H\right\},$$

for $\|\mathbf{z}\| \le z$, and $\|\tilde{\mathbf{Y}}\| \le \frac{B^2}{\lambda}$. For any two functions $V_1, V_2 \in \mathcal{V}$ with parameters $\left(\mathbf{z}^1, \tilde{\mathbf{Y}}^1\right)$ and $\left(\mathbf{z}^2, \tilde{\mathbf{Y}}^2\right)$, respectively, we have

$$\text{dist}(V_1, V_2) \le \sup_{(s,a) \in \mathcal{S} \times \mathcal{A}}\left|\left[\langle \mathbf{z}^1, \boldsymbol{\psi}(s,a,w)\rangle + \sqrt{\boldsymbol{\phi}(s,a)^\top \tilde{\mathbf{Y}}^1 \boldsymbol{\phi}(s,a)}\right] - \left[\langle \mathbf{z}^2, \boldsymbol{\psi}(s,a,w)\rangle + \sqrt{\boldsymbol{\phi}(s,a)^\top \tilde{\mathbf{Y}}^2 \boldsymbol{\phi}(s,a)}\right]\right|$$

$$\le \sup_{\boldsymbol{\psi}:\|\boldsymbol{\psi}\|\le 1, \boldsymbol{\phi}:\|\boldsymbol{\phi}\|\le 1}\left|\left[\langle \mathbf{z}^1, \boldsymbol{\psi}\rangle + \sqrt{\boldsymbol{\phi}^\top \tilde{\mathbf{Y}}^1 \boldsymbol{\phi}}\right] - \left[\langle \mathbf{z}^2, \boldsymbol{\psi}\rangle + \sqrt{\boldsymbol{\phi}^\top \tilde{\mathbf{Y}}^2 \boldsymbol{\phi}}\right]\right|$$

$$\le \sup_{\boldsymbol{\psi}:\|\boldsymbol{\psi}\|\le 1}\left|\langle \mathbf{z}^1 - \mathbf{z}^2, \boldsymbol{\psi}\rangle\right| + \sup_{\boldsymbol{\phi}:\|\boldsymbol{\phi}\|\le 1}\sqrt{\left|\boldsymbol{\phi}^\top\left(\tilde{\mathbf{Y}}^1 - \tilde{\mathbf{Y}}^2\right)\boldsymbol{\phi}\right|}$$

$$\text{(because } \left|\sqrt{a} - \sqrt{b}\right| \le \sqrt{|a - b|} \text{ for } a, b \ge 0)$$

$$= \left\|\mathbf{z}^1 - \mathbf{z}^2\right\| + \sqrt{\left\|\tilde{\mathbf{Y}}^1 - \tilde{\mathbf{Y}}^2\right\|}$$

$$\le \left\|\mathbf{z}^1 - \mathbf{z}^2\right\| + \sqrt{\left\|\tilde{\mathbf{Y}}^1 - \tilde{\mathbf{Y}}^2\right\|_F}. \tag{61}$$

Let $\mathcal{C}_{\mathbf{z}}$ be an $\epsilon/2$-cover of $\{\mathbf{z} \in \mathbb{R}^{d'} : \|\mathbf{z}\| \le z\}$ with respect to the 2-norm, and $\mathcal{C}_{\mathbf{Y}}$ be an $\epsilon^2/4$-cover of $\{\mathbf{Y} \in \mathbb{R}^{d \times d} : \|\mathbf{Y}\|_F \le \frac{B^2\sqrt{d}}{\lambda}\}$, with respect to the Frobenius norm. By Lemma 20, we know

$$|\mathcal{C}_{\mathbf{z}}| \le (1 + 4z/\epsilon)^{d'}, \quad |\mathcal{C}_{\mathbf{Y}}| \le \left(\frac{1 + 8B^2\sqrt{d}}{\lambda\epsilon^2}\right)^{d^2}.$$

According to (61), it holds that $\mathcal{N}_\epsilon(\mathcal{V}) \le |\mathcal{C}_{\mathbf{z}}||\mathcal{C}_{\mathbf{Y}}|$, and therefore

$$\log\left(\mathcal{N}_\epsilon(\mathcal{V})\right) \le d'\log(1 + 4z/\epsilon) + d^2\log\left(\frac{1 + 8B^2\sqrt{d}}{\lambda\epsilon^2}\right).$$

$\square$

**Lemma 23.** *For a fixed $w$, let $\mathcal{V}$ denote a class of functions mapping from $\mathcal{S}$ to $\mathbb{R}$ with following parametric form*

$$V(.) = \min\left\{\max_{a \in \mathcal{A}}\left\{\langle \mathbf{z}, \boldsymbol{\psi}(.,a,w)\rangle + 2L\beta\sqrt{\boldsymbol{\phi}(.,a)^\top \mathbf{Y} \boldsymbol{\phi}(.,a)} + \tilde{\beta}\sqrt{\boldsymbol{\phi}(.,a,w)^\top \tilde{\mathbf{Y}} \boldsymbol{\phi}(.,a,w)}\right\}^+, H\right\},$$

*where the parameters $\beta, \tilde{\beta} \in \mathbb{R}$, $\mathbf{z} \in \mathbb{R}^{d'}$, $\mathbf{Y} \in \mathbb{R}^{d \times d}$ and $\tilde{\mathbf{Y}} \in \mathbb{R}^{d' \times d'}$ satisfy $0 \le \beta \le B$, $0 \le \tilde{\beta} \le \tilde{B} \|\mathbf{z}\| \le z, \|\mathbf{Y}\| \le \lambda^{-1}$ and $\left\| \tilde{\mathbf{Y}} \right\| \le \lambda^{-1}$. Assume $\|\phi(s,a)\| \le 1$ and $\|\psi(s,a,w)\| \le 1$ for all $(s,a,w) \in \mathcal{S} \times \mathcal{A} \times \mathcal{W}$. Then*

$$\log\left(\mathcal{N}_\epsilon(\mathcal{V})\right) \le d' \log(1 + 4z/\epsilon) + d^2 \log\left(\frac{1 + 8B^2\sqrt{d}}{\lambda\epsilon^2}\right) + d'^2 \log\left(\frac{1 + 8\tilde{B}^2\sqrt{d'}}{\lambda\epsilon^2}\right).$$

*Proof.* First, we reparametrize $\mathcal{V}$ by letting $\mathbf{Z} = \beta^2 \mathbf{Y}$ and $\tilde{\mathbf{Z}} = \tilde{\beta}^2 \tilde{\mathbf{Y}}$. We have

$$V(.) = \min\left\{ \max_{a \in \mathcal{A}} \left\langle \mathbf{z}, \psi(.,a,w) \right\rangle + \sqrt{\phi(.,a)^\top \mathbf{Z}\phi(.,a)} + \sqrt{\phi(.,a)^\top \tilde{\mathbf{Z}}\phi(.,a)}, H \right\},$$

for $\|\mathbf{z}\| \le z, \|\mathbf{Z}\| \le \frac{B^2}{\lambda}$, and $\left\| \tilde{\mathbf{Z}} \right\| \le \frac{\tilde{B}^2}{\lambda}$. For any two functions $V_1, V_2 \in \mathcal{V}$ with parameters $\left(\mathbf{z}^1, \mathbf{Z}^1, \tilde{\mathbf{Z}}^1\right)$ and $\left(\mathbf{z}^2, \mathbf{Z}^2, \tilde{\mathbf{Z}}^2\right)$, respectively, we have

$$\begin{aligned}
\text{dist}(V_1, V_2) &\le \sup_{(s,a) \in \mathcal{S} \times \mathcal{A}} \left| \left[ \left\langle \mathbf{z}^1, \psi(s,a,w) \right\rangle + \sqrt{\phi(s,a)^\top \mathbf{Z}^1 \phi(s,a)} + \sqrt{\psi(s,a,w)^\top \tilde{\mathbf{Z}}^1 \psi(s,a,w)} \right] \right. \\
&\qquad \left. - \left[ \left\langle \mathbf{z}^2, \psi(s,a,w) \right\rangle + \sqrt{\phi(s,a)^\top \mathbf{Z}^2 \phi(s,a)} + \sqrt{\psi(s,a,w)^\top \tilde{\mathbf{Z}}^2 \psi(s,a,w)} \right] \right| \\
&\le \sup_{\psi:\|\psi\| \le 1, \phi:\|\phi\| \le 1} \left| \left[ \left\langle \mathbf{z}^1, \psi \right\rangle + \sqrt{\phi^\top \mathbf{Z}^1 \phi} + \sqrt{\psi^\top \tilde{\mathbf{Z}}^1 \psi} \right] - \left[ \left\langle \mathbf{z}^2, \psi \right\rangle + \sqrt{\phi^\top \mathbf{Z}^2 \phi} + \sqrt{\psi^\top \tilde{\mathbf{Z}}^2 \psi} \right] \right| \\
&\le \sup_{\psi:\|\psi\| \le 1} \left| \left\langle \mathbf{z}^1 - \mathbf{z}^2, \psi \right\rangle \right| + \sup_{\phi:\|\phi\| \le 1} \sqrt{\left| \phi^\top \left( \mathbf{Z}^1 - \mathbf{Z}^2 \right) \phi \right|} + \sup_{\psi:\|\phi\| \le 1} \sqrt{\left| \psi^\top \left( \tilde{\mathbf{Z}}^1 - \tilde{\mathbf{Z}}^2 \right) \psi \right|} \\
&\qquad\qquad\qquad\qquad\qquad \text{(because } \left| \sqrt{a} - \sqrt{b} \right| \le \sqrt{|a-b|} \text{ for } a, b \ge 0\text{)} \\
&= \left\| \mathbf{z}^1 - \mathbf{z}^2 \right\| + \sqrt{\|\mathbf{Z}^1 - \mathbf{Z}^2\|} + \sqrt{\left\| \tilde{\mathbf{Z}}^1 - \tilde{Z}^2 \right\|} \\
&\le \left\| \mathbf{z}^1 - \mathbf{z}^2 \right\| + \sqrt{\|\mathbf{Z}^1 - \mathbf{Z}^2\|_F} + \sqrt{\left\| \tilde{\mathbf{Z}}^1 - \tilde{\mathbf{Z}}^2 \right\|_F}.
\end{aligned} \qquad (62)$$

Let $\mathcal{C}_\mathbf{z}$ be an $\epsilon/2$-cover of $\{\mathbf{z} \in \mathbb{R}^{d'} : \|\mathbf{z}\| \le z\}$ with respect to the 2-norm, $\mathcal{C}_\mathbf{Z}$ be an $\epsilon^2/4$-cover of $\{\mathbf{Z} \in \mathbb{R}^{d \times d} : \|\mathbf{Z}\|_F \le \frac{B^2\sqrt{d}}{\lambda}\}$, and $\mathcal{C}_{\tilde{\mathbf{Z}}}$ be an $\epsilon^2/4$-cover of $\{\tilde{\mathbf{Z}} \in \mathbb{R}^{d' \times d'} : \left\| \tilde{\mathbf{Z}} \right\|_F \le \frac{\tilde{B}^2\sqrt{d}}{\lambda}\}$ with respect to the Frobenius norm. By Lemma 20, we know

$$|\mathcal{C}_\mathbf{z}| \le (1 + 4z/\epsilon)^{d'}, \quad |\mathcal{C}_\mathbf{Z}| \le \left(\frac{1 + 8B^2\sqrt{d}}{\lambda\epsilon^2}\right)^{d^2}, \quad |\mathcal{C}_{\tilde{\mathbf{Z}}}| \le \left(\frac{1 + 8\tilde{B}^2\sqrt{d'}}{\lambda\epsilon^2}\right)^{d'^2}.$$

According to (62), it holds that $\mathcal{N}_\epsilon(\mathcal{V}) \le |\mathcal{C}_\mathbf{z}||\mathcal{C}_\mathbf{Y}|$, and therefore

$$\log\left(\mathcal{N}_\epsilon(\mathcal{V})\right) \le d' \log(1 + 4z/\epsilon) + d^2 \log\left(\frac{1 + 8B^2\sqrt{d}}{\lambda\epsilon^2}\right) + d'^2 \log\left(\frac{1 + 8\tilde{B}^2\sqrt{d'}}{\lambda\epsilon^2}\right).$$

$\square$

**Lemma 24.** *Let $\mathcal{V}$ denote a class of functions mapping from $\mathcal{S}$ to $\mathbb{R}$ with following parametric form*

$$V(.,.) = \min\left\{ \max_{a \in \mathcal{A}} \left\{ \left\langle \mathbf{z}, \psi(.,a,.) \right\rangle + 2L\beta\sqrt{\psi(.,a,.)^\top \mathbf{Y}\psi(.,a,.)} \right\}^+, H \right\},$$

*where the parameters $\beta \in \mathbb{R}$, $\mathbf{z} \in \mathbb{R}^{d'}$ and $\mathbf{Y} \in \mathbb{R}^{d' \times d'}$ satisfy $0 \le \beta \le B, \|\mathbf{z}\| \le z$, and $\|\mathbf{Y}\| \le \lambda^{-1}$. Assume $\|\psi(s,a,w)\| \le 1$ for all $(s,a,w) \in \mathcal{S} \times \mathcal{A} \times \mathcal{W}$. Then*

$$\log\left(\mathcal{N}_\epsilon(\mathcal{V})\right) \le d' \log(1 + 4z/\epsilon) + d'^2 \log\left(\frac{1 + 8B^2\sqrt{d'}}{\lambda\epsilon^2}\right).$$

*Proof.* First, we reparametrize $\mathcal{V}$ by letting $\tilde{\mathbf{Y}} = \beta^2 \mathbf{Y}$. We have

$$V(.,.) = \min\left\{\max_{a \in \mathcal{A}} \left\langle \mathbf{z}, \boldsymbol{\psi}(.,a,.) \right\rangle + \sqrt{\boldsymbol{\psi}(.,a,.)^\top \tilde{\mathbf{Y}} \boldsymbol{\psi}(.,a,.)}, H\right\},$$

for $\|\mathbf{z}\| \le z$, and $\left\|\tilde{\mathbf{Y}}\right\| \le \frac{B^2}{\lambda}$. For any two functions $V_1, V_2 \in \mathcal{V}$ with parameters $\left(\mathbf{z}^1, \tilde{\mathbf{Y}}^1\right)$ and $\left(\mathbf{z}^2, \tilde{\mathbf{Y}}^2\right)$, respectively, we have

$$\begin{aligned}
\text{dist}(V_1, V_2) &\le \sup_{(s,a,w) \in \mathcal{S} \times \mathcal{A} \times \mathcal{W}} \left| \left[\left\langle \mathbf{z}^1, \boldsymbol{\psi}(s,a,w) \right\rangle + \sqrt{\boldsymbol{\psi}(s,a)^\top \tilde{\mathbf{Y}}^1 \boldsymbol{\psi}(s,a)}\right] \right. \\
&\qquad\qquad \left. - \left[\left\langle \mathbf{z}^2, \boldsymbol{\psi}(s,a,w) \right\rangle + \sqrt{\boldsymbol{\psi}(s,a,w)^\top \tilde{\mathbf{Y}}^2 \boldsymbol{\psi}(s,a,w)}\right] \right| \\
&\le \sup_{\boldsymbol{\psi}:\|\boldsymbol{\psi}\| \le 1} \left| \left[\left\langle \mathbf{z}^1, \boldsymbol{\psi} \right\rangle + \sqrt{\boldsymbol{\psi}^\top \tilde{\mathbf{Y}}^1 \boldsymbol{\psi}}\right] - \left[\left\langle \mathbf{z}^2, \boldsymbol{\psi} \right\rangle + \sqrt{\boldsymbol{\psi}^\top \tilde{\mathbf{Y}}^2 \boldsymbol{\psi}}\right] \right| \\
&\le \sup_{\boldsymbol{\psi}:\|\boldsymbol{\psi}\| \le 1} \left| \left\langle \mathbf{z}^1 - \mathbf{z}^2, \boldsymbol{\psi} \right\rangle \right| + \sup_{\boldsymbol{\psi}:\|\boldsymbol{\psi}\| \le 1} \sqrt{\left| \boldsymbol{\psi}^\top \left(\tilde{\mathbf{Y}}^1 - \tilde{\mathbf{Y}}^2\right) \boldsymbol{\psi} \right|} \\
&\qquad\qquad\qquad \text{(because } \left|\sqrt{a} - \sqrt{b}\right| \le \sqrt{|a-b|} \text{ for } a, b \ge 0\text{)} \\
&= \left\|\mathbf{z}^1 - \mathbf{z}^2\right\| + \sqrt{\left\|\tilde{\mathbf{Y}}^1 - \tilde{\mathbf{Y}}^2\right\|} \\
&\le \left\|\mathbf{z}^1 - \mathbf{z}^2\right\| + \sqrt{\left\|\tilde{\mathbf{Y}}^1 - \tilde{\mathbf{Y}}^2\right\|_F}.
\end{aligned} \tag{63}$$

Let $\mathcal{C}_{\mathbf{z}}$ be an $\epsilon/2$-cover of $\{\mathbf{z} \in \mathbb{R}^{d'} : \|\mathbf{z}\| \le z\}$ with respect to the 2-norm, and $\mathcal{C}_{\mathbf{Y}}$ be an $\epsilon^2/4$-cover of $\{\mathbf{Y} \in \mathbb{R}^{d' \times d'} : \|\mathbf{Y}\|_F \le \frac{B^2 \sqrt{d'}}{\lambda}\}$, with respect to the Frobenius norm. By Lemma 20, we know

$$|\mathcal{C}_{\mathbf{z}}| \le (1 + 4z/\epsilon)^{d'}, \quad |\mathcal{C}_{\mathbf{Y}}| \le \left(\frac{1 + 8B^2 \sqrt{d'}}{\lambda \epsilon^2}\right)^{d'^2}.$$

According to (63), it holds that $\mathcal{N}_\epsilon(\mathcal{V}) \le |\mathcal{C}_{\mathbf{z}}||\mathcal{C}_{\mathbf{Y}}|$, and therefore

$$\log\left(\mathcal{N}_\epsilon(\mathcal{V})\right) \le d' \log(1 + 4z/\epsilon) + d'^2 \log\left(\frac{1 + 8B^2 \sqrt{d'}}{\lambda \epsilon^2}\right).$$

$\square$

## H    DETAILS OF THE EXPERIMENTS

In all the experiments, we have chosen $\delta = 0.01$, $\lambda = 1$, $d = 5$, and $H = 5$. The parameters $\{\boldsymbol{\eta}_h\}_{h \in [H]}$ are drawn from $\mathcal{N}(0, I_{d'})$. In order to tune parameters $\{\boldsymbol{\mu}_h(.)\}_{h \in [H]}$ and the feature mappings $\boldsymbol{\phi}$ such that they are compatible with Assumption 1, we consider that the feature space $\{\boldsymbol{\phi}(s,a) : (s,a) \in \mathcal{S} \times \mathcal{A}\}$ is a subset of the $d$-dimensional simplex, $\{\boldsymbol{\phi} \in \mathbb{R}^d : \sum_{i=1}^d \phi_i = 1, \phi_i \ge 0, \phi_i \le 1, \forall i \in [d]\}$, and $\mathbf{e}_i^\top \boldsymbol{\mu}_h(.)$ is an arbitrary probability measure over $\mathcal{S}$ for all $i \in [d]$.

The results shown in Figure 2a depict averages over 50 realizations for the main setup considered throughout the paper with $m = 5$ and the results shown in Figure 2b depict averages over 50 realizations, for the more general setup of Remark 2 with $d' = 10$. For the results shown in Figure 2a, the mappings $\boldsymbol{\rho}(w)$ are drawn from $\mathcal{N}(0, I_m)$ except for the $n = m$ representative tasks $\{w^{(j)}\}_{j \in [m]}$ introduced in Assumption 3, for which we set $\boldsymbol{\rho}(w^{(j)}) = \mathbf{e}_j$ for $j \in [m]$. For the results shown in Figure 2b, the mappings $\boldsymbol{\psi}(s,a,w)$ are drawn from $\mathcal{N}(0, I_{d'})$ and we set $\boldsymbol{\psi}(s,a,w^{(j)}) = \mathbf{e}_j$ for $j \in [d']$, where $\{w^{(j)}\}_{j \in [d']}$ are $n = d'$ representative tasks introduced in Assumption 5 in Appx. D. The parameters $\{\boldsymbol{\eta}_h\}_{h \in [H]}$ are drawn from $\mathcal{N}(0, I_{d'})$, where $d' = m \times d = 25$ in Figure 2a. In our

experiments, the exact same settings are used for both UCBlvd and Lifelong-LSVI in both Figures 2a and 2b. We chose fairly large $d$, $m$, and $d'$ and by checking online, we noticed that the optimal value of QCQP in (8) happens always to be zero. All these together suggest that the assumptions made in the paper approximately hold. Figures 2a and 2b depict the average per-episode reward of UCBlvd and state the average number of planning calls and compare them to those of baseline algorithm Lifelong-LSVI, a direct extension of LSVI-UCB in Jin et al. (2020). The results emphasize the value of UCBlvd in terms of requiring much smaller numbers of planning calls. The plots verify that the performances of Lifelong-LSVI and UCBlvd are almost the same statistically, while UCBlvd uses much smaller numbers of planning calls (1000 vs $\sim 20$).

In Figure 3, we plot UCBlvd's number of planning calls for different number of task episodes, $K$, while the setting is same as that in 2a. In this figure, we empirically verify the logarithmic dependence of number of planning calls on $K$ as suggested by Theorem 2.

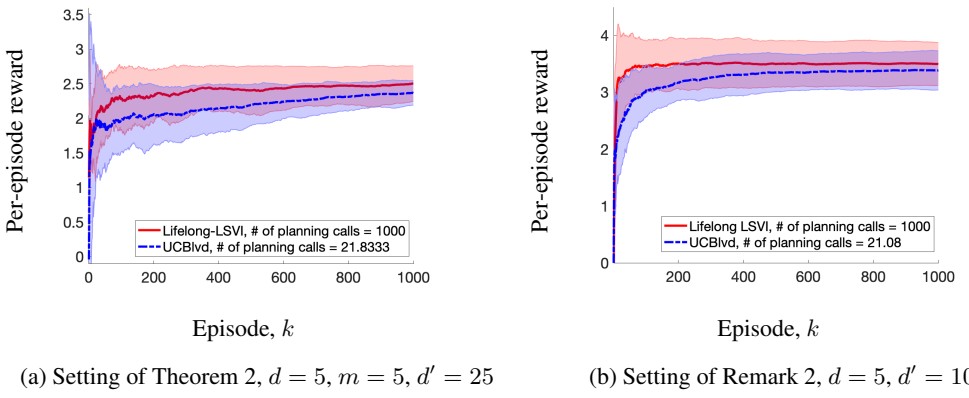

(a) Setting of Theorem 2, $d = 5$, $m = 5$, $d' = 25$      (b) Setting of Remark 2, $d = 5$, $d' = 10$

Figure 2: UCBlvd vs Lifelong-LSVI

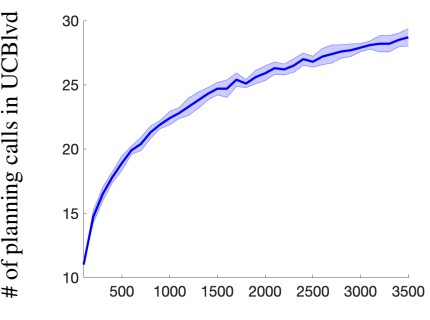

Total number of task episodes, $K$

Figure 3: Setting of Theorem 2, $d = 5$, $m = 5$, $d' = 25$

