# OpenReview forum: "Provably Efficient Lifelong Reinforcement Learning with Linear Representation"
_ICLR.cc/2023/Conference — ICLR 2023 poster_

### Official Review · Reviewer_vHiq · 2022-10-21

**Confidence:** 3
**Correctness:** 4
**Technical Novelty And Significance:** 3
**Empirical Novelty And Significance:** 3
**Recommendation:** 6

**Clarity, Quality, Novelty And Reproducibility:**

The clarity is reasonable, at least for someone familiar with prior work in the area. Reproducibility doesn't seem to be much of a concern, based on the supplemental material anyway. I believe the work is technically sound.

So the main question is novelty. As mentioned above, there is some modest novelty in this work: the emphasis on the number of times the MDP needs to be re-solved and means to control this. So it's reasonably novel, but not a breakthrough.

**Strength And Weaknesses:**

The main strengths of the paper are
1. It considers the computational cost in this online setting, which is usually ignored (e.g., in bandits) but would indeed be prohibitively substantial here using existing methods.
2. There is some technical novelty in the design and analysis of the test for the need to recompute the solution towards the above.
3. There are some experiments, albeit on synthetic tasks, so the method is potentially useable in practice.

The main weaknesses are
1. Technically, apart from the analysis of the number of re-solves, it seems to be a straightforward extension of the previous works on the non-contextual MDPs, e.g., Yang & Wang 2019.
2. The individual planning calls are now more expensive, as they have to solve a QCQP rather than a linear program.
3. The synthetic experiments don't give a sense of how the algorithm would perform in a real-world setting.

**Summary Of The Paper:**

This paper may most accurately be summarized as giving polynomial-time algorithms that obtain sublinear regret in linear contextual MDPs when the transition and Q-functions are linear in the state-action features and the context, and that only need to re-solve the MDP logarithmically many times w.r.t. the number of tasks.

More concretely, the contextual MDP has a fixed dynamics (independent of the context), but reward functions that vary with a side "context" w that is known to the learner. The learner interacts with this MDP for K episodes of horizon H. The main assumption is that the transition kernel and Q-function are linear in some known basis functions. A similar assumption was used previously to obtain a polynomial time algorithm for such MDPs sans context; so this paper is essentially extending that line of work to include context.

**Summary Of The Review:**

I think there's enough of a contribution to warrant accepting this paper.

---

> ### Author Response · Authors · 2022-11-08
> **Response to Reviewer vHiq**
>
> We thank the reviewer for their valuable time and feedback; we are deeply encouraged by the reviewer’s comments. We make the following clarifications below in response to the reviewer’s concerns in the order they were raised.
>
> **Novelty:** We emphasize that our work is beyond just a direct extension of existing non-contextual methods. In Section 3, we discuss how Lifelong-LSVI, as a direct extension of LSVI-UCB in Jin et al., 2020, which is a non-contextual single-task algorithm, is not computationally efficient and why it cannot be made computationally efficient in an easy manner. While in general we can directly extend Lifelong-LSVI to use feature $\boldsymbol\psi(s,a,w)\in\mathbb{R}^{d^\prime}$ with $d^\prime\geq d$ and make instead a completeness assumption based on $\psi(s,a,w)$ directly (see Remark 3), this would increase the regret from that with $\boldsymbol\phi(s,a)\in\mathbb{R}^d$, because it implies that dynamics are also context-dependent, which is not the case in our formulation. This alternate completeness assumption in Remark 3 is also in some sense stronger than Assumption 2, as it assumes completeness for a more complicated function class.
>
> Therefore, one of our key contributions is to show sublinear planning calls with regret of the same order as that of Lifelong-LSVI is possible (under Assumption 2). UCBlvd achieves this by separating the planning into a novel two-step process: 1) independent planning with $\boldsymbol\phi$ for a set of representative task contexts and 2) distilling the planned results into a multi-task value function parameterized by $\boldsymbol\psi$. As a result, UCBlvd is able to run a doubling schedule to decide whether replanning is necessary, which makes the total number of planning calls sublinear. This specific design of algorithm makes UCBlvd significantly different from Lifelong-LSVI which can be seen as a straightforward extension of the previous works on the non-contextual MDPs in Jin et al., 2020.
>
> **Computational cost:** In what follows, we clarify on how the time complexity of UCBlvd compares to that of Lifelong LSVI, where the former solves a QCQP and the latter solves a linear program in each planning call. When we compute $(\{\boldsymbol{\Lambda}\}_\{h\}^k)^\{-1\}$ by the Sherman-Morrison formula, the computational complexity of Lifelong-LSVI is dominated by line 5 in computing $\text{max}_\{a\in \mathcal{A}\} Q_\{h+1\}^k(s_\{h+1\}^\tau, a)$ for all $\tau\in[k]$. This takes $\mathcal{O}(d^2|\mathcal{A}|K)$ per step, which gives a total runtime $\mathcal{O}(d^2|\mathcal{A}|HK^2)$ as the total number of planning calls in Lifelong-LSVI is $K$. In UCBlvd, every planning call takes $\tilde{\mathcal{O}}(md^2|\mathcal{A}|K+m^3d^3)$, where the first term is the time-complexity of computing  $\tilde{\boldsymbol\theta}_h^k(w^\{(j)\})$ for all $j\in[n]$ (i.e., $\text{max}_\{a\in \mathcal{A}\} Q_\{h+1\}^k(s_\{h+1\}^\tau, a, w^\{j\})$ for all  $(j,\tau)\in[n]\times[k]$), and the second term is the time-complexity of our convex QCQP with $n+1$ constraints and $nd+md$ variables (note that $n\leq m$). Since UCBlvd's number of planning calls is at most $\tilde{\mathcal{O}}(dH)$, the total runtime of UCBlvd is $\tilde{\mathcal{O}}(H^2(md^3|\mathcal{A}|K+m^3d^4))$. Therefore, UCBlvd enjoys a smaller time complexity by a factor of $K$ compared to that of Lifelong-LSVI, which is a significant reduction in practical scenarios where $K>>d^\prime = md$.  We followed your suggestion and added a discussion on time-complexity comparison between UCBlvd and Lifelong-LSVI in Appendix B.5 of the revision.
>
> **Experiments for real-world settings:** As this work is mainly
> theoretical in nature, the purpose of our experiments is to complement our theoretical results. That said, we’d appreciate any real-world environment suggestions for lifelong RL and try our best to make such additions to the final revision.
>
>
> We thank the reviewer once again for their insightful comments that will help improve the quality of our paper. We would be happy to answer any further questions.

---

> ### Author Response · Authors · 2022-11-13
> **We would appreciate feedback on our response so that there is time to answer any questions**
>
> Dear reviewer vHiq,
>
> We would like to thank you for your valuable time spent on our submission. We made an effort to respond to all your concerns. Since we are approaching the deadline for the Discussion Stage 1, we would much appreciate it if you could please let us know whether you still have any remaining issues regarding our submission and whether you are willing to reconsider your score.
>
> Thank you in advance for your time and attention.

---

> ### Author Response · Authors · 2022-11-18
> **We would appreciate your feedback**
>
> Dear Reviewer vHiq,
>
> Since there is only one day left for the Discussion Stage 1, we would appreciate it if you could please let us know whether you still have any remaining questions or concerns and we would be happy to engage in further discussions.
>
> Thanks, Authors

---

> ### Author Response · Authors · 2022-11-28
> **Are there any additional questions or concerns?**
>
> Dear Reviewer vHiq,
>
> We would like to sincerely thank you again for your time spent on our submission! As the discussion stage 2 approaches its end, we were hoping to kindly ask whether you still have any remaining issues regarding our submission and whether you are willing to reconsider your score.
>
> Thank you,
> Authors

---

> ### Author Response · Authors · 2022-12-05
> **A friendly reminder that the discussion stage 2 will be closed in 7 days.**
>
> Dear Reviewer vHiq,
>
> We were wondering if our response has resolved your concerns. In our response, we shed light on the novelty of our work and how our work is beyond just a direct extension of existing non-contextual methods. We also clarified how time-complexity of our algorithm, UCBlvd, compares to that of the baseline algorithm, Lifelong-LSVI, and how our algorithm’s computation efficiency is superior in practical cases where $K>>md$. We look forward to receiving feedback from you as to whether you still have any remaining issues and questions.
>
> Thank you,
> Authors

---

> ### Author Response · Authors · 2022-12-13
> **Last day of discussion: we'd love to hear your feedback.**
>
> Dear reviewer vHiq,
>
> Thank you again for your valuable time and reviews. As today is the final day of the discussion period, we'd love to hear your feedback and try our best to address any remaining concerns in these final hours.
>
> Thank you for your time,
> Authors

---

### Official Review · Reviewer_yDMc · 2022-10-23

**Confidence:** 3
**Correctness:** 4
**Technical Novelty And Significance:** 2
**Empirical Novelty And Significance:** Not applicable
**Recommendation:** 6

**Clarity, Quality, Novelty And Reproducibility:**

The paper is clearly written and relatively easy to follow.

The algorithms proposed in the paper are limited in terms of novelty. Particularly, the main algorithm UCBlvd is close to existing low switching cost RL algorithms. The key contributions of UCBlvd, to me, seem to be Assumptions 2 and 3, which jointly identify problem settings under which this kind of low switching cost algorithm is feasible.

The proofs are reproducible. However, code used to generate Figure 1 is not included in the submission and cannot be verified easily. Nevertheless, the plot is inline with existing results and exhibits expected behavior.

The results in the paper are correct and justified theoretically.

**Strength And Weaknesses:**

Strength
- The setting is novel and is relevant to RL practitioners.
- Experimental results are able to show that UCBlvd enjoys roughly the same regret while requiring far less calls to the planning subroutine.

Weaknesses
- The assumption that the reward functions for all tasks are known beforehand is a strong one. In the lifelone learning environment described here, the only source of uncertainty lies in the transition kernel, making the setting close to a single-agent single-task RL problem, which has been thoroughly examined in the linear MDP setting. While Example 1 offers a good intuitive justification, it would be great if the authors could offer more justification for the setting they assume, perhaps in the appendix, similar to the discussion immediately after Assumption 2.

------- Post rebuttal update --------
My concerns have been addressed by the authors and I have adjusted my score accordingly.

**Summary Of The Paper:**

The paper studies **lifelong** reinforcement learning with linear function approximation. In a typical RL setup, the reward function depends only on the state-action pair. The **lifelong** component of this works comes from the assumption that the reward function is dependent on the state-action pair $(s, a)$ in addition to a task context variable, denoted $w$. It is assumed that the agent observes $w$ directly but $w$ is allowed to change from one episode to the next. The paper also addresses how to reduce the number of planning calls in the lifelong learning context.

Overall the contributions, to me, seem two-fold:
1. Introduces the lifelong RL setup and Lifelong-LSVI, a naive baseline in the lifelong learning problem.
2. Identifies two assumptions under which a low-switching cost algorithm is possible for lifelong RL. The authors justified the assumption using a toy example (see Example 1).

**Summary Of The Review:**

Overall, the paper introduces a novel setting for linear MDP. The authors are able to derive conditions under which low switching cost RL algorithms work. The extra assumptions needed to enable low switching cost RL is unique to the setting and is a novel contribution from the authors. Unfortunately, the mathematical model for lifelong RL seems overly reductionist at this moment and further explanation and justification would be appreciated.

---

> ### Author Response · Authors · 2022-11-08
> **Response to Reviewer yDMc**
>
> We thank the reviewer for their valuable time and feedback. We make the following clarifications below.
>
> **Unknownness of rewards:** We make the assumption of known reward functions **only** for simplicity of presentation. In Remark 1 and appendix C, we discuss in detail that our results can be easily extended to the settings with unknown task rewards and how this extension affects our bounds. In particular, when the rewards are unknown, we can adopt a slightly different completeness assumption with an extra bonus in terms of $\boldsymbol{\psi}$, and then combine tools from linear bandits (Abbasi-Yadkori et al., 2011)  and our proof of Theorem 2. Because reward learning affects the radius of the confidence intervals for $\boldsymbol{\theta}_h^k(w)$, the number of planning calls and regret would increase by factors of $\mathcal{O}(m)$ and $\mathcal{O}(\sqrt{m})$, respectively, compared to those in Theorem 2.
>
>
> **Novelty:**  In Section 3, we discuss how Lifelong-LSVI, as a direct extension of LSVI-UCB in Jin et al., 2020, is not computationally efficient and why it cannot be made computationally efficient in an easy manner. While in general we can directly extend Lifelong-LSVI to use feature $\boldsymbol\psi(s,a,w)\in\mathbb{R}^{d^\prime}$ with $d^\prime\geq d$ and make instead a completeness assumption based on $\psi(s,a,w)$ directly (see Remark 3), this would increase the regret from that with $\boldsymbol\phi(s,a)\in\mathbb{R}^d$, because it implies that dynamics are also context-dependent, which is not the case in our formulation. This alternate completeness assumption in Remark 3 is also in some sense stronger than Assumption 2, as it assumes completeness for a more complicated function class.
>
> Therefore, one of our key contributions is to show sublinear planning calls with regret of the same order as that of Lifelong-LSVI is possible (under Assumption 2). UCBlvd achieves this by separating the planning into a novel two-step process: 1) independent planning with $\boldsymbol\phi$ for a set of representative task contexts and 2) distilling the planned results into a multi-task value function parameterized by $\boldsymbol\psi$. As a result, UCBlvd is able to run a doubling schedule to decide whether replanning is necessary, which makes the total number of planning calls sublinear.
>
> This two-step design based on QCQP deviates significantly from the typical design of LSVI-UCB algorithms in the literature (e.g., in Jin et al., 2020), and the proof techniques developed here are novel to our knowledge.
>
> We hope that our response will address the reviewer’s concerns. We would be more than happy to engage in further discussions and respond to other questions.

---

> ### Author Response · Authors · 2022-11-13
> **We would appreciate feedback on our response so that there is time to answer any questions**
>
> Dear reviewer yDMc,
>
> We would like to thank you for your valuable time spent on our submission. We made an effort to respond to all your concerns. Since we are approaching the deadline for the Discussion Stage 1, we would much appreciate it if you could please let us know whether you still have any remaining issues regarding our submission and whether you are willing to reconsider your score.
>
> Thank you in advance for your time and attention.

---

> ### Author Response · Authors · 2022-11-18
> **We would appreciate your feedback**
>
> Dear Reviewer yDMc,
>
> Since there is only one day left for the Discussion Stage 1, we would appreciate it if you could please let us know whether you still have any remaining questions or concerns and we would be happy to engage in further discussions.
>
> Thanks, Authors

---

### Official Review · Reviewer_HSfE · 2022-10-23

**Confidence:** 4
**Clarity, Quality, Novelty And Reproducibility:** The result is novel to my best knowle…
**Correctness:** 4
**Technical Novelty And Significance:** 3
**Empirical Novelty And Significance:** 3
**Recommendation:** 6

**Strength And Weaknesses:**

Strength: The paper proposes a novel algorithm which is both sample-efficient and computation-efficient in theory. The computation-efficiency seems appealing and insightful, but some more detailed analysis might be helpful for people to fully understand the contribution. In particular, the paper decreases the $K^2$-cost of planning call to $log(K)$ by skipping planning in the case that no sufficient new information is obtained.

Weakness: 1) The sample-efficiency of the algorithm has very limited insight given previous works. 2) It will be more clear for the readers to understand the contribution of the paper if the paper can explicitly analyze how much computation cost is reduced, rather than just from the reduction of planning calls. In deed, the new algorithm still require to calculate a new covariance matrix in each iteration. When re-planning is triggered, it requires solving a QCQP, which is not required in Algorithm 1. 3) It is not very clear how strong the cross product structure in Assumption 3 is. Does it includes the case where there is only finite number of possible $w$?

**Summary Of The Paper:**

This paper studies lifelong reinforcement learning with different tasks. In particular, the paper proposes a new algorithm that is both sample-efficient and computation-efficient. The underlying models are variants of linear MDPs, which are widely studied by the reinforcement learning theory community.

**Summary Of The Review:**

The reviewer believes that paper is a little below the acceptance threshold. It will be helpful if the reviewers' concerns mentioned above can be resolved in the author's responses.

---

> ### Author Response · Authors · 2022-11-08
> **Response to Reviewer HSfE**
>
> We thank the reviewer for their valuable time and feedback.  Below, we address your concerns in the order they were raised:
>
> **1) Regret guarantees:** As stated in the paragraph after Theorem 2, UCBlvd has the same regret bound as Lifelong-LSVI in Theorem 1, but reduces the number of planning calls from $K$ to $dH\log(K)$, which is made possible by the unique QCQP-based distillation step of UCBlvd in (8). In the paragraph after Theorem 1, we mentioned that Theorem 1 implies that for the special case studied by Wu et al. (2021) (Example 1), the regret bound of Lifelong-LSVI becomes $\tilde{\mathcal{O}}(\sqrt{md^3H^3T})$. This rate is optimal in terms of its dependency on m, as shown in Wu et al. (2021). Furthermore, this rate matches the LSVI-UCB’s regret dependencies on d and H for the single-task setting (Jin et al., 2020). Since UCBlvd has the same regret as that of Lifelong-LSVI, it also enjoys all the above mentioned optimalities at a much smaller number of planning calls.
>
> **2) Computational cost:** We now clarify on how the time complexity of UCBlvd compares to that of Lifelong LSVI. When we compute $(\{\boldsymbol{\Lambda}\}_\{h\}^k)^\{-1\}$ by the Sherman-Morrison formula, the computational complexity of Lifelong-LSVI is dominated by line 5 in computing $\text{max}_\{a\in \mathcal{A}\} Q_\{h+1\}^k(s_\{h+1\}^\tau, a)$ for all $\tau\in[k]$. This takes $\mathcal{O}(d^2|\mathcal{A}|K)$ per step, which gives a total runtime $\mathcal{O}(d^2|\mathcal{A}|HK^2)$ as the total number of planning calls in Lifelong-LSVI is $K$. In UCBlvd, every planning call takes $\tilde{\mathcal{O}}(md^2|\mathcal{A}|K+m^3d^3)$, where the first term is the time-complexity of computing  $\tilde{\boldsymbol\theta}_h^k(w^\{(j)\})$ for all $j\in[n]$ (i.e., $\text{max}_\{a\in \mathcal{A}\} Q_\{h+1\}^k(s_\{h+1\}^\tau, a, w^\{j\})$ for all  $(j,\tau)\in[n]\times[k]$), and the second term is the time-complexity of our convex QCQP with $n+1$ constraints and $nd+md$ variables (note that $n\leq m$). Since UCBlvd's number of planning calls is at most $\tilde{\mathcal{O}}(dH)$, the total runtime of UCBlvd is $\tilde{\mathcal{O}}(H^2(md^3|\mathcal{A}|K+m^3d^4))$. Therefore, UCBlvd enjoys a smaller time complexity by a factor of $K$ compared to that of Lifelong-LSVI, which is a significant reduction in practical scenarios where $K>>d^\prime = md$.  We followed your suggestion and added a discussion on time-complexity comparison between UCBlvd and Lifelong-LSVI in Appendix B.5 of the revision.
>
>
> **3) Assumption 3:** The cross product structure of $\boldsymbol \psi$, combined with linear structure of the reward, would translate to a generalization/relaxation of the reward structure in Example 1, where the reward is a linear combination of entries of a reward vector in a multi-objective RL formulation. This weighted reward structure is common and a special case of our cross product structure. In particular, it is studied in Wu et al., 2021 for tabular MDPs. Our cross product structure also includes the case where there are only a finite number of possible task contexts.
>
>
> We hope that our answers above will address the reviewer's questions and concerns. We would be happy to answer any further questions.

---

> > ### Comment · Reviewer_HSfE · 2022-12-04
> > **Response to the Author(s)**
> >
> > Thanks for the very detailed clarification. I have no more questions and have changed the score to acceptance.

---

> > > ### Author Response · Authors · 2022-12-04
> > > **Thank you!**
> > >
> > > Dear Reviewer HSfE,
> > >
> > > We are glad that our response has addressed your concerns and would like to thank you for your valuable time spent on our submission and for increasing your score.
> > >
> > > Thanks, Authors

---

> ### Author Response · Authors · 2022-11-13
> **We would appreciate feedback on our response so that there is time to answer any questions**
>
> Dear reviewer HSfE,
>
> We would like to thank you for your valuable time spent on our submission. We made an effort to respond to all your concerns. Since we are approaching the deadline for the Discussion Stage 1, we would much appreciate it if you could please let us know whether you still have any remaining issues regarding our submission and whether you are willing to reconsider your score.
>
> Thank you in advance for your time and attention.

---

> ### Author Response · Authors · 2022-11-18
> **We would appreciate your feedback**
>
> Dear Reviewer HSfE,
>
> Since there is only one day left for the Discussion Stage 1, we would appreciate it if you could please let us know whether you still have any remaining questions or concerns and we would be happy to engage in further discussions.
>
> Thanks, Authors

---

> ### Author Response · Authors · 2022-11-28
> **Are there any additional questions or concerns?**
>
> Dear Reviewer HSfE,
>
> We would like to sincerely thank you again for your time spent on our submission! As the discussion stage 2 approaches its end, we were hoping to kindly ask whether you still have any remaining issues regarding our submission and whether you are willing to reconsider your score.
>
> Thank you,
> Authors

---

### Official Review · Reviewer_hkRh · 2022-10-31

**Confidence:** 3
**Correctness:** 4
**Technical Novelty And Significance:** 2
**Empirical Novelty And Significance:** 1
**Recommendation:** 6

**Clarity, Quality, Novelty And Reproducibility:**

The writing is clear and well structured.

I did not check all proofs in the supplementary material, the theoretical analysis in the main text appears sound.

The work's originality is limited in that it builds on an existing method to extend it to the lifelong case.

I believe the experiments should be reproducible with the given material.

**Strength And Weaknesses:**

Strengths

* Though quite dense, the work is presented in a digestible way. I found the background, the method and the theoretical part of paper clearly organised and concise.
* The problem studied is important and relevant, authors have identified gap in the field of RL (theoritical lifelong RL) and provided ideas and tools to start addressing it.

Weaknesses
* My main concern is that authors have not conveyed if or how the contributions of the work will extend beyond the limited case of linear environment under the completeness assumption. I appreciate that extensions to the non linear case / with fewer assumptions might not be possible yet, but then a discussion around how we expect the performance to be impacted would be necessary (experiments on non synthetic environments could help to show that).
* The reason for my previous point is that I did not grasp from reading the paper what kind of environments fall into these constraints in practice, so I don't see how it could be applicable. I do not have a strong enough understanding of how reasonable is the completeness-style assumption.
* The experiment section needs more information. What are the environments? A comparison with another lifelong RL baseline should also be included to be convince readers of UCBlvd's benefits.


Minor:
* page 5 CMDP appears without being defined
* page 6, $\boldsymbol S_{++}^d $ is not defined
* page 6, "the algoriTheorem"?


**Summary Of The Paper:**

This work tackles the challenges of lifelong reinforcement learning. In particular, authors propose an algorithm (UCBvld) for solving sequential contextual Markov decision processes with linear representation which (1) guarantees sublinear regret with (2) a sublinear number of planning calls, even when the sequence of tasks and the initial states are chosen adversarially.

UCBvld has its foundation in LSVI-UCB, with an added distillation step based on a convex quadratically constrained quadratic program defined on a set of representative task contexts (assumed to be known).

**Summary Of The Review:**

The paper tackles an important gap in the RL literature. However, it seems to be of very limited applicability, without any direction for future extension provided.

---

> ### Author Response · Authors · 2022-11-08
> **Response to Reviewer hkRh**
>
> We thank the reviewer for their valuable time and feedback. Below we address your comments.
>
>
> **Linear MDPs:** Due to the curse of dimensionality, practical problems with an enormous number of states cannot be tackled by tabular RL and instead, they must use function approximation. The core RL question of how to design provably efficient RL algorithms that incorporate function approximation persists even in a basic setting with linear dynamics and rewards. Therefore, there has been a surging interest in designing RL algorithms using linear function approximation. We use linear function approximation in a lifelong setting to deal with infinitely many states and task contexts. In real-world problems, we can use the context to model the task specification of a problem. For example, if we want to design household robots to assist humans with a series of tasks like cooking, cleaning, washing dishes, lawn mowing, vacuuming, we can treat the the context as a natural language instruction that the human user would give to the robot, and we can view the feature mapping $\boldsymbol\psi$ and $\boldsymbol\phi$ as the embedding of a deep neural network model (e.g. using a transformer) that has been pretrained. This paradigm is quite commonly used in the robot learning literature.
>
>
> **Extension to nonlinear environments:** While the field of lifelong learning has been drastically growing in recent years, the number of theoretical results has not seen a similar trend **even for the basic case of linear MDPs**. In our work, we take the first step to establish theoretical guarantees for lifelong RL where both regret performance and computational efficiency is of paramount importance. It turns out that guaranteeing computation efficiency through reducing the number of planning calls even in this seemingly simple linear formulation is more intricate than a setting where computational efficiency is ignored. Moreover, to our knowledge, there has been no or limited research focusing on the computation aspect of nonlinear algorithms, even for the single-task setup. Therefore, studying linear MDPs in the setting of lifelong RL, where the computational efficiency matters, is an appropriate and important first step. That said, we acknowledge that the extension to nonlinear settings, potentially through the use of general function approximation tools, is an interesting and important future direction that requires a certain amount of research investment, and we hope that our work motivates further investigation in this direction. In particular, the extension of our distillation step and the doubling schedule (to reduce the computation cost) to a nonlinear setting is not obvious and requires careful and potentially complex technical considerations.
>
> **Completeness-style assumption:** Our Assumption 2 is a natural extension of existing completeness assumptions in single-task settings (Wang et al., 2020, Chen and Jiang 2020). Completeness is a common assumption in TD-based methods and is almost necessary to guarantee computation efficiency. In practice, this assumption may hold approximately when the features $\boldsymbol{\psi}$ and $\boldsymbol{\phi}$ are expressive. In light of this, in Remark 4 and Appendix F, we discuss this practical extension where the equality in Assumption 2 holds up to an error of $\zeta$ and how this error would affect our regret bound.
>
>
>
> **Experiments:** We implemented our main algorithm UCBlvd on synthetic environments, with details on the environments and their parameters given in Appendix H. We would like to emphasize that we did compare our method to a baseline RL algorithm, Lifelong-LSVI, which is an extension of LSVI-UCB (Jin et al., 2020) to the lifelong setting without considering computational restrictions. We highlight that to the best of our knowledge, our work is the first focusing on both regret and computational guarantees in the literature. The single-task version of the setting we considered here exactly matches that in Jin et al., 2020. Therefore, we figured the most fair comparison would be to the extension of LSVI-UCB to the lifelong setting.
>
> We thank the reviewer for bringing the typos to our attention. We fixed notation inconsistencies/typos in the revision.
>
>
> We hope that our answers above will address the reviewer's questions and concerns. We would be happy to answer any further questions.

---

> ### Author Response · Authors · 2022-11-13
> **We would appreciate feedback on our response so that there is time to answer any questions**
>
> Dear reviewer hkRh,
>
> We would like to thank you for your valuable time spent on our submission. We made an effort to respond to all your concerns.
> Since we are approaching the deadline for the Discussion Stage 1, we would much appreciate it if you could please let us know whether you still have any remaining issues regarding our submission and whether you are willing to reconsider your score.
>
> Thank you in advance for your time and attention.

---

> ### Author Response · Authors · 2022-11-18
> **We would appreciate your feedback**
>
> Dear Reviewer hkRh,
>
> Since there is only one day left for the Discussion Stage 1, we would appreciate it if you could please let us know whether you still have any remaining questions or concerns and we would be happy to engage in further discussions.
>
> Thanks,
> Authors

---

### Decision · Program_Chairs · 2023-01-20

**Decision:**

Accept: poster

**Justification For Why Not Higher Score:**

The novelty of the work is somewhat limited (seems to be a straightforward extension of the previous works).

**Justification For Why Not Lower Score:**

It is a well-executed work, nicely written, interesting setting, reasonable theory with some technical novelty, and some synthetic experiments to show that the method has potential in practice.

**Metareview: Summary, Strengths And Weaknesses:**

The reviewers like the paper but they are not excited about it. They see it as a well-executed work (nicely written, interesting setting, reasonable theory, and simple experiments to show that the proposed method is potentially useable in practice), but with somewhat limited novelty (seems to be a straightforward extension of the existing results). What they particularly like is that the authors take the computational cost into account and design and analyse a test (for the need to recompute the solution) that decreases the quadratic planning call to logarithmic.

**Note From Pc:**

if the above contains the word "oral" or "spotlight" please see: "oral" presentation means -> notable-top-5% and "spotlight" means -> notable-top-25%. As stated in our emails, we are disassociating presentation type from AC recommendations